# DALE: Generative Data Augmentation for Low-Resource Legal NLP

Sreyan Ghosh♠*   Chandra Kiran Evuru♠*   Sonal Kumar♠   S Ramaneswaran♥

S Sakshi♣   Utkarsh Tyagi♠   Dinesh Manocha♠

♠University of Maryland, College Park, USA,

♣UMass, Amherst, ♥NVIDIA, Bangalore, India

{sreyang, utkarsht, ckevuru, sonalkum, dmanocha}@umd.edu

fsakshi@umass.edu, ramanr@nvidia.com

## Abstract

We present **DALE**, a novel and effective generative **D**ata **A**ugmentation framework for low-resource **LE**gal NLP. DALE addresses the challenges existing frameworks pose in generating effective data augmentations of legal documents - legal language, with its specialized vocabulary and complex semantics, morphology, and syntax, does not benefit from data augmentations that merely rephrase the source sentence. To address this, DALE, built on an Encoder-Decoder Language Model, is pre-trained on a novel unsupervised text denoising objective based on *selective masking* - our masking strategy exploits the domain-specific language characteristics of templatized legal documents to mask collocated spans of text. Denoising these spans help DALE acquire knowledge about legal concepts, principles, and language usage. Consequently, it develops the ability to generate coherent and diverse augmentations with novel contexts. Finally, DALE performs conditional generation to generate synthetic augmentations for low-resource Legal NLP tasks. We demonstrate the effectiveness of DALE on 13 datasets spanning 6 tasks and 4 low-resource settings. DALE outperforms all our baselines, including LLMs, qualitatively and quantitatively, with improvements of 1%-50%.[1]

## 1   Introduction

With recent advances in deep learning for NLP, many systems have achieved state-of-the-art and near-human performance on benchmark Natural Language Understanding (NLU) datasets (Wang et al., 2018, 2019). Following this closely, the legal NLP literature has also been thriving with new datasets and frameworks (Chalkidis et al., 2021c; Niklaus et al., 2023; Chalkidis* et al., 2023). However, one common observation is that most techniques, built and evaluated on NLP tasks involving

---

[1]Code: https://github.com/Sreyan88/DALE

*These authors contributed equally to this work.

| Method | Original 1: Buyer has full power and authority to enter into this Agreement. Original 2: The Borrower is organized, validly existing and in good standing under the laws of the jurisdiction of its organization. |
|---|---|
| EDA (Wei and Zou) | 1: buyer has wide cut power and authority to enter into this agreement 2: the borrower is organized validly existing and in good standing under the laws the jurisdiction its organization |
| Legal-EDA (Perçin et al.) | 1: Purchaser has full-of-the-moon major power and self-assurance to enter into this agreement. 2: The borrower is organized, validly existing and in just stand up under the law of the legal power of its organization. |
| SSMBA (Ng et al.) | 1. buyer is full custody and agrees to enter into this agreement. 2: the borrower is organized, validly existing and in good peace under the laws in the jurisdiction or and organization |
| GENIUS (Guo et al.) | 1: Who has the authority to do this? 2: The Borrower is organized into three categories: validly existing, validly new, and validly old. The first category is new. The second category is old. |
| ChatGPT | 1: The buyer possesses complete authority to engage in this agreement. 2: The Borrower is legally established, currently active, and in compliance with the laws of the jurisdiction where it is organized. |
| **DALE** (ours) | 1: The Company has full power and authority to enter into this Agreement and to perform its obligations hereunder. 2: The Company is a corporation duly organized, validly existing and in good standing under the laws of the State of Delaware. |

Table 1: Comparison of augmentations generated using DALE and our baselines. DALE generates coherent and diverse augmentations in addition to introducing new context while preserving label consistency (1.Payments 2.Authority).

everyday natural language, do not easily transfer to the legal domain (Zhong et al., 2020a; Chalkidis et al., 2020; Katz et al., 2023). Legal language, also known as *legalese* and commonly classified as a "sublanguage" (Sinsheimer, 2007; Williams, 2007; Haigh, 2023), is governed by logical rules and is distinct from everyday natural language in terms of specialized vocabulary, morphology, complex syntax, and knowledge-specific semantics, which makes the transfer difficult. Interestingly, modern Large Language Models (LLMs), both open- and closed-source (like ChatGPT), that have shown to possess excellent reasoning abilities and achieved impressive performance in zero-shot NLU tasks (HuggingFace, 2023), often do not perform well in Legal Language Understanding (LLU) tasks (Chalkidis, 2023). With state-of-the-art instruction-tuned LLMs as our baselines, we also show that LLMs struggle to generate effective augmentations for LLU tasks and fail to preserve label consistency when the source legal document is long.

Improving the performance of deep learning

models on downstream LLU tasks requires sufficient good-quality training data. Beyond being an expensive and noisy task (Abad and Moschitti, 2016; Nguyen et al., 2017), high-quality annotation in specialized domains like legal or biomedical is prohibitively expensive due to the requirement of expert and requisite domain knowledge that lay annotators may not possess. One common approach taken by researchers for NLU tasks is data augmentation, either online (Guo et al., 2019; Ng et al., 2020a; Sun et al., 2020; Guo, 2020; Sawhney et al., 2021) or offline in the form of generated synthetic data (Wei and Zou, 2019; Kumar et al., 2020; Zhou et al., 2021; Kim et al., 2022; Guo et al., 2022a). Though most offline techniques perform well when employed for low-resource NLU tasks, we show that they tend to struggle in almost all LLU tasks, often generating in-coherent and non-diverse augmentations, eventually leading to sub-optimal performance. We attribute this to algorithmic biases of existing augmentation approaches towards natural language and the varying characteristics of legal language (see Section 2 for more details). For example, most of these techniques often just tend to rephrase the source document, which is ineffective for LLU tasks due to the formalized nature of legal language, adversely affecting both generation diversity and downstream model generalization. Longpre et al. also emphasize that task-agnostic augmentation frameworks lead to reduced performance. To overcome these issues, researchers in specialized domains (e.g., biomedical) have developed specialized algorithms (Kang et al., 2020; Ghosh et al., 2023), but to the best of our knowledge, no such approach has been proposed for the legal domain.

**Main Contributions.** In the paper, we present **DALE**, a novel data augmentation technique based on conditional generation for low-resource legal NLP. Based on our initial analysis of legal documents, we propose that augmentations enhancing LLU task performance can be achieved by *not* just rephrasing documents but also by modifying existing contexts or introducing novel ones. DALE, designed to perform this, builds on BART (Lewis et al., 2019) and is first pre-trained on a large-scale unlabeled legal corpus using a novel text denoising objective based on *selective masking*. Specifically, we leverage the inherent properties of templatized legal language to mask co-occurring and highly correlated spans of text in a legal document and avoid masking random and emerging entities

or facts. Our masking algorithm preserves valuable hints and prevents the model from learning redundant knowledge by *not* asking it to reconstruct document-specific entities or facts. Rather, it promotes acquiring broad legal knowledge and knowledge of legalese that enables DALE to advance its capability in generating augmentations of legal documents with novel contexts that possess remarkable levels of coherence and diversity. We call this masked document a *template*, and it serves as input to DALE for denoising-based pre-training. We optionally fine-tune DALE on the downstream dataset, followed by conditional generation to generate augmentations. We show that our domain-specific sentence corruption algorithm enables DALE to generate diverse and coherent augmentations of legal documents, which are entity-rich, semantically complex, and formal in nature. To summarize, our primary contributions are:

1. We propose DALE, the first generative data augmentation framework designed for low-resource legal NLP.

2. Through extensive empirical evaluation on 6 LLU tasks, 13 datasets, and 4 low-resource settings, we show that DALE outperforms all prior works with significant gains of 1%-50%.

3. Additionally, through extensive ablative experiments and qualitative comparison, we show that DALE generates much more diverse and coherent augmentations than prior works.

## 2  Related Work

**Legal NLP.** Recently, the legal NLP literature has been flourishing with new resources like datasets (Leitner et al., 2019; Zhong et al., 2020b; Zheng et al., 2021; Hendrycks et al., 2021), benchmarks (Chalkidis et al., 2021c; Niklaus et al., 2023; Chalkidis* et al., 2023) and PLMs (Chalkidis et al., 2020; Xiao et al., 2021; Mamakas et al., 2022; Niklaus and Giofré, 2022). However, despite much progress, the specialized domain of legal language lags behind in available resources when compared to natural language or domains like bio-medical (Katz et al., 2023). As also mentioned earlier, most techniques employed for building better deep learning NLU models do not transfer well to the legal domain due to characteristics that make it distinct from natural language (Morrison, 1989; Nair and Modani, 2023; Glogar, 2023), including its highly

Figure 1: Comparison of various span masking algorithms in legal documents rich in emerging entities and case-specific facts. **RM** stands for random masking, **GM** stands for GENIUS extreme masking (Guo et al., 2022a), **PMI** stands for PMI masking (Levine et al., 2021) and **DM** stands for *our* proposed DALE masking. Unlike other masking algorithms that make a model learn redundant knowledge through denoising entities or random tokens, our proposed masking formulation promotes learning of broader legal knowledge and legalese by masking co-occurring spans that consistently provide high signals.

formal, technical, entity-rich and knowledge-rich nature, along with semantically complex phrases. Simply put, the task of training machines to "understand" legal language has proven to be non-trivial (Katz et al., 2023). For quite some time, researchers tried to teach models to solve complex LLU problems through prior findings in NLU, e.g., pre-training LMs (Chalkidis et al., 2020). However, this has come with varying success (Zheng et al., 2021). Exploiting domain-specific characteristics to build custom pre-training strategies has shown better success (Nair and Modani, 2023; Chalkidis* et al., 2023), and we emphasize that there is a similar need for all tasks in legal NLP.

**Data Augmentation for Low-Resource NLP.** Data augmentation, both online (Guo et al., 2019; Ng et al., 2020a; Sun et al., 2020; Kumar et al., 2020; Guo, 2020; Sawhney et al., 2021) and offline (Wei and Zou, 2019; Kumar et al., 2020; Zhou et al., 2021; Kim et al., 2022; Guo et al., 2022a), has seen great success in overcoming the data scarcity issue in low-resource NLU tasks. While the former employs techniques like latent space interpolation or mixing, the latter is based on generating synthetic data that can be augmented with the original data to aid low-resource or few-shot learning (Chen et al., 2023). However, though the data scarcity issue is exacerbated in specialized domains like legal, where annotation becomes prohibitively expensive (Yang et al., 2019), domain-specific data augmentation techniques in literature are thin and almost non-existent, especially for the legal domain. Perçin et al. (2022) proposes the only legal domain-specific approach for data augmentation. However, they substitute phrases from the WordNet (Miller, 1995), failing to generate diverse augmentations for legal text by only editing common natural language phrases in the WordNet. For example, the

performance of back-translation (Yu et al., 2018) is affected by the inability of machine-translation systems to translate entity-rich and formal legal language effectively. The work closest to ours is Guo et al. (2022a) and Wang et al. (2022), where the PLM is trained on a keyword-to-sentence reconstruction task. However, these systems rely on unsupervised keyword discovery, which is naturally biased towards rare entities prevalent in legal documents. Denoising entities are case- or document-specific and would lead a model to learn redundant knowledge by reconstructing the case-specific fact around it, of which it has no prior knowledge. Without informed masking, a similar conclusion could be made for other PLM-based approaches in literature (Kumar et al., 2020; Guo et al., 2022a).

## 3 Methodology

### 3.1 DALE Pre-training

**Primary Goal.** Our primary goal is to devise a denoising-based seq-to-seq pre-training algorithm crafted to favor our final objective, i.e., generating diverse and coherent data augmentations. Sentence denoising is better suited to our task (compared to other methods like prompt- or instruction-tuning) as it gives us better control over long-document generations (explained further in Appendix E). The type of knowledge acquired through denoising objectives has been seen to be highly dependent on the masking algorithm (Sadeq et al., 2022). Thus, to achieve our objective and devise a suitable masking algorithm, we first try to answer a question crucial to the success of our approach: *Which attributes should an augmentation of a legal document possess to be considered effective, enabling improved generalization in downstream LLU tasks?* After conducting an analysis of legal documents, we hypothesize that formal language used in the

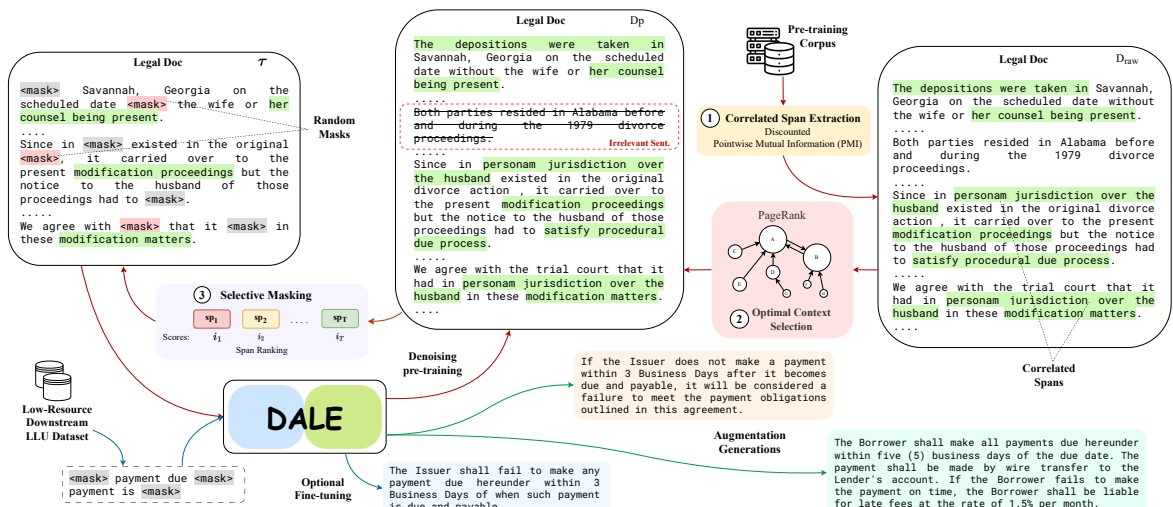

Figure 2: Illustration of **DALE**. ① We extract all correlated spans from a legal corpus using our discounted PMI formulation. ② We shorten a legal document by selecting only the top-*k* sentences that are the most relevant to the document and removing the rest. ③ We rank all the spans based on their importance and length using our novel scoring metric. Finally, we create a template by retaining the top-*p* spans and masking all other spans with added randomness. This process is followed by optional fine-tuning on the downstream dataset and conditional generation of augmentations from corrupted legal documents.

legal domain rarely allows for the occurrence of a rephrased version of the original document, unlike in everyday natural language. In fact, effective augmentations need to add new context to legal documents or modify existing ones.

**What to mask?** To modify the existing or introduce a novel context in legal documents while maintaining the formal legal style and plausibility of events in the generated context, DALE, like a legal practitioner, should possess both broad legal knowledge and knowledge of legalese. However, acquiring either from legal documents with complex semantics and syntax is not trivial. Legal documents, written by law practitioners, consist of clauses that are primarily document- (or case-) specific facts. The text is entity-rich, and entities are usually emerging as they are unique to that document. Beyond entities, these documents also contain text fragments outlining these entities and can be seen as an outcome of broad legal knowledge possessed by the practitioner. These co-occurring fragments, generally genre- or corpus-specific, are commonly reused by practitioners across documents. Their presence is a core property of legalese which can be attributed to its trait of being a formalized language (Nair and Modani, 2023). Fig. 1 shows an example sentence from a document with such a structure (more examples in Table 17). Thus, we hypothesize that learning to denoise these fragments with appropriate context and hints will eventually lead DALE to acquire knowledge about legal

concepts, principles, and language usage by consistently providing high signals and avoiding noise. This will in turn allow DALE to generate consistent, plausible, and diverse augmentations. Fig. 1 pictorially describes the problem with current masking algorithms and how our proposed algorithm favors our task. We call our final masked or corrupted document a *template* and denote it as $\mathcal{T}$. DALE pre-training involves multiple steps for template creation followed by training to denoise these templates. We next describe each step to create $\mathcal{T}$, which is done corpus-wise due to the variability of legalese across domains and genres.

**(1) Correlated Span Extraction.** To extract these reusable text fragments from an unlabeled legal corpus without supervision, we identify these fragments as correlated spans of tokens. First, we denote the set of all *n*-gram spans in a corpus C, as $N_C = \{n_0, \cdots, n_K\}$, where every span $n_k = \{w_1, \cdots, w_n\}$. Here *n* ranges from 2 to *q*. Our objective now is to extract a set of distinct spans $S_C = \{sp_0, \cdots, sp_T\}$ from $N_C$ where each span $sp_t$ exhibits high co-occurrence over the corpus. Though modeling such correlations is widely studied in computational linguistics (Zuidema, 2006; Ramisch et al., 2012), we choose to use Pointwise Mutual Information (PMI) (Fano, 1961) as a metric to score all individual *n*-grams in a corpus. PMI, by definition, quantifies how often two tokens occur, compared with what we would expect if they were independent. Our proposed strategy is based on the

PMI formulation proposed by Levine et al. (2021) that extends PMI to $n$-grams as follows:

$$\mathbf{PMI}_{(1,n)} = \min_{\sigma \in \mathrm{seg}(w_1 \ldots w_n)} \log \frac{p(w_1 \ldots w_k)}{\prod_{s \in \sigma} p(s)} \quad (1)$$

where $\mathbf{PMI}_{(1,n)}$ is the PMI for the $n$-gram $\{w_1, \cdots, w_n\}$ and $\mathrm{seg}(w_1, \cdots, w_n)$ is the set of all contiguous segmentations of the $n$-gram. We request our readers to refer to the original paper for more algorithmic details. However, this base formulation faces two main challenges when extended to legal documents: **(a)** The PMI formulation is designed to favor tokens with a lower frequency, making it choose rare tokens and not the text fragments of interest. This is further exacerbated by the fact that text in the legal domain is rich in case-specific, rare, and emerging entities.**(b)** There is no clear way to retain *hints* for reconstruction in the original formulation. Since legal language is highly domain-specific, not doing so might lead a model to hallucinate or training to collapse (Li et al., 2021; Sadeq et al., 2022). We describe how we overcome **(b)** in step **(3)**. To overcome **(a)**, we propose modifying the existing formulation by imposing a discounting factor to penalize rare tokens (Pantel and Lin, 2002). Thus, our modified formulation is as follows:

$$\mathbf{PMI}_{(1,n)} * \frac{\log f(w_1 \ldots w_n)}{\log(c) + \log f(w_1 \ldots w_n)} \quad (2)$$

where $f(.)$ is the frequency of occurrence of the $n$-gram, and $c$ is the constant factor used as a threshold to remove rare tokens. Precisely, $c$ refers to the minimum frequency of occurrence of an $n$-gram in the corpus below which the $n$-gram will be penalized. $c$ is calculated based on the density of rare tokens in the corpus and is usually set to the $pc^{th}$ percentile of the frequency distribution of all $n$-grams in the corpus. We choose $c$ specific to the value of $n$ in the $n$-gram in the specific corpus. Generally, PMI for datasets with a higher degree of rare entities per document is discounted with a $c$ corresponding to a frequency at a higher $pc$ (like Caselaw (cas, 2018) and Edgar (Henderson et al., 2022)). In contrast, datasets with a lower degree of entities or lower overall degree of formal language are discounted with a $c$ corresponding to a frequency at a lower $pc$ (like r/legaladvice (Henderson et al., 2022)). Finally, we select the top $j\%$ of $n$-grams with the lowest PMI score to construct $\mathrm{S_C}$. We provide more details in Appendix

B.1, including examples to show the effect of $c$ on correlated span extraction.

**(2) Optimal Context Selection.** Legal corpora, labeled and unlabeled, are generally structured at the granularity of document-level (collection of sentences). However, they are generally long (see Appendix H for dataset details), and denoising-based pre-training with an enc-dec model allows us to scale only to the maximum output sequence length $l_y$ of the decoder (irrespective of the encoder input sequence length). As mentioned earlier, LEGA employs BART-large with a maximum output sequence length of 1024 tokens (Appendix E explains the rationale behind our choice.). A common choice for such a scenario would be to just select the first $l_y$ tokens from the document $\mathrm{D_{raw}}$ to form a shorter document $\mathrm{D_p}$. However, this creates a text-informativeness mismatch between pre-training and fine-tuning instances, as raw legal documents have sparse information compared to fine-tuning instances (Sugathadasa et al., 2019). Thus, we choose to perform optimal context selection or select sentences from the document with a high informativeness measure. To this end, we propose to use the PageRank algorithm (Page et al., 1999), boosted by sentence similarity. Given a document $\mathrm{D_{raw}}$, with sentences $[\mathrm{s_0^{D_{raw}}}, \cdots, \mathrm{s_n^{D_{raw}}}]$, we use an encoder $\mathbf{E}_{pre}$ to calculate the embedding of each sentence $[\mathrm{e_{s_0}}, \cdots, \mathrm{e_{s_n}}]$ and the entire document $\mathrm{e_{D_{raw}}}$. This is followed by calculating the cosine similarity between every 2 sentences in the corpus, indexed $i$ and $j$, as follows:

$$s_{i,j} = \frac{\mathrm{e_{s_i}} \cdot \mathrm{e_{s_f}}}{\|\mathrm{e_{s_i}}\| \, \|\mathrm{e_{s_f}}\|} \quad (3)$$

where $i, j \in \{1, \cdots, n\}$ and $\mathrm{e_{s_f}}$ is defined as $\mathrm{e_{s_f}} = \lambda \mathrm{e_{s_j}} + (1 - \lambda)\mathrm{e_{D_{raw}}}$. Post this step; we construct an n × n similarity matrix, which serves as an adjacency matrix for constructing a graph $\mathcal{G} = (\mathcal{V}, \mathcal{E})$ where the sentences form the vertices $\mathcal{V}$ and the similarity scores form the edges $\mathcal{E}$. Finally, we apply $\mathrm{PageRank}(\mathcal{G})$ to assign every sentence an importance score and select the top-$k$ sentences not exceeding 1024 tokens. Following this, we sort the sentences in the document's original order of occurrence. We sample a probability $\varepsilon$ from a Gaussian distribution $\mathcal{N}(\mu, \sigma^2)$, and only do this step if $\varepsilon$ crosses a set threshold $\beta$.

**(3) Selective Masking.** Once we obtained the set of correlated spans $\mathrm{S_C}$ from step **(1)** and $\mathrm{D_p}$ from step **(2)**, we now want to select the best candidates for

masking from all spans in $S_{D_p}$. $S_{D_p}$ are the spans in $S_C$ only present in document $D_p$. To this end, we devise a novel span-ranking metric to construct our template such that we preserve valuable hints but also prefer longer spans. Formally put, we first use a pre-trained encoder $\mathbf{E}_{pre}$ to calculate the embedding of each span as $[e_{sp_0}, \cdots, e_{sp_T}]$ and the entire document as $e_{D_p}$ followed by assigning an importance score $i_t$ to each span $sp_t$ as follows:

$$i_t = \frac{\text{sim}(e_{sp_t}, e_{D_p})}{\text{norm}(\text{len}(sp_t))} \tag{4}$$

where $\text{sim}(.)$ is the cosine similarity between each $e_{sp_t}$ and $e_D$ calculated similarly to Eqtn. 3. The denominator is the length of the span normalized across all spans in $S_{D_p}$ to assign higher importance to smaller spans. Finally, to create our template, we preserve the top-$p$ spans in S, not exceeding 20% of the entire document length, and mask all other spans in $S_{D_p}$. Finally, Each span is replaced by a single mask token. To introduce randomness into the process, we sample a probability $\gamma$ from a Gaussian distribution $\mathcal{N}(\mu, \sigma^2)$ and randomly preserve a token in a contiguous span of tokens to be masked if $\gamma$ crosses a set threshold $\alpha$. After obtaining template $\mathcal{T}$ for all documents in the corpus for all corpora, we pre-train DALE on the denoising objective to reconstruct $D_p$ from $\mathcal{T}$.

## 3.2 DALE Fine-tuning

Though pre-trained DALE serves as an effective general-purpose data augmentation model for low-resource LLU tasks, we prefer to fine-tune BART on our downstream dataset so that our generated augmentations exhibit an underlying data distribution similar to our gold dataset. This has been seen as critical to improving in-domain performance with scale (Geiping et al., 2023). However, extracting correlated spans with PMI from fine-tuning datasets with few samples is generally ineffective as PMI becomes effective only with scale (Fano, 1961). Thus, to construct a template, we extract all $n$-grams N = $\{n_0, \cdots, n_t, \cdots, n_T\}$ from a particular document (or training instance) $D_f$ and assign an importance score to each by calculating cosine similarity, similar to Eqtn. 3, between $\mathbf{E}_{pre}(n_t)$ and $(\lambda \times \mathbf{E}_{pre}(D_f) + (1 - \lambda) \times \mathbf{E}_{pre}(L_{D_f}))$. $L_{D_f}$ here is the label for the document $D_f$. We elaborate in Appendix I.1 on how we construct $L_{D_f}$ for tasks beyond multi-class classification. Finally, we preserve the top-$p$ $n$-grams and mask everything else in the sentence, before merging consecutive

masks. For datasets with documents exceeding 1024 tokens, we propose a sliding window mechanism for fine-tuning. Specifically, with a window of size $w$ tokens, we break down a long sequence into its constituent segments of 1024 tokens, with each segment beyond the initial segment having additional non-masked context from the previous window. This context is additionally bounded between special tokens <context> and </context> to provide the model with explicit supervision. We provide a detailed explanation in Appendix D on why the DALE fine-tuning masking algorithm is not well suited for pre-training and better fits the fine-tuning stage.

## 3.3 DALE Generation

To generate data augmentations using DALE, we construct a template by corrupting a sentence similar to the fine-tuning stage and condition it to the model to generate augmentations. We use beam search with random multinomial sampling to generate diverse augmentations. Finally, we employ a sliding window mechanism for long documents, combining outputs from all sliding window segments for the final augmentation. After generating augmentations, we add them to the gold annotated data to fine-tune our downstream evaluation model.

## 4 Experiments and Results

### 4.1 Tasks and Datasets

**Pre-training.** To pre-train DALE, we use a combination of multiple datasets from Pile of Law (Henderson et al., 2022), CaseLaw (cas, 2018), and MAUD (Wang et al., 2023). The final pre-training corpus comprised $\approx$ 4.1M documents amounting to $\approx$ 48GB. Detailed statistics are in Appendix H.

**Downstream Evaluation.** To prove the efficacy of DALE, we conducted experiments on 13 legal datasets based on 6 tasks across 4 low-resource settings. These tasks include Multi-class classification (MCC), Multi-label classification (MLC), Named Entity Recognition (NER), Multiple choice QA (MCQ) (identify the correct (masked) holding statement from a selection of choices), Rhetorical Role Prediction (RR) (sequential text classification for assigning a label to each sentence in a legal document for semantic document segmentation), and Document-level NLI (DLI). For MCC, we experiment on SCOTUS (Spaeth et al., 2013), LEDGAR (Tuggener et al., 2020), ILDC (Malik et al., 2021) and OTS-UL (Drawzeski et al., 2021)

| #Gold | 100 | 200 | 500 | 1000 | 100 | 200 | 500 | 1000 | 100 | 200 | 500 | 1000 | 100 | 200 | 500 | 1000 | 100 | 200 | 500 | 1000 |
|---|---|---|---|---|---|---|---|---|---|---|---|---|---|---|---|---|---|---|---|---|
| **Dataset** | | OTS-TOPICS | | | | EUR-LEX | | | | ECtHR-A | | | | ECtHR-B | | | | UNFAIR-ToS | | |
| Gold-only | 0.10 | 11.47 | 51.16 | 53.87 | 8.68 | 4.30 | 10.32 | 42.26 | 25.26 | 27.30 | 17.14 | 31.52 | 37.69 | 47.47 | 44.89 | 50.98 | 0.10 | 33.88 | 70.02 | 76.21 |
| EDA | 9.72 | 38.43 | 37.56 | 46.99 | 12.11 | 22.93 | 49.26 | 51.54 | 10.10 | 35.64 | 41.91 | 49.67 | 43.01 | 48.70 | 56.32 | 59.40 | 13.93 | 26.31 | 72.15 | 78.14 |
| Legal-EDA | 10.10 | 39.15 | 40.40 | 50.48 | 12.45 | 23.61 | 51.24 | 53.27 | 12.24 | 36.75 | 43.89 | 52.93 | 43.86 | 54.72 | 57.71 | 61.53 | 15.86 | 27.54 | 72.98 | 78.69 |
| SSMBA | 10.41 | 15.28 | 47.31 | 52.63 | 4.10 | 21.32 | 45.67 | 48.70 | 7.55 | 18.10 | 34.39 | 37.58 | 35.32 | 45.43 | 48.08 | 52.65 | 6.53 | 18.21 | 63.96 | 68.59 |
| AEDA | 14.06 | 52.63 | 60.29 | 72.32 | 3.07 | 33.33 | 50.33 | 52.21 | 28.12 | 30.94 | 32.29 | 45.48 | 39.15 | 50.85 | 50.48 | 51.26 | 8.08 | 52.34 | 70.48 | 73.67 |
| SMERTI | 3.41 | 17.90 | 57.26 | 60.54 | 6.62 | 27.86 | 44.45 | 47.68 | 28.51 | 22.61 | 23.43 | 38.59 | 38.43 | 51.02 | 52.07 | 53.71 | 20.46 | 47.31 | 59.38 | 69.27 |
| BackTrans | 8.26 | 37.44 | 47.47 | 50.85 | 5.03 | 19.63 | 37.86 | 42.65 | 14.73 | 17.37 | 35.36 | 39.41 | 37.61 | 49.88 | 50.77 | 52.83 | 12.84 | 39.28 | 46.51 | 62.64 |
| C-MLM | 3.85 | 17.95 | 58.54 | 61.45 | 7.17 | 28.21 | 45.04 | 47.85 | 27.95 | 23.24 | 23.89 | 39.23 | 39.46 | 52.17 | 53.26 | 54.68 | 20.42 | 48.52 | 59.87 | 69.62 |
| GENIUS | 25.58 | 54.31 | 63.71 | 67.29 | 5.79 | 34.03 | 53.19 | 57.95 | 28.68 | 28.66 | 36.38 | 43.67 | 40.40 | 44.03 | 50.54 | 54.29 | 11.20 | 47.18 | 67.71 | 75.79 |
| ChatGPT | 23.42 | 53.31 | 62.17 | 65.87 | 5.52 | 33.22 | 52.21 | 56.45 | 27.52 | 27.89 | 34.03 | 41.83 | 39.61 | 43.12 | 49.76 | 53.87 | 10.78 | 44.62 | 65.87 | 72.91 |
| Falcon | 12.36 | 37.84 | 48.66 | 51.74 | 5.11 | 22.02 | 46.19 | 49.03 | 17.68 | 20.39 | 35.81 | 38.62 | 36.12 | 46.53 | 47.27 | 53.85 | 5.44 | 16.10 | 62.82 | 67.51 |
| DALE-BART | 25.77 | 54.01 | 58.29 | 68.04 | 12.32 | 34.39 | 53.65 | 56.27 | 23.01 | 35.68 | 40.13 | 52.47 | 43.91 | 52.76 | 54.58 | 60.24 | 18.43 | 46.60 | 68.21 | 75.04 |
| DALE-pt | 24.58 | 52.17 | 58.18 | 69.97 | 11.50 | 29.51 | 51.63 | 53.12 | 24.19 | 33.87 | 40.87 | 48.85 | 42.97 | 51.67 | 51.63 | 59.23 | 18.54 | 47.59 | 63.21 | 73.56 |
| DALE-ft | 24.63 | 53.22 | 59.64 | 70.15 | 11.61 | 33.54 | 52.38 | 57.62 | 24.21 | 34.76 | 41.78 | 51.65 | 43.33 | 53.74 | 55.12 | 60.95 | 19.11 | 48.71 | 67.42 | 74.86 |
| **DALE** (ours) | **33.91** | **61.23** | **71.56** | **73.24** | **13.50** | **37.93** | **55.99** | **59.45** | **29.43** | **37.57** | **44.38** | **55.72** | **46.72** | **56.13** | **59.18** | **64.01** | **22.32** | **54.62** | **74.84** | **82.98** |

Table 2: Results for Multi-label classification. DALE outperforms baselines by 1%-49.8%.

| #Gold | 100 | 200 | 500 | 1000 | 100 | 200 | 500 | 1000 | 100 | 200 | 500 | 1000 | 100 | 200 | 500 | 1000 |
|---|---|---|---|---|---|---|---|---|---|---|---|---|---|---|---|---|
| | | LEDGAR | | | | ILDC | | | | SCOTUS | | | | OTS | | |
| Gold-only | 22.65 | 61.39 | 71.43 | 75.13 | 51.48 | 54.24 | 55.83 | 58.03 | 63.69 | 65.93 | 70.75 | 75.92 | 66.72 | 68.59 | 70.21 | 72.54 |
| EDA | 42.65 | 59.31 | 72.34 | 75.76 | 49.76 | 49.83 | 59.32 | 61.72 | 53.00 | 61.57 | 72.51 | 73.29 | 68.93 | 69.66 | 72.13 | 73.28 |
| Legal-EDA | 53.00 | 60.57 | 73.28 | 76.72 | 52.15 | 52.23 | 60.38 | 62.27 | 55.21 | 61.39 | 73.69 | 75.57 | 69.51 | 71.67 | 76.31 | 79.72 |
| SSMBA | 47.86 | 60.34 | 70.06 | 74.21 | 47.62 | 50.21 | 58.53 | 60.12 | 43.00 | 60.57 | 72.51 | 76.26 | 60.12 | 70.17 | 75.47 | 76.04 |
| AEDA | 46.99 | 58.06 | 71.01 | 75.35 | 48.93 | 49.62 | 56.36 | 59.05 | 62.15 | 62.65 | 71.24 | 73.55 | 61.29 | 67.08 | 74.26 | 81.26 |
| SMERTI | 33.23 | 60.65 | 62.24 | 67.25 | 42.34 | 44.82 | 51.27 | 58.73 | 63.78 | 66.71 | 70.92 | 71.57 | 66.99 | 68.72 | 76.58 | 80.58 |
| BackTrans | 51.23 | 58.96 | 63.84 | 69.04 | 40.72 | 41.33 | 59.18 | 62.01 | 42.01 | 45.63 | 57.22 | 67.56 | 59.69 | 65.81 | 66.23 | 71.53 |
| C-MLM | 34.12 | 60.95 | 63.11 | 68.15 | 43.18 | 45.65 | 52.01 | 58.98 | 61.56 | 65.54 | 71.25 | 71.95 | 67.05 | 68.97 | 77.52 | 79.62 |
| GENIUS | 48.76 | 62.14 | 71.17 | 74.48 | 51.35 | 54.26 | 53.39 | 52.14 | 59.42 | 61.71 | 63.14 | 70.28 | 66.71 | 68.65 | 76.20 | 79.73 |
| GPT3-Mix | 30.37 | 58.74 | 61.62 | 66.44 | 41.87 | 43.73 | 50.45 | 57.52 | 63.42 | 65.82 | 70.87 | 71.03 | 66.73 | 67.53 | 77.07 | 79.21 |
| PromDA | 45.76 | 51.24 | 65.40 | 68.27 | 41.30 | 43.08 | 49.21 | 51.27 | 44.59 | 53.86 | 59.72 | 61.58 | 63.72 | 65.73 | 70.38 | 73.28 |
| ChatGPT | 46.87 | 61.18 | 70.41 | 73.92 | 50.74 | 52.93 | 52.34 | 51.21 | 58.69 | 60.56 | 62.81 | 69.40 | 65.01 | 67.88 | 75.32 | 78.19 |
| Falcon | 43.07 | 58.32 | 68.48 | 73.62 | 46.29 | 48.27 | 57.83 | 58.03 | 42.11 | 59.83 | 60.32 | 70.54 | 59.19 | 66.25 | 73.17 | 75.08 |
| DALE-BART | 50.95 | 57.90 | 64.28 | 70.87 | 52.26 | 51.54 | 54.31 | 62.68 | 60.01 | 65.27 | 62.02 | 72.13 | 69.12 | 70.89 | 71.99 | 77.97 |
| DALE-pt | 48.26 | 55.39 | 65.27 | 67.94 | 52.02 | 51.87 | 57.26 | 58.51 | 59.61 | 63.25 | 66.72 | 68.85 | 69.93 | 70.21 | 73.68 | 75.89 |
| DALE-ft | 52.01 | 58.67 | 68.38 | 72.24 | 52.14 | 53.88 | 58.15 | 61.92 | 59.70 | 64.62 | 65.46 | 72.41 | 68.85 | 70.91 | 74.31 | 77.58 |
| **DALE** (ours) | **55.13** | **63.76** | **74.89** | **78.36** | **54.47** | **55.95** | **62.45** | **63.11** | **65.85** | **67.86** | **74.89** | **78.96** | **71.64** | **72.89** | **77.74** | **83.75** |

Table 3: Results for Multi-class classification. DALE outperforms baselines by 1%-49.8%.

datasets. For MLC, we experiment on ECtHR Task A and B (Chalkidis et al., 2019, 2021b), EUR-LEX (Chalkidis et al., 2021a), UNFAIR-ToS (Lippi et al., 2019) and OTS-CT (Drawzeski et al., 2021) datasets. For NER, we experiment on EDGAR (Au et al., 2022), and the Indian-Legal-NER (Kalamkar et al., 2022) datasets. For RR, we experiment on the BUILD dataset (Malik et al., 2022). Finally, for DLI, we experiment on the ContractNLI (Koreeda and Manning, 2021). We perform class-balanced sampling to create low-resource splits and down-sample the dev set accordingly. Dataset statistics are in Appendix H. We report micro-averaged $F_1$ scores averaged across 3 runs for 3 random seeds.

## 4.2 Experimental Setup

**DALE.** As mentioned earlier, we use BART-large (Lewis et al., 2019) as our encoder-decoder model for training DALE. We detail in Appendix E why we think BART$_{large}$ is the most suitable for our task and setup. We pre-train DALE for 5 epochs using Adam optimizer with a learning rate of $1e^{-5}$ and

a batch size of 32. We use the same setting for fine-tuning DALE but with a learning rate of $1e^{-3}$.

**Downstream Task-Specific Setups.** For down-stream task-specific evaluation, we fine-tune legal-longformer$_{large}$ (Chalkidis* et al., 2023). For fine-tuning legal-longformer$_{large}$, we fine-tune for 20 epochs with a batch size of 16 using Adam optimizer with a learning rate of $1e^{-5}$.

Details about the hyper-parameter setup for our experiments can be found in Appendix B including hyper-parameter tuning experiments.

## 4.3 Baselines

Details on the working of each baseline can be found in Appendix F.

**Gold-only Baseline.** This baseline is common across tasks and uses only gold data without any additional augmentations.

**Classification Baselines.** For MLC, we compare DALE against EDA (Wei and Zou, 2019), Legal-EDA (Perçin et al., 2022), GENIUS(-**ft**) (Guo et al.,

| #Gold | 100 | 200 | 500 | 1000 | 100 | 200 | 100 | 200 |
|---|---|---|---|---|---|---|---|---|
| **Dataset** | | CaseHOLD | | | BUILD-RR | | ContractNLI | |
| Gold-only | 33.92 | 66.38 | 70.06 | 70.80 | 74.62 | 78.24 | 72.03 | 82.06 |
| EDA | 56.38 | 64.71 | 66.42 | 69.45 | 77.33 | 81.83 | 73.92 | 75.40 |
| AEDA | 57.96 | 65.10 | 69.12 | 70.05 | 77.95 | 82.01 | 77.24 | 83.02 |
| SSMBA | 62.01 | 67.65 | 69.59 | 69.75 | 77.77 | 81.66 | 76.27 | 82.93 |
| SMERTI | 56.52 | 64.13 | 69.15 | 69.85 | 77.42 | 80.65 | 76.23 | 81.95 |
| BackTrans | 55.69 | 65.72 | 69.29 | 69.74 | 77.59 | 81.08 | 75.98 | 81.19 |
| GENIUS | 55.84 | 61.37 | 64.17 | 68.20 | 78.99 | 79.30 | 77.28 | 81.28 |
| ChatGPT | 54.67 | 60.83 | 62.57 | 67.59 | 77.32 | 78.37 | 76.29 | 80.10 |
| Falcon | 52.57 | 58.76 | 62.41 | 63.22 | 75.11 | 77.61 | 75.17 | 77.54 |
| DALE-BART | 61.21 | 66.09 | 67.91 | 70.64 | 78.59 | 80.01 | 76.56 | 81.27 |
| DALE-pt | 59.25 | 65.69 | 67.81 | 69.70 | 78.15 | 79.01 | 76.97 | 80.55 |
| DALE-ft | 60.31 | 66.56 | 68.46 | 70.15 | 78.50 | 79.72 | 77.10 | 81.73 |
| **DALE** (ours) | 63.71 | 68.14 | 71.53 | 72.70 | 81.83 | 83.04 | 79.26 | 85.13 |

Table 4: Results for MCQA (CaseHOLD), RR (BUILD-RR), and DLI (ContractNLI). DALE outperforms by 0.5%-29.8%.

| #Gold | 100 | 200 | 500 | 1000 | 100 | 200 | 500 | 1000 |
|---|---|---|---|---|---|---|---|---|
| **Baselines** | | EDGAR | | | | INDIAN LEGAL NER | | |
| Gold-only | 0.75 | 0.27 | 34.86 | 57.84 | 8.41 | 13.61 | 33.28 | 42.6 |
| LwTR | 22.10 | 36.84 | 50.33 | 54.15 | 12.53 | 17.87 | 35.54 | 44.15 |
| DAGA | 13.21 | 24.54 | 36.15 | 42.58 | 5.13 | 14.52 | 26.13 | 31.74 |
| MulDA | 8.17 | 21.33 | 42.61 | 50.16 | 13.75 | 19.28 | 31.96 | 40.69 |
| MR | 19.13 | 36.62 | 50.95 | 58.33 | 18.62 | 25.26 | 43.14 | 49.68 |
| MELM | 12.32 | 24.35 | 48.72 | 60.59 | 14.55 | 21.69 | 38.73 | 48.64 |
| GENIUS | 13.79 | 28.44 | 50.93 | 62.69 | 19.05 | 29.28 | 48.72 | 53.61 |
| PromDA | 10.10 | 27.31 | 45.77 | 55.62 | 16.46 | 26.91 | 45.34 | 44.62 |
| ChatGPT | 12.65 | 26.32 | 49.25 | 60.67 | 18.24 | 27.58 | 46.44 | 51.41 |
| Falcom | 11.24 | 25.71 | 48.69 | 59.84 | 18.11 | 26.23 | 43.05 | 49.38 |
| DALE-BART | 17.76 | 34.20 | 48.71 | 57.99 | 16.43 | 29.19 | 46.03 | 49.96 |
| DALE-pt | 18.38 | 33.12 | 47.67 | 53.67 | 17.25 | 27.86 | 45.57 | 48.28 |
| DALE-ft | 19.10 | 35.39 | 48.20 | 58.74 | 17.65 | 28.32 | 46.71 | 49.98 |
| **DALE** (ours) | 23.65 | 39.82 | 55.99 | 64.32 | 21.31 | 32.47 | 49.93 | 54.27 |

Table 5: Results for NER. DALE outperforms by 1% - 39.6%.

| Method | Perplexity(↓) | Diversity(↑) | Diversity-L(↑) | Perplexity(↓) | Diversity(↑) | Diversity-L(↑) |
|---|---|---|---|---|---|---|
| | | 200 | | | 500 | |
| EDA | 82.22 | 12.49 | 83.48 | 86.14 | 12.72 | 86.28 |
| Legal-EDA | 55.38 | 25.71 | 13.51 | 58.92 | 26.70 | 14.26 |
| SSMBA | 37.96 | 54.74 | 17.74 | 37.84 | 56.85 | 19.29 |
| AEDA | 26.93 | 2.17 | 176.68 | 27.05 | 13.67 | 145.13 |
| SMERTI | 28.56 | 56.84 | 13.76 | 29.20 | 59.62 | 14.58 |
| BackTrans | 27.94 | 45.05 | 27.62 | 27.85 | 49.05 | 28.62 |
| C-MLM | 50.39 | 41.04 | 23.85 | 51.69 | 44.86 | 25.69 |
| GENIUS | 24.37 | 106.08 | 226.65 | 24.65 | 105.04 | 278.64 |
| GPT3-Mix | 52.76 | 42.21 | 29.74 | 53.21 | 45.73 | 33.68 |
| PromDA | 174.67 | 65.69 | 15.74 | 187.68 | 73.93 | 16.84 |
| LWTR | 481.34 | 86.91 | 49.87 | 413.66 | 76.37 | 21.42 |
| MR | 82.72 | 75.65 | 29.23 | 79.65 | 81.46 | 32.76 |
| MELM | 211.94 | 12.49 | 83.48 | 183.23 | 12.72 | 86.28 |
| ChatGPT | 26.29 | 64.31 | 32.85 | 26.17 | 66.94 | 35.85 |
| Falcon | 45.24 | 13.64 | 17.63 | 44.97 | 15.74 | 18.59 |
| DALE-BART | 20.36 | 172.54 | 222.37 | 21.65 | 193.32 | 231.86 |
| DALE-pt | 58.09 | 66.99 | 260.00 | 60.12 | 59.84 | 294.05 |
| DALE-ft | 18.75 | 149.77 | 219.22 | 20.21 | 156.54 | 200.99 |
| **DALE** (ours) | 18.63 | 175.38 | 227.39 | 18.44 | 194.20 | 234.86 |

Table 6: Quantitative evaluation of generation quality on the measures of perplexity, token diversity (Diversity), and length diversity (Diversity-L). DALE outperforms all our baselines.

2022a), SSMBA (Ng et al., 2020b), AEDA (Karimi et al., 2021), SMERTI (Feng et al., 2019), Back-Trans (Yu et al., 2018), C-MLM (Kumar et al., 2020), ChatGPT (Dai et al., 2023) and instruction-tuned Falcon (Penedo et al., 2023). For MCC, we add to this list GPT3-Mix (Yoo et al., 2021) and PromDA (Wang et al., 2022). Since GENIUS and C-MLM involve pre-training, we pre-trained it on our data with their respective masking algorithms.

**Other Task Baselines** For NER, we compare DALE against LwTR (Dai and Adel, 2020), DAGA (Ding et al., 2020), MulDA (Liu et al., 2021), MELM (Zhou et al., 2022b), PromDA , ChatGPT and instruction-tuned Falcon. For RR, DLI and MCQA, we compare DALE against EDA, GE-NIUS, SSMBA, AEDA, and BackTrans.

**DALE Ablations.** To evaluate the effectiveness of the core steps in the DALE augmentation framework, we also compare DALE with other baselines on DALE-pt (augmentations generated with only a pre-trained DALE without any fine-tuning) and DALE-ft (augmentations generated with only a fine-tuned Legal-BART without DALE Pre-training). DALE-BART is DALE pre-trained on Pile-of-Law with random masking. We provide additional results in Appendix B.

## 4.4 Results

**Quantitative Comparison.** Table 3 compares the performance of DALE with other baselines on MCC (top-row) and MLC (bottom-row). DALE outperforms baselines with absolute improvements in the range of 1%-32.5% for MLC and 1%-49.8% for MCC. Table 5 compares the performance of DALE with other baselines on NER. DALE outperforms baselines with absolute improvements in the range of 1%-39.6%. Table 4 compares the perfor-

mance of DALE with other baselines on MCQA, RR, and DLI. DALE outperforms baselines with absolute improvements in the range of 0.5%-29.8% in MCQA, 1%-7.2% in RR, and 2%-9.7% in DLI. DALE-BART performs similarly to DALE-ft and is inferior to DALE, thereby showing the ineffectiveness of random masking for the legal domain.

**Qualitative Comparison.** Table 6 compares the generation quality of DALE with all our baselines (averaged baseline-wise across all tasks and splits) on the measures of perplexity (Jelinek et al., 1977), diversity (average number of new tokens introduced in $R$ augmentations) and length diversity (average absolute difference in length of source and $R$ augmentations). DALE outperforms most of our baselines in all settings. DALE-pt generates more diverse augmentations but at the cost of not maintaining underlying data distribution. Beyond Table 1, Table 18 provides more augmentation examples. Contrary to our baselines, that are too conservative or too aggressive, DALE, especially for long documents, generates augmentations that are diverse, coherent, and consistent with the source label.

Table 7: Comparison of augmentations generated by DALE and all other baselines for the UNFAIR TOS dataset. All augmentations were generated in a low-resource setting (500). Each augmentation was marked by a law student on 3 parameters: (1) If the augmentation is coherent, (2) If it adds new plausible context, and (3) if it is label-consistent and matches the underlying data distribution. We present the results of the study as ✓or ✗next to each augmentation in the same order as above. Pink signifies the change from the Original. More examples can be found in Table 18.

| UNFAIR ToS | |
|---|---|
| Original | The most recent version of this agreement will be posted on the services under settings and also on gotinder.com, and you should regularly check for the most recent version. |
| EDA | recent version of this agreement will be posted on the services under settings and also on gotinder com and you should regularly check for the most recent version ✗  ✗  ✓ |
| AEDA | the most ; recent version of ; this agreement will be posted on the , services under settings and also on gotinder.com . , and you should regularly check for the most recent version . , ✗  ✗  ✓ |
| SMERTI | This most recent version of Windows will be posted on power under settings available on gotinder. , and you should regularly check our most recent version. ✗  ✗  ✗ |
| GENIUS | The terms of this agreement will be contingent on the services they provide. For more information, please visit www.sos.gov. ✓  ✗  ✗ |
| ChatGPT | The latest edition of this agreement will be made available on the services, specifically under the settings section and on gotinder.com. It is advisable to frequently review the most recent version. ✓  ✗  ✓ |
| Falcon | The most recent version of this agreement will be posted on the services under settings and also on gotinder.com, and you should regularly check for the most recent version. ✓  ✗  ✓ |
| DALE-pt | The most recent version of this agreement shall be accepted as the most recent amendment . ✓  ✗  ✗ |
| DALE-ft | the most recent version of this agreement will be posted on the services under settings and also on gotinder.com, and you should regularly check for the most most recent versions. ✓  ✗  ✓ |
| DALE | The most recent version of this agreement will be posted on the services's website at https://www.adr.nianticlabs.com/ where you can download and view the services, and you should be aware that this is not a guarantee that the services will be up to code or up to date, and we reserve the right to discontinue using the services at any time. ✓  ✓  ✓ |

# 5  Conclusion

This paper presents DALE, a novel generative data augmentation framework for low-resource legal NLP. We evaluate DALE on 13 datasets spanning across 6 tasks under 4 low-resource settings and show that DALE outperforms all prior art quantitatively and qualitatively by a significant margin.

# Acknowledgement

This work was supported by ARO grants W911NF2310352 and W911NF2110026.

# Limitations and Future Work

In this section, we list down some potential limitations of DALE:

1. DALE is still restricted to generating augmentations for legal datasets that consist of documents only in English. Though English is prevalent in the legal literature across domains and genres, recent work shows the importance of multi-lingual legal language modeling (Niklaus et al., 2023). As part of future work, we would like to overcome this shortcoming by introducing multi-lingual DALE.

2. At extreme low-resource scenarios, DALE accompanied by optional fine-tuning might be prone to over-fitting, generating almost similar augmentations. Though using pre-trained DALE can overcome this problem, our experiments clearly show the benefits of fine-tuning. Thus, as part of future work, we would

like to explore the combination of augmentations generated by pre-training and fine-tuned DALE.

3. Our masking algorithm involves PMI, which is beneficial only at scale. Though benefiting from scale is an inherent property of pre-training, we would like to explore possible ways to overcome this problem.

## Ethics Statement

We acknowledge that augmentations generated by DALE might not be always factual, i.e., contain events that have occurred in the real world. However, DALE is not directly meant for helping a legal practitioner in his everyday practice through its generations. Instead, DALE is meant for only generating augmentations to help train downstream models that can help legal practitioners in their practice.

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

# A  Algorithm

We show DALE algorithmically in Algorithm 1.

---

**Algorithm 1** DALE: Our proposed augmentation framework

---

**Given** pre-training dataset C, Enc-Dec PLM $\mathcal{L}$ and Enc-only PLM $\mathcal{P}$
$C_{masked} \leftarrow \varnothing$
$N_C \leftarrow C$       ▷Extract all $n$-grams
$S_C \leftarrow N_C$     ▷Extract all correlated spans from $n$-grams
$S_C \leftarrow S_C$        ▷Select only top $j\%$
**for** $D_{raw} \in C$ **do**       ▷Masking Loop
 $D_P \leftarrow D_{raw}$     ▷Optimal Context Selection
 $S_{D_P} \leftarrow S_C$     ▷Filter only spans present in $D_P$
 Rank all spans in $S_{D_P}$
 $\mathcal{T} \leftarrow D_P$ Keep top-$p$ spans and mask the rest   ▷Selective Masking
**end for**
Pre-train $\mathcal{L}$ with denoising to reconstruct $D_p$ from $\mathcal{T}$
**Given** low-resource fine-tuning dataset $D_{train}$, and DALE   ▷Optional FT
**for** $\{X, Y\} \in \mathbb{D}_{t}rain$ **do**
 $\mathcal{T} \leftarrow X$        ▷Selective Masking
**end for**
Fine-tune $\mathcal{L}$ with denoising to reconstruct $X$ from $\mathcal{T}$
**for** $\{X, Y\} \in \mathbb{D}_{train}$ **do**     ▷Generation Loop
 **repeat** $\mathcal{R}$ **times**:
 $\mathcal{T} \leftarrow X$        ▷Selective masking
 $X_{aug} \leftarrow \text{GENAUG}(\text{DALE}(\mathcal{T}))$   ▷Generate augmented data
  $\mathbb{D}_{aug} \leftarrow \mathbb{D}_{aug} \cup \{X_{aug}\}$
**end for**
Fine-tune $\mathcal{P}$ with $\mathbb{D}_{aug}$
return $\mathcal{P}$

---

# B  Hyperparameter Tuning

**Hyperparameters.** We set $q$ to 7 for $n$-gram extraction. Values of $c$ and $pc$ are provided in Appendix B.1. We choose legal-longformer$_{\text{large}}$ as $\mathbf{E}_{pre}(.)$. For PMI selection we set $j$ to 50%. For optimal context selection we set $\mu$, $\sigma^2$, and $\beta$ to be 0.5, 0.7,

and 0.3 respectively. For selective masking, we set $\mu$, $\sigma^2$, and $\alpha$ to be 0.4, 0.6, and 0.4 respectively. For optimal context selection we set $\lambda$ to 0.7 and 0.5 for downstream DALE fine-tuning. We set augmentation rounds $R$ to be 5. All hyperparameters were tuned on the dev set. We also show the tuning results of some important hyperparameters in the following sub-sections.

## B.1  Discounting Factor $c$

Table 15 details the discounting factor $c$ corresponding to the percentile $pc$ for each corpus used in DALE pre-training. A corpus with documents that are entity-rich has a higher discounting factor (Caselaw) compared to a corpus with more natural language sentences and, thus, lesser entities (r/legaladvice).

Table 15 provides examples of correlated spans extracted through PMI calculation before and after discounting. Clearly, the discounting factor plays a major role in extracting spans that are reusable text fragments with fewer entities.

## B.2  Augmentation rounds $R$

Table 8 compares the performance of DALE at different values of $R$. Augmenting the training dataset with several augmentation rounds $R$ proves effective until a saturation point is reached. Downstream LLU performance improves when more DALE augmentations are added to the gold, similar to findings in Geiping et al. (2023).

| 1 | 2 | 3 | 4 | 5 | 6 | 7 |
|---|---|---|---|---|---|---|
| 53.67 | 54.58 | 55.02 | 58.94 | **59.35** | 59.31 | 59.09 |

Table 8: F1 for various settings of $R$. All values are averaged across all datasets and all low-resource settings.

## B.3  DALE without Optimal Context Selection

Table 9 compares the performance of DALE with and without optimal context selection. We show that optimal context selection plays a significant role in improving the performance of DALE.

| w/ Optimal Context | w/o Optimal Context |
|---|---|
| **59.35** | 57.46 |

Table 9: F1 with and without optimal context selection. All values are averaged across all datasets and all low-resource settings.

## C   More Results

As discussed earlier, for most of our experiments in Section 4.4, we adhere to simple encoder-only architectures. However, we hypothesize that RR on the BUILD dataset (Malik et al., 2022) and DLI, we on the ContractNLI (Koreeda and Manning, 2021) dataset might benefit from complex architecture due to the nature of their task. Thus, we compare the performance of our baseline augmentation strategies with DALE augmentations on the current state-of-the-art for RR and DLI tasks. Table 10 shows results. As clearly visible, when compared to GENIUS augmentations (also the second-best baseline in Table 4), DALE shows better margins than using a simple baseline. This proves our hypothesis that better architectures can lead to better performances with DALE for more complex tasks beyond just classification.

| #Gold | 100 | 200 | 100 | 200 |
|---|---|---|---|---|
| **Dataset** | **BUILD-RR** | | **ContractNLI** | |
| Gold-only | 76.1 | 78.3 | 75.3 | 84.2 |
| AEDA | 79.6 | 84.6 | 79.0 | 86.5 |
| Genius | 80.2 | 81.9 | 79.2 | 85.6 |
| **DALE** | **85.3** | **88.9** | **84.7** | **89.7** |

Table 10: Result comparison of DALE on BUILD-RR and ContractNLI datasets using systems proposed in Marino et al. (2023) and Ivgi et al. (2023) respectively.

## D   Comparison of Masking Algorithms

The main objective of correlated span extraction (using our modified formulation) is to mask informative and co-occurring text fragments that usually outline the emerging and case-specific facts and entities (Section 3.1 explains why this is important for the success of DALE). Using the masking process described in Section 3.2 (named importance masking hereof) does not satisfy our needs. Without the label information, the importance masking algorithm will merely retain the "most important" n-gram spans (and mask everything else), where importance is measured with respect to the context of the entire sentence. This leads to two additional problems:

1. Beyond just not explicitly masking co-occurring spans (which we iterate is important for effective learning), the importance masking algorithm often does the exact opposite and masks case-specific facts, entities, and random spans (as they

are deemed non-important by the algorithm). We show two examples below, where we compare the masking algorithms on two pre-training sentences:

1. **DALE Masking:** <mask> abuse its discretion <mask> dismissing Morgans appeal <mask> to exhaust administrative <mask>

2. **Importance Masking:** <mask> the superior court abuse its discretion <mask> to exhaust administrative <mask>

1. **DALE Masking:** <mask> payment due <mask> payment is due <mask>

2. **Importance Masking:** The Borrower shall make all payment due hereunder <mask>

As we clearly see, denoising using DALE masks exactly replicates how a legal practitioner would gain knowledge about legal concepts, principles, and language usage (powered by co-occurring and principled span masking). On the other hand, importance masking masks random spans that hurt learning. However, with label information, importance masking works well for our purpose and retains spans most informative of the instance label (important for maintaining label consistency in generations).

The quality of the spans retained also largely depends on the encoder used for similarity scoring. Additionally, our DALE pre-training masking algorithm is a principled masking algorithm asking the model to recreate and learn a similar nature of knowledge across the corpus. For importance masking, the high variability in the nature of words or phrases masked breaks this principality, thus reducing the effectiveness. In the final version of our paper, we will also include a comparison of pre-training on the two algorithms on a smaller corpus to show the effectiveness of our proposed algorithm.

2. Finally, label information is a key ingredient to importance masking and is ineffective without it. The importance masking algorithm is designed with the intuition that retaining the "most important" n-gram spans with label information will lead to augmentations that maintain label consistency. Maintaining label consistency (i.e., the augmentations should be of the same label as the source sentence) is key to any data augmentation algorithm. Without label information, the importance of each span will be measured only with the help of the document context, which will capture non-informative

spans. Also, legal documents are generally long, and different parts of the document play different roles (Malik et al., 2022) . Without a label, using just these documents for importance scoring leads to ambiguity in the selected spans for masking.

# E Comparison of Pre-trained Language Models

In this section, we first try to answer why we think *denoising* is an appropriate training objective to generate better data augmentations for the legal domain. Following this, we try to justify our choice of PLM among all open-source PLMs available.

**Why denoising?** Synthetic data augmentation can be seen as a document (or sentence) *editing* or *re-writing* task, where the primary aim is to generate diverse and coherent forms of the original document while maintaining *consistency* with the original document in terms of underlying data distribution and factuality. Generating augmentations with plausible contexts has been seen as an important measure in knowledge-intensive domains like legal and biomedical (Ghosh et al., 2023). Legal documents, by nature, are filled with domain- and case-specific facts and entities, which are, in turn, derived from the general knowledge of law. For example: An ideal augmentation, which might also help the model generalize better, should be allowed to change the context of the sentence (or the context of the facts and events occurring in the sentence), but only to the extent that it maintains plausibility and does not contradict general legal knowledge. Thus, we hypothesize that this task can be best framed as a *text infilling* task, which allows the model to re-write the document in the presence of *key hints*, thereby avoiding *hallucination*. Re-writing requires the LM to possess the knowledge of legalese and general legal knowledge, and our masking algorithm is designed to make the model acquire this knowledge.

**Why do decoder-only LLMs struggle to generate coherent and factual data augmentations in the legal domain?** Legal corpora, both unlabeled and downstream labeled, are structured at a document level as opposed to natural language, which is generally structured at a sentence level. Additionally, legal documents are generally much longer. Decoder-only LLMs suffer from **attention degeneration problem**, where, as the length of the target sequence grows, less and less attention will be focused on the source sequence (Fu et al.,

2023). This gives rise to two specific problems with both instruction-tuned and prefix-tuned LMs: **(1)** With an increase in output length, the properties in output generations deviate from the original sentence and attributes specified in the input. **(2)** The model's tendency to hallucinate increases, generating non-factual and non-plausible augmentations. We show examples in Table 18.

**Why BART?** The choice of PLM depends on the task (Tay et al., 2023). Based on denoising training and conditional generation, our algorithm better suits the encoder-decoder paradigm. Tay et al. also show that decoder-only LMs are ineffective for denoising-based training. Open source encoder-decoder models include T5 (Raffel et al., 2020), BART (Lewis et al., 2019), LongT5 (Guo et al., 2022b), Longformer Encoder-Decoder (Beltagy et al., 2020), FlanT5 (Tay et al., 2021) and Flan-UL2 (Tay et al., 2023). Though some of these models support input lengths ≥ 1024, to the best of our knowledge, the maximum decoder output length is 1024 (for BART-large), except Flan-UL2. Flan-UL2 LLM is difficult to train even on commercial GPUs, and we found BART-large, much smaller in size than Flan-UL2, to perform exceptionally well already in our case. We leave further exploration for future work.

# F Baselines

In this section, we provide details about the working of each of our baselines taken from prior art.

**EDA.** EDA (Wei and Zou, 2019) performs synonym replacement from WordNet, random insertion, random swap, and random deletion of tokens in the source sentence to generate additional synthetic augmentations. Legal text generally has semantically and syntactically complex phrases and entities, and finding matches from the WordNet leads to in-coherent augmentations.

**Legal-EDA.** Legal-EDA (Perçin et al., 2022), similar to EDA, performs replacement from WordNet but employs pre-trained Word Embeddings to calculate a similarity metric to choose the best candidates for replacement.

**GENIUS.** GENIUS (Perçin et al., 2022), similar to DALE, pre-trains and optionally fine-tunes BART on a denoising objective using sketches generated with an extreme masking algorithm. This algorithm just preserves keywords in a sentence and masks everything else. As mentioned earlier, we pre-

| Dataset | Source | Sub-domain | Task Type | Training/Dev/Test Instances | Classes |
|---|---|---|---|---|---|
| ECtHR (Task A) | Chalkidis et al. (2019) | ECHR | Multi-label classification | 9,000/1,000/1,000 | 10+1 |
| ECtHR (Task B) | Chalkidis et al. (2021b) | ECHR | Multi-label classification | 9,000/1,000/1,000 | 10+1 |
| SCOTUS | Spaeth et al. (2013) | US Law | Multi-class classification | 5,000/1,400/1,400 | 14 |
| EUR-LEX | Chalkidis et al. (2021a) | EU Law | Multi-class classification | 55,000/5,000/5,000 | 100 |
| LEDGAR | Tuggener et al. (2020) | Contracts | Multi-class classification | 60,000/10,000/10,000 | 100 |
| UNFAIR-ToS | Lippi et al. (2019) | Contracts | Multi-label classification | 5,532/2,275/1,607 | 8+1 |
| CaseHOLD | Zheng et al. (2021) | US Law | Multiple choice QA | 45,000/3,900/3,900 | n/a |
| ILDC | Malik et al. (2021) | IN Law | Multi-class classification | 32,305/994/1,517 | 2 |
| OTS-UL | Drawzeski et al. (2021) | EU Law | Multi-class classification | 2074/191/417 | 3 |
| OTS-CT | Drawzeski et al. (2021) | EU Law | Multi-class classification | 19,942/1,690/4,297 | 8+1 |
| EDGAR | Au et al. (2022) | US Law | Named Entity Recognition | 8156/1744/1740 | 7 |
| Indian-Legal-NER (Preamble) | Kalamkar et al. (2022) | IN Law | Named Entity Recognition | 1560/125/441 | 14 |
| Indian-Legal-NER (Judgment) | Kalamkar et al. (2022) | IN Law | Named Entity Recognition | 9435/949/4060 | 14 |
| ContractNLI | Koreeda and Manning (2021) | NDA | Natural Language Inference | 423/61/123 | 17 |
| BUILD | Malik et al. (2022) | IN Law | Sequential Text Classification | 247/30/30 | 13 |

Table 11: Statistics for each downstream LLU dataset used in our experiments. As described in Section 4.2, we derive low-resource splits from these original datasets for our experiments.

train GENIUS warm-starting from BART, using the extreme masking algorithm on our pre-training dataset. It proves ineffective for legal texts as legal documents are rich in entities (i.e., keywords determined by its unsupervised keyword extraction algorithm), and the algorithm generally leads the model to reconstruct case-specific facts around these entities.

**SSMBA.** SSMBA (Ng et al., 2020b) generates synthetic training examples by using a pair of corruption and reconstruction functions to move randomly on a data manifold.

**AEDA.** AEDA (Karimi et al., 2021) is similar to EDA but only employs random insertion of punctuation marks in the original text to generate synthetic augmentations. Legal text, being formal in nature, is already punctuated; thus, this proves ineffective on legal documents.

**SMERTI.** SMERTI (Feng et al., 2019) employs techniques like semantic text exchange using masked language models, keyword replacement (with keyword extraction similar to GENIUS), and adding synthetic noise using LMs. Though effective for NLP, these methods generate incoherent augmentations for formal language like legal. For example, randomly replacing tokens generally replaces tokens in a complex phrase, and keyword replacement using RAKE generally tends to edit emerging entities, both of which do not lead to efficient augmentations for the legal domain.

**BackTrans.** BackTrans or BackTranslation (Yu et al., 2018) translates a sentence into a target language and then translates it back into a source language. Machine Translation systems generally prove to be ineffective in translating formal and entity-rich language in legal documents, thus generating incomplete and incoherent augmentations.

**C-MLM.** C-MLM (Kumar et al., 2020) employs BART to replace random tokens via mask infilling in a source sentence to generate augmentations. As mentioned, we pretrain a BART using random masking on our pre-training data for this baseline. Though effective for NLP, augmentations generated by replacing random tokens do not help in legal text. Moreover, BART trained on a random masking algorithm fails to infill masks and generate coherent legal text as the random masking algorithm does not promote learning of legal language.

**ChatGPT.** Chataug (Dai et al., 2023) based on ChatGPT employs ChatGPT to rephrase existing sentences and generate more synthetic examples. The prompts are designed to generate single or multiple augmentations at a time, and we use the former. We emphasize that just rephrasing a sentence does not serve as effective augmentation for the legal domain, adding to the fact that ChatGPT starts hallucinating with rephrasing long legal documents, a common problem with decoder only LLMs (Fu et al., 2023). We show examples of ChatGPT generations in Table 18. We use the March 24 release of ChatGPT (version: 6825453).

**Falcon.** Falcon (Penedo et al., 2023), similar to ChatGPT, employs open-source instruction-tuned LLM falcon to rephrase existing sentences and generate more synthetic examples. We use a similar set of prompts, adding to an additional prompt which is: "Generate 5 different and diverse forms of the sentence:". We found Falcon to struggle in following instructions like "Rephrase the sentence:" and "Generate diverse augmentation for the sentence:". Additionally, Falcon also refuses to generate diverse forms of legal sentences at times. Falcon proves to be inferior in both rephrasing and gener-

ating diverse forms of legal documents. We show examples of generations in Table 18.

**GPT3-Mix.** GPT3-Mix (Yoo et al., 2021) prompts GPT3 (Brown et al., 2020) to generate new training samples by mixing 2 existing samples of opposite labels. This is followed by pseudo-labeling using GPT3. Mixing samples have been very often experimented in NLP for boosting diversity. However, we noticed that it leads to incoherent sentences in the case of legal language due to its formal nature.

**PromDA.** PromDA (Wang et al., 2022) proposes a data augmentation framework based on T5 that trains soft prompts using a novel keyword-to-sentence algorithm.

**MELM.** MELM (Zhou et al., 2022b), which stands for Masked Entity Language Modeling, suggests the fine-tuning of a transformer-encoder-based PLM on linearized labeled sequences through masked language modeling. In low-resource scenarios, MELM surpasses all other baselines and prior techniques on the CoNLL 2003 NER dataset across four languages, including mono-lingual, cross-lingual, and multi-lingual settings.

**DAGA.** DAGA (Ding et al., 2020), short for Data Augmentation with a Generation Approach, suggests the training of a one-layer LSTM-based recurrent neural network language model (RNNLM) by maximizing the probability of predicting the next token using linearized sentences. For sentence generation, they employ random sampling to create entirely new sentences, with the model being fed only the [[**BOS**]] token.

**MulDA.** The Multilingual Data Augmentation Framework (MulDA) (Liu et al., 2021), an extension of DAGA, enhances generation-based multilingual data augmentation by training a pre-trained mBART model on next token prediction using linearized sentences. To ensure a fair comparison, we substitute mBART with mBART-50 in the MulDA approach.

**LwTR.** LwTR (Dai and Adel, 2020) replaces a token in a sentence with another token of the same label; the token is randomly selected from the training set.

**FlipDA.** We do not consider this baseline. FlipDA (Zhou et al., 2022a) trains a generative model to generate label-flipped data. Our initial experimentation revealed that label-flipping generated highly in-coherent augmentations for the legal domain. Thus, we conclude that label-flipping to be non-trivial for legal language compared to natural lan-

guage.

**Style-Transfer.** We do not consider this baseline. Style-Transfer (Chen et al., 2022) generates augmentations by changing style-related attributes. Our initial experimentation revealed that style-transfer generated highly in-coherent augmentations for the legal domain. Thus, we conclude that style-transfer to be non-trivial for legal language compared to natural language.

## G  Examples of generated augmentations

We provide additional augmentation examples in Table 18. Each augmentation was marked by a law student on 3 parameters: (1) If the augmentation is coherent, (2) If it adds new plausible context, and (3) if it is label-consistent and matches the underlying data distribution. We present the results of the study as ✓or ✗next to each augmentation in the same order as above.

## H  Dataset Details

### H.1  Pre-training Dataset Details.

For pre-training DALE, we use the Pile of Law dataset (Henderson et al., 2022). The dataset is a collection of multiple (unlabeled) legal corpora (Huang et al., 2021; Borchmann et al., 2020; Blair-Stanek et al., 2020; Hendrycks et al., 2021; Koehn et al., 2005; Lippi et al., 2018; Ruggeri et al., 2021) with ≈256 GB of text. Detailed statistics for each dataset can be found in Table 14

### H.2  Fine-tuning Dataset Details.

In this section, we list a detailed description of each of our downstream LLU datasets and dataset

| Data Source | Data Size | Word Count | Document Count |
|---|---|---|---|
| U.S. Board of Veterans' Appeals Decisions | 13.21GB | 1.74B | 630K |
| U.S. Supreme Court Oral Argument Transcripts | 1.51GB | 151.05M | 47K |
| Edgar Contracts (Borchmann et al., 2020) | 10.76GB | 1.44B | 741K |
| Reddit r/legaladvice & r/legaladviceofftopic | 299.04MB | 40.42M | 110K |
| **Total** | ~ 26GB | ~ 3.4B | ~ 1.5M |

Table 12: Statistics of various legal corpora in Pile of Law considered for building our pre-training dataset.

| Data Source | Data Size | Word Count | Document Count |
|---|---|---|---|
| Caselaw | ~22GB | ~4.57B | ~2.54M |
| Total | ~ 22GB | ~ 4.6B | ~ 2.5M |

Table 13: Statistics of Caselaw legal corpus.

| Data Source | Data Size | Word Count | Document Count |
|---|---|---|---|
| MAUD | 124.5MB | 21.8M | 39.2K |
| Total | ~ 125MB | ~ 22M | ~ 39K |

Table 14: Statistics of MAUD legal corpus.

| Data Source | Discounting factor | Cut-Off Value |
|---|---|---|
| MAUD | 75% | 57, 43, 38, 36, 34 |
| Reddit r/legaladvice & r/legaladviceofftopic | 75% | 6, 3, 2, 1, 1 |
| U.S. Board of Veterans' Appeals Decisions | 95% | 20, 10, 6, 5, 4 |
| U.S. Supreme Court Oral Argument Transcripts | 95% | 27, 19, 12, 8, 5 |
| Edgar Contracts | 95% | 13, 9, 7, 6, 5 |
| Caselaw | 95% | 10, 5, 3, 3, 2 |

Table 15: Discounting values for different datasets used in DALE Pre-training. Cut-Off values for each value of $n$ (in the order of 3,4,5,6 and 7) for the $n$-grams considered in our experiments.

statistics for each.

### H.2.1 Multi-class Classification

**SCOTUS.** The US Supreme Court (SCOTUS) serves as the highest federal court in the United States of America, primarily handling highly contentious or intricately complex cases that have not been adequately resolved by lower courts. We utilized the SCDB (Supreme Court Database) (Spaeth et al., 2013), in a setting similar to (Chalkidis et al., 2021c), to classify court opinions across 14 distinct issue areas. These issue areas encompass a range of subjects, such as Criminal Procedure, Civil Rights, Economic Activity, and more. Our classification task is a single-label multi-class classification. The 14 issue areas effectively group together 278 specific issues, all centered around the subject matter of the disputes being presented before the court. Dataset statistics are provided in Table 11.

**LEDGAR.** (Tuggener et al., 2020) introduced a dataset called LEDGAR (Labeled EDGAR) specifically designed for contract provision classification at the paragraph level. The contract provisions within this dataset are sourced from contracts obtained from the US Securities and Exchange Commission (SEC) filings, which are publicly accessible through the EDGAR (Electronic Data Gathering, Analysis, and Retrieval system) platform. The dataset setting used in our paper is similar to (Chalkidis et al., 2021c). Dataset statistics are provided in Table 11.

**ILDC.** ILDC (Malik et al., 2021), a substantial corpus comprising 35,000 Indian Supreme Court cases, stands out as it includes annotations of original court decisions. Within this corpus, a specific portion has been annotated by legal experts, providing gold-standard explanations. Building upon ILDC, we introduce the Court Judgment Prediction and Explanation (CJPE) task. The model is tasked with predicting and providing comprehensi-

ble justifications for the outcome of a case. Dataset statistics are provided in Table 11.

**OTS-UL.** Online Terms of Service (OTS) (Drawzeski et al., 2021) attempt to automatically detect unfair clauses in Terms of Service. The input to the model is a sentence, and the output presents the sentence classified into three levels of unfairness. The dataset setting used in our paper is similar to (Niklaus et al., 2023). Dataset statistics are provided in Table 11.

### H.2.2 Multi-label Classification

**ECtHR Tasks A & B.** Allegations are brought before the European Court of Human Rights (ECtHR) regarding the violation of human rights provisions outlined in the European Convention of Human Rights (ECHR) by a state. We use the datasets from (Chalkidis et al., 2019) and (Chalkidis et al., 2021b). In Task A, the model takes the factual paragraphs of a case as input and predicts the set of violated ECHR articles. Task B focuses on the same aspect, where the input remains the list of factual paragraphs, but the model predicts the set of allegedly violated ECHR articles. The dataset setting used in our paper is similar to (Chalkidis et al., 2021c). Dataset statistics are provided in Table 11.

**EURLEX.** The EUR-Lex portal is the platform for publishing legislation about the European Union (EU). These laws are extensively annotated by the EU's Publications Office, incorporating multiple concepts sourced from EuroVoc. EuroVoc is a multilingual thesaurus actively maintained by the Publications Office, comprising over 7,000 concepts that cover a wide range of activities undertaken by the EU and its Member States, such as economics, healthcare, and trade. For our research, we utilize the English portion of the dataset provided by (Chalkidis et al., 2021a). This dataset comprises 65,000 EU laws (documents) sourced from EUR-Lex, allowing us to explore and analyze legislative content within the EU context. Given a document, the task is to predict its EuroVoc labels (concepts). The dataset setting used in our paper is similar to (Chalkidis et al., 2021c). Dataset statistics are provided in Table 11.

**UNFAIR-ToS.** The dataset known as UNFAIR-ToS, developed by (Lippi et al., 2019), encompasses 50 Terms of Service (ToS) documents extracted from various online platforms such as YouTube, eBay, Facebook, and others. These ToS

| Dataset | Vanilla PMI | 75$^{th}$ $pc$ | 95$^{th}$ $pc$ |
|---|---|---|---|
| **MAUD** | November 3 , 2020 United States federal
SARS - CoV - 2 virus
Rule 14e - 1c under
2020 United States
body empowered or appointed thereby | transportation delays including work stoppages or port
delays including work stoppages or port closures
return tendered shared promptly
use commercially reasonable efforts
independent exploration and production companies primarily | more orders that impose a clinical hold
return tendered Shares promptly
unanimously adopted resolutions
use reasonable best efforts
generally accepted accounting |
| **Reddit r/legaladvice & r/legaladviceofftopic** | Island of Puerto Rico
Planned Parenthood Federation of America
City of Hong Kong
Beep Boop
Jiffy Lube: Car Maintenance | one count of first degree homicide
eviction from the rented house
custody and divorce agreements
unlawfully destroy public property belonging
obstruction of the legal process | consideration of the sum of 5200.00
gross disfigurement and asymmetry
nationally recognized reputation
meeting duly called
other hazardous environmental |
| **U.S. Board of Veterans' Appeals Decisions** | 2003R S 4597b
Rhabdomyoblastic Differentiation Malignant Triton
Liposarcoma Leiomyosarcoma Epithelioid Leiomyosarcoma
Centralized Accounts Receivable Online
World Dictionary of American English | courts have imposed a requirement
reference to the diagnostic criteria
eligible persons who served
Baton Rouge, Louisiana
Department of Veterans Affairs | adequate responses to the specific opinions requested
motion for review for clear and unmistakable
respond to the following inquiries
appearance at oral argument
statement that the claims folder |
| **U.S. Supreme Court Oral Argument Transcripts** | Frankie Sue Del Papa
Neth L. Leachman
Racketeer Influenced and Corrupt Organizations

Blanca Bianchi De La Torre
Goose Foods and Sunshine Biscuits | impair binding contracts or debts
repealing certain constitutional provisions to conform
prohibiting certain persons from serving as active

Tulare Lake Basin Water Storage
Fountain Packing Company versus Haydel | reviling or using obscene or opprobrious
convicted of certain crimes
refuse to submit to arbitration after agreeing

possess to have the dispute litigated
judgmentor when it indisputably |
| **Edgar Contracts** | MEDBOX INC : MDBX
FINEGOLD Daniel W. Finegold
Kinsella Assistant Treasurer None Brett Scribner
Krispy Kreme Company
New York Agreement Amendment | request including exchanges from other vanguard funds
remit subsequent payments and forward communications any
track', rather than outperform
Eilleen M. Clavere
Janette E. Farragher | rates which can fluctuate significantly over short
laws it is not intended as tax
significant accounting policies
imply that the commission has verified
superseded by documents or reports subsequently |
| **Caselaw** | House Of Representatives
Parchment Co. v. Paterson Parchment
Reneau P. Almon Janie
Blue Cross and Blue Shield
caution it is important that you thoroughly | entry of a judgment not inconsistent
voluntarily and knowingly waive any right they
entered in any court
LUCILLE A. ROPER
Uniformed Services Former Spouses | advisory opinion of the justices
entered in any court having jurisdiction
requesting an advisory opinion of the justices
interstate commerce
either pursuant to arbitration |

Table 16: Comparison of correlated spans extracted from **Vanilla PMI** and discounting factor $c$ applied at 75$^{th}$ and 95$^{th}$ percentile ($pc$). As we clearly see, the spans extracted improve gradually with increasing $pc$. A higher $pc$ allows us to extract reusable fragments from entity-rich legal documents.

documents have undergone sentence-level annotation to identify eight distinct categories of unfair contractual terms. These categories represent sentences within the ToS that potentially infringe upon user rights, as per the guidelines outlined in EU consumer law. The model takes a sentence as input and generates the set of unfair categories, if applicable, associated with that particular sentence. The aim is to detect and classify instances of unfair contractual terms present in online platform ToS documents. The dataset setting used in our paper is similar to (Chalkidis et al., 2021c). Dataset statistics are provided in Table 11.

**OTS-CT.** Online Terms of Service (OTS) (Drawzeski et al., 2021) attempt to automatically detect unfair clauses in Terms of Service. The input to the model is a sentence, and the model identifies the sentence for various clause topics. The dataset setting used in our paper is similar to (Niklaus et al., 2023). Dataset statistics are provided in Table 11.

### H.2.3 Named Entity Recognition

**EDGAR.** EDGAR (Au et al., 2022) is based on legal company filings available from the US Securities and Exchange Commission's EDGAR data set. EDGAR is annotated with 7 named entity classes, namely Location, Person, Business, Government, Court, Legislation/Act, and Miscellaneous. Dataset statistics are provided in Table 11.

**Indian-Legal-NER.** Indian-Legal-NER (Kalamkar et al., 2022) is derived from Indian Court Judgments and consists of two separate sub-datasets, namely the judgment and the preamble. The preamble of a judgment contains formatted metadata like names of parties, judges, lawyers, date, court, etc. The text following the preamble till the end of the judgment is called "judgment." The dataset is annotated with 14 named entities, namely, COURT, PETITIONER, RESPONDENT, JUDGE, LAWYER, DATE, ORG, TYPE, GPE, STATUTE, PROVISION, PRECEDENT, CASE-NUMBER, WITNESS and OTHER-PERSON. Dataset statistics are provided in Table 11. All results in the main paper are averaged for judgment and preamble.

### H.2.4 Other Tasks

**ContractNLI.** The ContractNLI dataset (Koreeda and Manning, 2021) has been developed specifically for document-level natural language inference (NLI) tasks focused on contracts. This dataset aims to automate and facilitate the labor-intensive process of contract review. In this task, a system is provided with a set of hypotheses, such as "Some obligations of Agreement may survive termination," along with a contract. The system's role is to classify whether the contract entails each hypothesis, contradicts the contract, or is not mentioned in the contract (neutral). Additionally, the system is expected to identify the specific evidence that supports its decision in the form of spans within the contract. Dataset statistics are provided in Table 11.

**BUILD.** BUILD (Malik et al., 2022) is a dataset built for Rhetorical Role (RR) Prediction - given a

document, the task is to predict the text segments corresponding to various roles. The task can be seen as a sequential text classification task (Qian et al., 2020). The dataset is labeled with 13 fine-grained RRs: Fact, Argument, Statute, Dissent, Precedent, Ruling By Lower Court, Ratio Of The Decision, Ruling By Present Court, and None.

**CaseHOLD.** The CaseHOLD (Case Holdings on Legal Decisions) dataset (Zheng et al., 2021) contains multiple choice questions about holdings of US court cases from the Harvard Law Library case law corpus. Holdings are short summaries of legal rulings accompanying referenced decisions relevant to the present case. The input consists of an excerpt (or prompt) from a court decision that references a particular case, where the holding statement (in boldface) is masked. The model must identify the correct (masked) holding statement from five choices.

**Indian- and UK-Abstractive datasets.** Indian-Abstractive and UK-Abstractive datasets (Shukla et al., 2022), are datasets built for abstractive summarization, were collected from Indian Supreme Court judgments from the website of the Legal Information Institute of India [2] and The UK Supreme Court website [3] respectively. The dataset setting used in our paper is similar to (Shukla et al., 2022). Dataset statistics are provided in Table 11.

# I  Additional Details

## I.1  $L_{D_f}$ for fine-tuning

**Classification.** For multi-class classification, we take $L_{D_f}$ as the gold annotated label of the document. For multi-label classification, we concatenate the label strings for all the gold annotated labels of the document.

**NER.** For NER, we take $L_{D_f}$ as the template "*Entity-1* is a *label-1* [SEP] $\cdots$ [SEP] *Entity-n* is a *label-n* " where *Entity-i* corresponds to the $i^{th}$ named entity in the sentence and *label-i* corresponds to the gold annotated label of the named entity.

**MCQ.** For MCQ, we take $L_{D_f}$ as the actual godl annotated answer of the question.

**RR.** For rhetorical role prediction, we take $L_{D_f}$ as the rhetorical role of the sentence in the document (we generate augmentations sentence-wise).

**DLI.** For DLI, we take $L_{D_f}$ as the gold annotated hypothesis of the document.

## I.2  Other Details

**Model Parameters:** legal-longformer$_{large}$ has $\approx$ 409M parameters with 24-layers of encoder, 1027-hidden-state, 4096 feed-forward hidden-state and 16-heads. BART$_{large}$ $\approx$ has 680M parameters with 12 layers of encoder, 12 layers of decoder, 1024-hidden-state, and 16-heads.

**Compute Infrastructure:** All our experiments are conducted on a single NVIDIA A100 GPU. An entire DALE fine-tuning pipeline takes $\approx$ 40 minutes. We pre-trained DALE for 7 days on 4 NVIDIA A100 GPUs.

**Implementation Software and Packages:** We implement all our models in PyTorch [4] and use the HuggingFace [5] implementations of BART$_{large}$ and legal-longformer$_{large}$ [6]. For multi-class classification and multi-label classification, we use the HuggingFace Trainer implementations of the corresponding tasks. For NER, we use the FLAIR toolkit (Akbik et al., 2019) to fine-tune all our NER models. For CaseHOLD MCQ, we follow the setup proposed by (Zheng et al., 2021) [7]. For ContractNLI DLI, we follow the setup proposed by (Koreeda and Manning, 2021)[8]. For BUILD RR, we follow the setup proposed by (Malik et al., 2022) [9]. For CaseHold, ContractNLI and BUILD we replace the original encoder with legal-longformer$_{large}$.

**Potential Risks:** Conditional Language Models used for Natural Language Generation often tend to *hallucinate* (Ji et al., 2022) and potentially generate nonsensical, unfaithful or harmful sentences to the provided source input that it is conditioned on.

---

[2] http://www.liiofindia.org/in/ cases/cen/INSC/

[3] https://www. supremecourt.uk/decided-cases/

[4] https://pytorch.org/

[5] https://huggingface.co/

[6] https://huggingface.co/lexlms/legal-longformer-large

[7] https://github.com/reglab/casehold

[8] https://github.com/stanfordnlp/contract-nli-bert

[9] https://github.com/Legal-NLP-EkStep/rhetorical-role-baseline

| Document 1 |
| --- |

The case was tried before a jury, which returned a verdict in favor of the plaintiff.
This action was based on claims of fraud and breach of contract under a credit life insurance policy. ⋯
In the second count, she claims breach of contract because Magic City Dodge and Peninsular Life Insurance Company refused to pay the benefits of the credit life policy. ⋯
Dennis died with a balance remaining on his obligation. ⋯

| Document 2 |
| --- |

The complaint alleges that the realtor, Barnett, was elected and duly qualified as marshal of the city of Nobles. ⋯
Prayer for a writ of mandate to restore the relator to his office as marshal
The question of the power of the common council to remove the relator is properly presented by his application for a writ of mandate. ⋯
Errors are assigned upon these decisions. ⋯

| Document 3 |
| --- |

The existence of a labor dispute is not questioned here , and no claim is made for compensation during the time the dispute was in progress. ⋯
Appellant filed claim for unemployment compensation. ⋯
On June 30th , without reporting back to the company , claimant registered with the Alabama Unemployment Service and made application for unemployment compensation. ⋯
Pier claim was allowed by the claims examiner of the Department of Industrial Relations. ⋯

| Document 4 |
| --- |

In this case , the former wife presented undisputed evidence as to the financial and other circumstances of the parties existing at the time of the entry of the 1990 judgment. ⋯
Mary Jo Blount the former wife filed a petition in the Madison Circuit Court the trial court , seeking a modification of the provisions of a 1990 divorce that incorporated a settlement agreement in which the former husband agreed to pay the former wife 500 per month in periodic alimony. ⋯
The former wife timely appealed to this court and argues that the trial court findings are clearly erroneous. ⋯
In 2012 , she reported 28,951 in gross income. ⋯

| Document 5 |
| --- |

validity of a statute can never depend upon the antecedent consultation of the people by the legislature , nor upon the affording to them an opportunity to express their sentiments through petitions. ⋯
The removal of the court - house of a county , and its permanent location , is indisputably a permissible exercise of legislative authority. ⋯
If the question of the power of the legislature , to make the removal of the court - house to Selma dependent upon the condition of its approval by a popular vote , were res integra , there would be room for much argument. ⋯
Mr. Justice Byrd not sitting in the case. ⋯

Table 17: Examples of legal documents from our pre-training corpus with correlated spans. Spans highlighted in green co-occur within the same document, while spans in yellow co-occur across documents.

Table 18: Comparison of augmentations generated by DALE and all other baselines. All augmentations were generated in a low-resource setting (500). Each augmentation was marked by a law student on 3 parameters: (1) If the augmentation is coherent, (2) If it adds new plausible context, and (3) if it is label-consistent and matches the underlying data distribution. We present the results of the study as ✓or ✗ next to each augmentation in the same order as above.

| **BUILD-RR** |
| --- |

| | |
|---|---|
| Original | In the decision reported in 2003(7) SCC 141, supra, the Apex Court has held that "It is true that the incident in question has prematurely terminated the life of twenty-one people and the number of deaths cannot be the sole criterion for awarding the maximum punishment of death." In the decision reported in (2011)10 SCC 389, supra, Apex Court has held that the appellant/Accused would not be a menace to society and no reason to believe that the appellant cannot be reformed or rehabilitated or would constitute a continued threat to society and it is not the 'rarest of the rare case' causing for extreme penalty of death. |
| **EDA** | **(1)** In the decision reported in 2003(7) SCC 141, supra, the Apex Court has held that "It is true that the incident in question has prematurely terminated the life of twenty-one people and the number of deaths cannot be the sole criterion for awarding the maximum punishment of death. reformed the decision continued in would supra apex court has held that the appellant causing scc not be a menace to society and no reason to believe that the appellant cannot be in or rehabilitated or constitute would a reported threat to society and it is not the rarest of the rare case accused for extreme penalty of death ✗ ✗ ✓ |
| | **(2)** in the decision reported in scc supra the apex has held that it is true that incident question prematurely terminated the life of twenty one people and the of deaths cannot be the sole criterion for awarding the maximum punishment of deathin the decision reported in scc supra apex court has held that the appellant accused would not be menace to society and no reason to believe that the appellant cannot be reformed or rehabilitated or constitute a to society and it is not the of the rarecase causing for extreme penalty of death ✗ ✗ ✓ |
| **SSMBA** | **(1)** under the decision reported in 2003 ( 7 ) scra 1411 supra ) the apex court has noted stated 'it is true that the incident in question actually prematurely terminated from only about twenty - five people and the number of deaths cannot be the correct cause for awarding this maximum punishment of death. in the cases reported in ( 2011 ] at scc 383, supra, apex court has decided that the appellant a accused would not be a risk of society and no reason to believe that the appellant cannot have reformed or utilified or would constitute a real threat to society and it is not the ' rarest of the rare cases'causing for death penalty of death. ✓ ✗ ✓ |
| | **(2)** in his decision reported in 2003 ( 7 ) scs 1419 supra, the trial court has held that " it is understood that the incident without question whichly terminated the life of twenty - one people. the number of deaths cannot be the sole criterion for awarding a maximum punishment on death. "in his decision reported in ( 2011 ) 10 scc 389, 2 43a, j he has held that the appellant / accused would not present a menace within society and no reason to believe that the appellant would be reformed or mobilitated or would constitute the non danger to society and it is not the ' rarest of the rare casećausing any further lack of victims. ✓ ✗ ✓ |

| | |
|---|---|
| **AEDA** | **(1)** In the decision reported ; in 2003(7) SCC ; 141, supra, : the Apex Court has held that "It ? is ! true that the incident in ; question has prematurely terminated the life of twenty-one people and the . number of deaths cannot . be ! the sole : criterion ! for awarding the ; maximum punishment of death. In . the . decision reported in (2011)10 SCC 389, supra, Apex Court has held that the ! appellant/Accused would not be a menace to society and ! no reason to believe that the appellant cannot be reformed or rehabilitated ? or would constitute a continued . threat to society and it is not the : 'rarest ? of the ? rare case' causing for extreme penalty of death. ✗ ✗ ✓ |
| | **(2)** In ! the ; decision reported in 2003(7) SCC 141, supra, the Apex Court has held . that "It is true that the ? incident in question has prematurely terminated the life of twenty-one people and the number of deaths cannot be the sole criterion for awarding the maximum punishment of death. In the decision reported in (2011)10 ; SCC 389, , supra, Apex ? Court has , held that the appellant/Accused would not be ! a menace ; to society and no reason : to : believe that the appellant cannot ; be reformed or rehabilitated : or would constitute a , continued threat to society and it ! is not the 'rarest of the rare ? case' causing for ? extreme ! penalty of death. ✗ ✗ ✓ |
| **BackTrans** | **(1)** In the decision SCC 141 published in 2003(7), the Apex Court stated: "It is true that the incident in question has prematurely ended the lives of 21 people and that the number of deaths cannot. In the decision cited above in (2011)10 SCC 389, the Apex Court ruled that the complainant/applicant presents no danger to the company and that there is no reason to believe that the ✓ ✗ ✓ |
| | **(2)** In the decision SCC 141 notified in 2003(7), the Apex Court found that "it is true that the incident in question has prematurely ended the lives of 21 people and that the number of deaths is not. In the decision SCC 389 published above in (2011)10, the Apex Court found that the complainant/applicant did not present any danger to the company and ✓ ✗ ✓ |
| **GENIUS** | **(1)** The decision reported in the Apex Court has indicated that the government in question has prematurely terminated the right to cause death and that the cause of deaths can be ascertained. The court also said that there is no need for the government to intervene in the matter. I believe that the appellant/Accused would not be able to have a fair trial. I think that the appellants can be either injured, or rehabilitated or would have to be tried in a very rare case. ✗ ✗ ✗ |
| | **(2)** The decision reported in the Apex Court has said that the court in question has prematurely terminated the right of deaths can be done. The court has also ordered that the government should be allowed to use the power of the courts to ensure that the citizens of the state are not deprived of the right to life. I believe that the appellant/Accused would not be able to do so, and I believe it would be difficult for him to do it. I also believe, however, that the appellants can be tried or rehabilitated or would be in a very rare case. ✗ ✗ ✗ |

| | |
|---|---|
| **ChatGPT** | **(1)** The Supreme Court, in its 2003(7) SCC 141 ruling, acknowledged that the incident in question tragically claimed the lives of twenty-one individuals. However, the court emphasized that the number of deaths alone should not be the sole basis for imposing the maximum penalty of death. In the 2011(10) SCC 389 decision, the Apex Court held that the appellant/accused did not pose a threat to society and there was no reason to believe that they could not be reformed or rehabilitated. Consequently, the court deemed it not to be a 'rarest of the rare case' warranting the extreme punishment of death. ✓ ✗ ✓ |
| | **(2)** According to the 2003(7) SCC 141 ruling, the Supreme Court acknowledged the unfortunate event that resulted in the premature loss of twenty-one lives. However, the court emphasized that the number of deaths alone should not be the sole criterion for imposing the maximum punishment of death. As per the (2011)10 SCC 389 case, the Apex Court determined that the appellant/accused posed no threat to society and showed no indications of being beyond reform or rehabilitation. The court concluded that this particular case did not meet the criteria of being an extremely rare instance warranting the death penalty. ✓ ✗ ✓ |
| **Falcon** | **(1)** In the decision reported in 2003(7) SCC 141, supra, the Apex Court has held that "It is true that the incident in question has prematurely terminated the life of twenty-one people and the number of deaths cannot be the sole criterion for awarding the maximum punishment of death." In the decision reported in (2011)10 SCC 389, supra, Apex Court has held that the appellant/Accused would not be a menace to society and no reason to believe that the appellant cannot be reformed or rehabilitated or would constitute a continued threat to society and it is not the 'rarest of the rare case' causing for extreme penalty of death. ✓ ✗ ✓ |
| | **(2)** In the decision reported in 2003(7) SCC 141, supra, the Apex Court has held that "It is true that the incident in question has prematurely terminated the life of twenty-one people and the number of deaths cannot be the sole criterion for awarding the maximum punishment of death." In the decision reported in (2011)10 SCC 389, supra, Apex Court has held that the appellant/Accused would not be a menace to society and no reason to believe that the appellant cannot be reformed or rehabilitated or would constitute a continued threat to society and it is not the 'rarest of the rare case' causing for extreme penalty of death. ✓ ✗ ✓ |
| **DALE-pt** | **(1)** The decision reported in 6 N. J. Apex Court has determined that the insurance policy in question has prematurely terminated the period within which proof of deaths can be made .The Court finds that the appellant/Accused would not be entitled to relief on habeas corpus because the evidence does not show that the the appellant can not be adequately supervised or rehabilitated or would not suffer irreparable harm except in a rare case. ✓ ✗ ✓ |
| | **(2)** The decision reported in the Apex Court has been modified to reflect that the statute in question has prematurely terminated the period during which claims for benefits on account of deaths can be made. The Court finds that the appellant/Accused would not be entitled to relief unless it is shown that there is a reasonable likelihood that the the appellant can be cured or rehabilitated or would suffer irreparable harm except in a rare case. ✓ ✗ ✓ |

| | |
|---|---|
| **DALE-ft** | **(1)** In the decision reported in (2015)4 SCC 709, supra, the Apex Court has held that the death of the woman in question has prematurely terminated the life of the deceased and therefore the number of deaths cannot be taken into consideration as a basis for awarding maximum punishment. It is submitted that the appellant/Accused would not be a menace to society and, therefore, it is to be held (1) [1960] 1 S.C.R. 287.n(2) [1952] 1 K.L.J. 438. that the appellant cannot be reformed or rehabilitated or would attract the punishment of death or imprisonment for life in the rare case of his release on bail. ✓ ✗ ✓ |
| | **(2)** In the decision reported in 2003(2) SCC (Cri.) 941, supra, the Apex Court has held that when the act in question has prematurely terminated the life of a person, the number of deaths cannot be taken as a basis for awarding death sentence.Therefore, it cannot be said that the appellant/Accused would not be a menace to society and society as a whole, and I am of the considered view that there is no reason to believe that the convict cannot be reformed or rehabilitated or would be a threat to society in the rare case. ✓ ✗ ✓ |
| **DALE** | **(1)** In the decision reported in 2003(2) SCC 16, the Apex Court has held as follows : "It is true that the incident in question has prematurely terminated the life of Sangita and the number of deaths cannot be the sole criterion for awarding death sentence. In the light of the above discussion, I am of the considered view that the appellant/Accused would not be a menace to society as a whole and there is no reason to believe that the benefit of doubt remains to be given to the appellant; and, further, I do not find that the case falls under the category of rarest of the rare cases for imposition of death sentence. ✓ ✓ ✓ |
| | **(2)** In the decision reported in 2012(3) SCC (Cri.) 908, the Apex Court has held that where the incident in question has prematurely terminated the life of a particular victim and the number of deaths cannot be the sole criterion for awarding the maximum punishment of death. It is further contended that the appellant/Accused would not be a menace to society and society as a whole and that the chances of reformation and rehabilitation of the appellant cannot be considered in the light of the report of the Institute of Forensic Laboratory which has been held by the Hon'ble Supreme Court in the case of State of Himachal Pradesh vs. Raghubir Singh (1999) 6 SCC 695, that the sentence of imprisonment for life cannot be altered or rehabilitated or would constitute a more than 'rarest of the rare case' for release on probation of good conduct. ✓ ✓ ✓ |

**ContractNLI**

| | |
|---|---|
| Original | Confidential Information' shall mean: in respect of Information provided in documentary or by way of a model or in other tangible form, Information which at the time of provision is marked or otherwise designated to show expressly or by necessary implication that it is imparted in confidence; and in respect of Information that is imparted orally, any information that the Disclosing Party or its representatives informed the Receiving Party at the time of disclosure was imparted in confidence; and in respect of Confidential Information imparted orally, any note or record of the disclosure and any evaluation materials prepared by the Receiving Party that incorporate any Confidential Information; and any copy of any of the foregoing; and the fact that discussions are taking place between the Receiving Party and the Disclosing Party. 'Disclosing Party' shall mean the party to this Agreement that discloses Information, directly or indirectly to the Receiving Party under or in anticipation of this Agreement. |

**EDA**  **(1)** confidential information shall mean imparted respect of information provided in confidence or by way designated a model or in other tangible form information which any the the of provision is marked or the of to information expressly or by necessary implication that it is imparted in confidential and in respect of information that is imparted orally any information that time disclosing party or its representatives informed the receiving party at the time of in was imparted disclosure confidence and in respect of note information the orally any party or the of the disclosure and any evaluation materials prepared by the receiving confidential that incorporate any documentary show and at copy of any of record foregoing and in fact that discussions are taking place between the receiving party and otherwise disclosing party discloses party shall mean the party to this agreement that disclosing information agreement or indirectly to the receiving party under or in anticipation of this directly. ✗ ✗ ✓

**(2)** confidential information shall bastardly in respect of information offer in objective or by way of a good example or in other tangible cast information which at the time of supply is marked or otherwise delegate to testify expressly or by necessary implication that it is imparted in confidence and in respect of information that is imparted orally any information that the disclosing party or its representatives informed the receiving party at the time of disclosure was imparted in confidence and in respect of confidential information imparted orally any note or commemorate of the disclosure and any evaluation fabric prepared by the receiving party that contain any confidential information and any simulate of any of the foregoing and the fact that discussions are take on place between the receiving party and the disclosing party disclosing party shall mean the party to this agreement that discloses information directly or indirectly to the receiving party under or in anticipation of this agreement ✗ ✗ ✓

**SSMBA**  **(1)** ' of information ' shall mean : in respect of information clearly in documentary or by means of other model or in other tangible form, information which at the time of provision is marked or otherwise designated will show expressly any the necessary implication that it is imparted in confidence ; and in respect of information that was impiculated verbally, any declaration that the disclosing party between its representatives and any receiving party at aation of disclosure was imparted in confidence ; and in respect of confidential information imparted orally, any note or record of the disclosure and any such materials prepared by its receiving country that incorporate any confidential information ; or any copy and any record such foregoing ; and the same or, are that in between either receiving party of the disclosing party "the disclosing party ' shall mean either party to this agreement that discloses information, directly or indirectly through the receiving party under or in anticipation of this agreement...' ✓ ✗ ✓

**(2)** Confidential information' shall mean: in respect of information clearly in written or by means of a model or in other tangible form, information which at the time of provision is marked or otherwise designated will show expressly any the necessary implication that it is imparted in confidence; and in respect of information that was orally imparted, any procedures that the disclosing party between its representatives and any receiving party at time of disclosure was imparted in confidence; and in respect of confidential information imparted orally, any note or record of the disclosure and any such provisions prepared by its receiving country that incorporate any confidential information; or any copy and any record such foregoing; and the same or, are that in between either receiving party of the disclosing party "the disclosing party' shall mean either party to this agreement that discloses information, directly or indirectly through the receiving party under or in anticipation of this agreement' ✓ ✗ ✓

**AEDA** **(1)** 'Confidential Information' shall mean: in respect of Information provided in documentary or by way of a model or , in other : tangible form, Information which at the time of provision is marked or otherwise designated to show expressly or by necessary ! implication that it is imparted in ! confidence; and in respect of Information that is imparted ? orally, any information that the Disclosing Party or its representatives informed the Receiving Party at , the time of disclosure was imparted in confidence; , and in respect : of Confidential Information imparted ; orally, any note or : record of the disclosure ? and any evaluation materials prepared by the Receiving Party that incorporate any Confidential Information; and any copy of any of the foregoing; ! and the : fact , that discussions are taking place between the Receiving Party and , the , Disclosing Party. 'Disclosing Party' shall mean the party to this Agreement that discloses Information, directly or indirectly to ? the Receiving Party under or in anticipation of this Agreement. ✓ ✗ ✓

**(2)** 'Confidential Information' shall mean: in respect of Information provided in documentary or by way of a model or in other ; tangible form, Information which at the time of provision ? is marked or otherwise designated to show expressly or ? by necessary implication that ; it is imparted in confidence; and in respect of Information that is imparted ? orally, ; any information that the Disclosing Party or its representatives informed the Receiving Party at the time of disclosure was imparted in confidence; and in respect of Confidential Information imparted orally, any note or record of the disclosure and any evaluation materials prepared by the Receiving Party that incorporate any Confidential Information; and any copy of any of the foregoing; and the fact that discussions are taking place between the Receiving Party and the Disclosing Party.. 'Disclosing Party' shall mean the party to this Agreement that discloses Information, directly or : indirectly . to the Receiving Party under or ! in anticipation of this Agreement. ! ') ✓ ✗ ✓

**BackTrans** **(1)** 'confidential information' means information made available at the time of provision, express or through the necessary implication that it is confidential, or in the form of a model or other tangible form; information marked or otherwise intended at the time of provision, express or'The contracting party to this Agreement shall be the party to this Agreement which, under or in anticipation of this" ✓ ✗ ✓

**(2)** 'confidential information' means information that, at the time of making it available, is expressly or through the necessary conclusion that it is confidential or information that is identified at the time of making it available or is otherwise intended to be displayed in order to express or by the necessary conclusion'Disclosing Party' means the Party to this Agreement which discloses information directly or indirectly to the receiving Party, under or in anticipation of this Agreement. ✓ ✗ ✓

**GENIUS** **(1)** Confidential Information. The disclosure of Information provided to you by the Information which is imparted in confidence to you. 2. The Information that is exparted orally. 3. The information that the disclosure was imparted to you in confidence. 4. The respect of Confidential Information imparted orally to you and to the disclosure and any other person. 5. The confidentiality of the information. 6. The content of the content. 7. The manner in which the information is communicated to you or to any other persons. 8. The way in which you incorporate any Confidential information. The Disclosing Party is an Agreement that discloses Information to the Receiving Party under the terms of the Agreement. The Agreement may be signed by the parties to this Agreement. ✓ ✗ ✗

**(2)** Confidential Information i.e. the whole of Information provided. ii. Information which is imparted in confidence to the person and/or the person or the person in the possession of the information that is being disclosed. iii. information that the person, or the persons or persons in the persons responsible for the preparation of the disclosure. iv. information which is the subject of disclosure. c. Information that the disclosure was imparted to the individual or the individual in the respect of ConfidentialInformation imparted orally. d. information concerning the disclosure and any other information which may be included in the form of the Information. e. information about the persons, entities or persons who may incorporate any Confidential information into the information contained therein.The Disclosing Party is an Agreement that discloses Information about the Receiving Party under the terms of the Agreement. This Agreement is not binding on the parties. ✓ ✗ ✗

**ChatGPT** **(1)** The term 'Confidential Information' includes two types. Firstly, it refers to Information provided in written or tangible form, explicitly marked or designated as confidential at the time of provision. Secondly, it encompasses orally communicated information that the Disclosing Party or its representatives confirm as confidential during disclosure. It also includes any notes, records, or evaluation materials created by the Receiving Party that incorporate the orally shared Confidential Information. Moreover, it covers copies of any of the aforementioned forms and acknowledges ongoing discussions between the Receiving Party and the Disclosing Party. The term 'Disclosing Party' in this Agreement denotes the party disclosing Information to the Receiving Party directly or indirectly. ✓ ✗ ✓

**(2)** In this Agreement, the term 'Confidential Information' refers to two aspects. Firstly, it includes Information provided in written or tangible form, marked or designated as confidential upon provision. Secondly, it encompasses orally imparted information explicitly identified as confidential by the Disclosing Party or its representatives during disclosure. It also covers any notes, records, or evaluation materials created by the Receiving Party that incorporate the orally shared Confidential Information. Additionally, it comprises copies of any of the aforementioned forms and acknowledges the existence of discussions between the Receiving Party and the Disclosing Party. The term 'Disclosing Party' signifies the party disclosing Information to the Receiving Party, whether directly or indirectly, under this Agreement. ✓ ✗ ✓

| | |
|---|---|
| **Falcon** | **(1)** Confidential Information' shall mean: in respect of Information provided in documentary or by way of a model or in other tangible form, Information which at the time of provision is marked or otherwise designated to show expressly or by necessary implication that it is imparted in confidence; and in respect of Information that is imparted orally, any information that the Disclosing Party or its representatives informed the Receiving Party at the time of disclosure was imparted in confidence; and in respect of Confidential Information imparted orally, any note or record of the disclosure and any evaluation materials prepared by the Receiving Party that incorporate any Confidential Information; and any copy of any of the foregoing; and the fact that discussions are taking place between the Receiving Party and the Disclosing Party. 'Disclosing Party' shall mean the party to this Agreement that discloses Information, directly or indirectly to the Receiving Party under or in anticipation of this Agreement. ✓ ✗ ✓ |
| | **(2)** Confidential Information' shall mean: in respect of Information provided in documentary or by way of a model or in other tangible form, Information which at the time of provision is marked or otherwise designated to show expressly or by necessary implication that it is imparted in confidence; and in respect of Information that is imparted orally, any information that the Disclosing Party or its representatives informed the Receiving Party at the time of disclosure was imparted in confidence; and in respect of Confidential Information imparted orally, any note or record of the disclosure and any evaluation materials prepared by the Receiving Party that incorporate any Confidential Information; and any copy of any of the foregoing; and the fact that discussions are taking place between the Receiving Party and the Disclosing Party. 'Disclosing Party' shall mean the party to this Agreement that discloses Information, directly or indirectly to the Receiving Party under or in anticipation of this Agreement. ✓ ✗ ✓ |
| **DALE-pt** | **(1)**Confidential Information means any portion of Information provided in writing other than Information which is imparted in confidence ; Confidential Information consists of Information that is imparts orally or in writing or information that the other party agrees in writing prior to disclosure was imparts in confidence. In respect of Conf confidential Information imparted orally , the disclosure and any representation or warranty shall not be deemed to incorporate any .Knowledge Disclosing Party means any Person as defined in the Agreement that discloses Information to the Receiving Party under the ✓ ✗ ✓ |
| | **(2)** Confidential Information means any Confidential Information with respect of Information provided in writing or Information which is imparted in confidence with respect to a written communication or of Information that is imparts orally or written information that the person making the disclosure knew or should have known prior to the disclosure was imparts in confidence . Confidentiality does not include any information that was known to the party making disclosure or reasonably believed to be known to such party when making disclosure and any written communication does not expressly or impliedly incorporate any Confictional Information by reference .Exhibit 10.1 Disclosing Party Non - Disclosure Agreement that discloses Information to the Receiving Party under Section 7.1 ✓ ✗ ✓ |

| | |
|---|---|
| **DALE-ft** | **(1)** 'Confidential Information' means any and all of Information provided by the Disclosing Party to the Receiving Party, including, without limitation, Information which is marked 'proprietary,' 'confidential' or other similar legend, or if applicable, information which is imparted in confidence, including but not limited to, any and ALL notes, analyses, compilations, studies, interpretations or other documents prepared by or on behalf of Information that is Imarted orally or visually, or any other information that the Recipient reasonably should have known at the time of disclosure was imparted IN confidence, whether or not it is reduced to writing or other tangible form, and that is marked or otherwise designated as 'prior to the disclosure and any copies made thereof that incorporate any Confidential Information.'Disclosing Party' means a party to this Agreement that discloses Information to the Receiving Party under this Agreement ✓ ✓ ✓

**(2)** 'Confidential Information' shall mean any and all of Information provided by the Disclosing Party to the Receiving Party, including, without limitation, Information which is marked 'proprietary' or 'confidential', which is not generally known to the public, whether or not it is imparted in confidence, and any compilation of Information that is impersed orally, visually, electronically or otherwise, including without limitation information that the Recipient can demonstrate by documentary evidence immediately after the time of disclosure that the information received was imparted IN confidence, or that, with respect of Confidential Information imparted orally, was already in the possession of the Recipi-ent on a non-confidential basis prior to the disclosure and any compilations, studies, interpretations or other documents prepared by the Reconsequent Party that contain or incorporate any Confidential information. 'Disclosing Party' means a party to this Agreement that discloses Information to the Receiving Party under this Agreement.' ✓ ✓ ✓ |
| **DALE** | **(1)** 'Confidential Information' means, in respect of Information provided by either Party, Information which at the time of disclosure is marked or otherwise designated in writing as ' imparted in confidence' or 'proprietary' (and, in the case of Information that is imparted orally, information that the Disclosing Party or its Representatives should reasonably have understood from the circumstances surrounding the disclosure that the information sought to be obtained was being provided in confidence and that the disclosure was imparted in confidence); provided, however, that the obligations of the Receiving Party with respect of Confidential Information imparted oral shall survive the termination of the disclosure and any discussions between the Parties, notwithstanding the fact that the Parties may incorporate any information in written form that is subsequently incorporated in any compilations, analyses, studies or other documents prepared by or for the purposes of the Parties.'Disclosing Party' means a Party to this Agreement that discloses Information to the Receiving Party under or in connection with this Agreement. ✓ ✓ ✓

**(2)** Confidential Information' shall mean that portion of Information provided by the Disclosing Party ('Information') to the Receiving Party and its Representatives (as defined below) in written, graphic, electronic or other tangible form and any copies made thereof or extracts therefrom, except Information which at the time of disclosure was imparted in confidence; provided, however, that in respect of Information that is imparted orally, the information that the Recipient or its Representatives were informed by the Owner or its Representative that the information was imparted in confidence shall be protected hereunder in accordance with this Section 1.1.1 in the same manner in which it was protected prior to the disclosure and any software that incorporates any Confidential Information.'Disclosing Party' means a Party to this Agreement that discloses Information to the Receiving Party under or in connection with this Agreement. ✓ ✓ ✓ |

**UNFAIR ToS**

| | |
|---|---|
| Original | The most recent version of this agreement will be posted on the services under settings and also on gotinder.com, and you should regularly check for the most recent version. |
| **EDA** | recent version of this agreement will be posted on the services under settings and also on gotinder com and you should regularly check for the most recent version ✗ ✗ ✓ |
| **AEDA** | the most ; recent version of ; this agreement will be posted on the , services under settings and also on gotinder.com . , and you should regularly check for the most recent version . , ✗ ✗ ✓ |
| **SMERTI** | This most recent version of Windows will be posted on power under settings available on gotinder. , and you should regularly check our most recent version. ✗ ✗ ✗ |
| GENIUS | The terms of this agreement will be contingent on the services they provide. For more information, please visit www.sos.gov. ✓ ✗ ✗ |
| ChatGPT | The latest edition of this agreement will be made available on the services, specifically under the settings section and on gotinder.com. It is advisable to frequently review the most recent version. ✓ ✗ ✓ |
| **Falcon** | The most recent version of this agreement will be posted on the services under settings and also on gotinder.com, and you should regularly check for the most recent version. ✓ ✗ ✓ |
| **DALE-pt** | The most recent version of this agreement shall be accepted as the most recent amendment . ✓ ✗ ✗ |
| **DALE-ft** | the most recent version of this agreement will be posted on the services under settings and also on gotinder.com, and you should regularly check for the most most recent versions. ✓ ✗ ✓ |
| **DALE** | The most recent version of this agreement will be posted on the services's website at https://www.adr.nianticlabs.com/ where you can download and view the services, and you should be aware that this is not a guarantee that the services will be up to code or up to date, and we reserve the right to discontinue using the services at any time. ✓ ✓ ✓ |

| CASEHOLD | |
|---|---|
| Original | Juan Lopez appeals from his conviction and sentence for harboring aliens in violation of 8 U.S.C. 1324(a)(l)(A)(iii), and conspiracy in violation of 8 U.S.C. § 1324(a)(l)(A)(v)(I). We have jurisdiction pursuant to 28 U.S.C. § 1291 and 18 U.S.C. § 3742, and we affirm. Lopez argues that the district court erred when it redacted his indictment to remove reference to commercial advantage and private financial gain. Even assuming this argument was not waived, Lopez has failed to demonstrate plain error on the part of the district court. Clearly, the court's redaction did not violate the Fifth Amendment because it did not broaden the charges against Lopez; if anything, it narrowed them. |
| **AEDA** | Juan Lopez appeals from his conviction and sentence for harboring aliens in violation of 8 U.S.C. 1324(a)(l)(A)(iii), and conspiracy in violation of 8 U.S.C. § ? 1324(a)(l)(A)(v)(I). We have jurisdiction pursuant to 28 U.S.C. § 1291 and 18 ; U.S.C. § 3742, and we affirm. Lopez argues that the ? district court ; erred when it redacted his indictment to remove reference to commercial advantage and private financial ! gain. Even assuming this argument was not waived, Lopez has failed to demonstrate plain error on the part of the district court. Clearly, the court's redaction did not violate the Fifth Amendment because it did not ? broaden the charges against Lopez; if anything, it narrowed them. ✗ ✗ ✓ |

| | |
|---|---|
| **GENIUS** | Juan Lopez appeals from a conspiracy in violation of U.S. jurisdiction pursuant to the First Amendment. Lopez argues that the district court erred when it redacted his indictment to allow the Fifth Amendment to be used in the charges against Lopez, and that the indictment did not include enough evidence to prove that he had committed a crime. In his appeal, Lopez also argues that he was not charged with a crime, but rather was charged with conspiracy to commit a crime against the United States. ✓ ✓ ✗ |
| **ChatGPT** | Juan Lopez is appealing his conviction and sentencing for the offense of harboring aliens in violation of 8 U.S.C. 1324(a)(l)(A)(iii) and conspiracy in violation of 8 U.S.C. § 1324(a)(l)(A)(v)(I). Our jurisdiction is established under 28 U.S.C. § 1291 and 18 U.S.C. § 3742, and we affirm the decision. Lopez argues that the district court made an error by redacting his indictment to exclude any mention of commercial advantage and private financial gain. Even if we assume that this argument was not waived, Lopez has not provided sufficient evidence of plain error committed by the district court. It is clear that the court's redaction did not violate the Fifth Amendment, as it did not expand the charges against Lopez; if anything, it narrowed them. ✓ ✗ ✓ |
| **Falcon** | Juan Lopez appeals from his conviction and sentence for harboring aliens in violation of 8 U.S.C. 1324(a)(l)(A)(iii), and conspiracy in violation of 8 U.S.C. § 1324(a)(l)(A)(v)(I). We have jurisdiction pursuant to 28 U.S.C. § 1291 and 18 U.S.C. § 3742, and we affirm. Lopez argues that the district court erred when it redacted his indictment to remove reference to commercial advantage and private financial gain. Even assuming this argument was not waived, Lopez has failed to demonstrate plain error on the part of the district court. Clearly, the court's redaction did not violate the Fifth Amendment because it did not broaden the charges against Lopez; if anything, it narrowed them. ✓ ✗ ✗ |
| **DALE-pt** | Michael A. Lopez appeals from his conviction and sentence . We have jurisdiction pursuant to Arizona Rule of Appellate Procedure 9.160 . Lopez argues that the district court erred when it redacted his indictment to remove his privilege against self - incrimination under the Fifth Amendment because it did not broaden the charges against him ✓ ✓ ✓ |
| **DALE-ft** | Lopez appeals from his conviction and sentence. The district court did not have jurisdiction pursuant to 28 U.S.C. 1824(b)(1). Lopez argues that the district court erred when it redacted his indictment to remove reference to the use of a firearm in violation of the Fifth Amendment because it did not broaden the charges against him to reflect the nature of the firearm. ✓ ✓ ✗ |
| **DALE** | Juan Lopez appeals from his conviction and sentence for possession of a controlled substance with intent to distribute. We have jurisdiction pursuant to 28 U.S.C. § 1291, and we affirm. Lopez argues that the district court erred when it redacted his indictment to remove reference to his invocation of the Fifth Amendment because it did not broaden the charges against him. We find no error and affirm. First, we note that Lopez argument is not supported by the record. See United States v. Mendozar-Paz, 511 F.3d 1360, 1364 (11th Cir.2008) (en banc) (We review questions of law de novo.). Second, Lopez failed to preserve his argument for appeal because he did not file a motion to quash or a motion in arrest of judgment. See id. at 1364. Finally, we conclude that Lopez has failed to establish plain error under 28 C.F.R. 702.2 and 803.1(d), and we decline to address the merits of his other arguments. Lopez does not cite any authority for his argument and we know of none. In any event, we have considered and rejected the argument made by Lopez in his reply brief and find it unpersuasive. ✓ ✓ ✗ |

**JUDGEMENT**

| | |
|---|---|
| Original | if the argument of the learned counsel for the respondents is accepted, it would mean that a person whose bail under POTA has been rejected by the Special Court will have two remedies and he can avail any one of them at his sweet will. |
| **MR** | , if the argument of the learned counsel for the respondents is accepted, it would mean that a person whose bail under Act XLIII of 1953, has been rejected by the Special Court will have two remedies and he can avail any one of them at his sweet will. ✗ ✗ ✗ |
| **LwTR** | , if the issued of the learned incident, for the resignation is accepted, it would mean that a person whose decision under POTA has been custody. by the Special Court will have two remedies and he can avail any one of them at his sweet will. ✗ ✗ ✗ |
| **ChatGPT** | Should the argument presented by the counsel for the respondents be embraced, it would imply that an individual whose bail application under POTA has been denied by the Special Court will have two available courses of action, and they can choose either remedy according to their preference. ✓ ✗ ✓ |
| **Falcon** | The State represented by the learned counsel for the respondents, the respondents in the sentence is also accepted, it will mean that the learned counsel for the respondents will have one of the TADA provisions. ✓ ✗ ✓ |
| **DALE-pt** | This does not mean that a person whose bail under POTA has been rejected by the Special Court will have two remedies and he can avail any one ✓ ✗ ✓ |
| **DALE-ft** | It will , therefore , mean that a person whose bail under POTA has been rejected by the Special Court will have two remedies and he can avail any one of them at his sweet will . He also referred to the decision of Hon'ble Supreme Court in the case of Hidayatullah v . Commissioner of Income-tax , Bombay High Court ( supra ) in support of his contention that the ratio of 1:3+bunching of bail bonds does not mean that he can be convicted of an offence under Section 498-A ( 1 ) ( g ) of the I.T . Act , 1961 , as he had claimed that he could not guarantee the security of possession of the bonds without the consent of his co-accused . In this case , the point in dispute related to the validity of the assessment proceedings reopened after lapse of seven years . ✓ ✓ ✓ |
| **DALE** | Therefore , it would mean that a person whose bail under POTA has been rejected by the Special Court will have two remedies and he can avail any one of them at his sweet will . He has not bothered to see that by his own reckless and irresponsible conduct the defense contracts obtained from the Government of India and others can be canceled and the firm will not be able to furnish bank guarantees for keeping the contracts alive , and the consequence of cancellation of such orders would be penalties , damages , etc . apart from the permanent damage to the goodwill of the firm . It is , therefore , clear that if this Court decrees the suit , the decree can be executed under the reciprocal arrangement in the U.K . ✓ ✓ ✗ |
| Original | Under Article 21, Constitution of India no person can be deprived of his life or personal liberty except according "to procedure established bylaw". |
| **MR** | Under Article 21, U. P. Sales Tax Act no person can be deprived of his life or personal liberty except according "to procedure established bylaw". ✓ ✗ ✓ |
| **LwTR** | Under Article 21, Constitution of India no relating can be together. of his could or personal liberty except according "to procedure established bylaw". ✗ ✗ ✗ |
| GENIUS | In Article 21 , Constitution of India no person shall be deprived of his or her right to life or personal liberty under any procedure established bylaw of the Supreme Court , said Susamma Thomas , professor of law at Mysore University . ✓ ✗ ✗ |

| | |
|---|---|
| **ChatGPT** | According to Article 21 of the Constitution of India, no individual can be deprived of their life or personal liberty except in accordance with the "procedure established by law". ✓ ✗ ✓ |
| **Falcon** | The right to life and personal liberty is guaranteed under Article 21 of the Constitution of India. ✓ ✗ ✓ |
| **DALE-pt** | Under the Constitution of India no person can be deprived of his life or personal liberty except according to the case of Susamma Thomas . ✓ ✓ ✗ |
| **DALE-ft** | Under Article 21 , Constitution of India no person can be deprived of his life or personal liberty except according " to procedure established bylaw " . In the case of Susamma Thomas ( supra ) , the Supreme Court held as under : " 21 . Having heard learned Counsel for both the parties I am satisfied that the order of the learned Magistrate is legally correct . ✓ ✓ ✓ |
| **DALE** | Under Article 21, Constitution of India no person can be deprived of his life or personal liberty except according " to procedure established bylaw " . The Supreme Court , in the case of Susamma Thomas , 1994 ACJ 1 ( SC ) has awarded a sum of Rs . 15,000/- each on the above two heads . In the case OF SusamMA Thomas ( supra ) , the point in dispute related to the principle on which the profits should be computed with reference to certain payments which the appellant-company made , under the laws of the Republic of Karnataka , to its employees on their retirement from their service with it . After finishing the letter PW4 posted the same in the Edad Post Office in the address of PW10 , brother of the deceased . ✓ ✓ ✓ |

## PREAMBLE

| | |
|---|---|
| Original | Before The Madurai Bench Of Madras High Court Dated : 11.03.2015 Coram The Honourable Mr.Justice M.Sathyanarayanan Criminal Appeal (MD)No.256 of 2013 1. A.Abdul Rahim 2. Sheik Babu @ S.K.Babu ... Appellants/ Accused Nos.1 and 2 Vs. State represented by The Intelligence Officer, Narcotics Control Bureau, South Zonal Unit, Chennai, O.R.No.48/1/7/2006- N.C.B. Madras. ... Respondent/ Complainant Prayer : Appeal filed under Section 374(2) of the Code of Criminal Procedure, to call for the records relating to judgment in C.C.No.523 of 2006, dated 27.04.2013 passed by the learned Additional District Judge for Principal Special Judge for N.D.P.S. Act Cases, Madurai and to set aside the conviction and sentence passed by the trial Court. !For Appellants: Mr.M.Ramu for Mr.B.Pandiarajan F̂or Respondent: Mr.C.Arul Vadivel @ Sekar Special Public Prosecutor for Ncb :Judgment |
| **MR** | Before The Madurai Bench Of Madras High Court Dated : 11.03.2015 Coram The Honourable Mr.Justice M.Sathyanarayanan Criminal Appeal (MD)No.256 of 2013 1. A.Abdul Rahim 2. Sheik Babu @ S.K.Babu ... Appellants/ Accused Nos.1 and 2 Vs. Superintendent of Prison, Yerwada Central Prison, Pune. represented by The Intelligence Officer, Narcotics Control Bureau, South Zonal Unit, Chennai, O.R.No.48/1/7/2006- N.C.B. Madras. ... Respondent/ Complainant Prayer : Appeal filed under Section 374(2) of the Code of Criminal Procedure, to call for the records relating to judgment in C.C.No.523 of 2006, dated 27.04.2013 passed by the learned Additional District Judge for Principal Special Judge for N.D.P.S. Act Cases, Madurai and to set aside the conviction and sentence passed by the trial Court. !For Appellants: Mr.M.Ramu for Mr.B.Pandiarajan F̂or Respondent: Mr.C.Arul Vadivel @ Sekar Special Public Prosecutor for Ncb :Judgment ✗ ✗ ✓ |

| | |
|---|---|
| **LwTR** | Before The Madurai Bench Of Addl.City Judge/Special Court Senior : Hon'ble Nos.1 The Honourable Mr.Justice M.Sathyanarayanan Criminal Appeal Jai of P N A.Abdul Rahim Pur, Sheik Bank Velentina S.K.Babu ... Appellants/ Accused 2004(B) and 2 ... State 11781, by The Intelligence Officer, Narcotics Control Bureau, South Mr. 04.12.2014 Chennai, O.R.No.48/1/7/2006- (Orig. Madras. ... Respondent/ Complainant Prayer : Justice filed under Section 374(2) of the Code of Nand Procedure, to call for the records relating to judgment in C.C.No.523 of Shri. Summons 27.04.2013 passed by the learned Additional District Jayant for High Kerala Court for 02 Act Cases, Madurai and to set 1. the 002. and sentence passed by the trial Court. !For Appellants: Mr.M.Ramu for Mr.B.Pandiarajan Ekta Appellants Mr.C.Arul Vadivel @ Sekar R/o Public Prosecutor for Ncb :Judgment ✗ ✗ ✓ |
| **GENIUS** | Before The Madurai Bench Of Madras High Court Dated , the Chief Justice of the Madras Bench of the Supreme Court had issued a notice to the President of India . The Chief Justice had directed the President to take immediate action against the President ✓ ✗ ✗ |
| **ChatGPT** | Before The Madurai Bench Of Madras High Court Dated: 11.03.2015 Coram The Honourable Mr. Justice M. Sathyanarayanan Criminal Appeal (MD) No. 256 of 2013 1. A. Abdul Rahim 2. Sheik Babu @ S.K. Babu ... Appellants/ Accused Nos. 1 and 2 Vs. State represented by The Intelligence Officer, Narcotics Control Bureau, South Zonal Unit, Chennai, O.R.No. 48/1/7/2006- N.C.B. Madras. ... Respondent/ Complainant Prayer: Appeal filed under Section 374(2) of the Code of Criminal Procedure, to obtain the records pertaining to the judgment in C.C.No.523 of 2006, dated 27.04.2013, passed by the learned Additional District Judge for Principal Special Judge for N.D.P.S. Act Cases, Madurai, and to overturn the conviction and sentence imposed by the trial Court. For Appellants: Mr. M. Ramu for Mr. B. Pandiarajan For Respondent: Mr. C. Arul Vadivel @ Sekar, Special Public Prosecutor for NCB Judgment ✓ ✗ ✓ |
| **Falcon** | Before The Madurai Bench Of Madras High Court Dated : 11.03.2015 Coram The Honourable Mr.Justice M.Sathyanarayanan Criminal Appeal (MD)No.256 of 2013 1. A.Abdul Rahim 2. Sheik Babu @ S.K.Babu .. Appellants/ Accused Nos.1 and 2 Vs. State represented by The Intelligence Officer, Narcotics Control Bureau, South Zonal Unit, Chennai, O.R.No.48/1/7/2006- N.C.B. Madras. .. Respondent/ Complainant Prayer : Appeal filed under Section 374(2) of the Code of Criminal Procedure, to call for the records relating to judgment in C.C.No.523 of 2006, dated 27.04.2013 passed by the learned Additional District Judge for Principal Special Judge for N.D.P.S. Act Cases, Madurai and to set aside the conviction and sentence passed by the trial Court. For Appellants: Mr.M.Ramu for Mr.B.Pandiarajan For Respondent: Mr.C.Arul Vadivel Sekar Special Public Prosecutor for Ncb :Judgment ✓ ✗ ✓ |
| **DALE-pt** | Before The Madurai Bench Of Madras High Court Dated May 1 In The High Court Of Judicature At Bombay. ✓ ✗ ✗ |

| | |
|---|---|
| **DALE-ft** | Before Mad The CourtInP Before Of High1TheableBeforeBeforeBefore Mad MadBefore Mad MadBefore TheBeforeInBeforePBefore TheBeforeBeforeBefore TheBeforeInBefore BeforeBefore CourtBefore HighBefore OfBeforePBeforeicuraiCase : JusticerasHigh . Bench D At In No . : etitionBleicBeforeTheBefore1Before CourtBefore HighBefore OfBefore BeforeBeforeTheBefore1BeforeBefore MadBeforeBeforeIn The The The MadIn Mad TheIn Mad Court Mad The MadP MadInIn The Of The Court Mad High MadP Mad Before Mad1Beforeable'Crature 1No Coram Honour PMad-atedD CrA Appeal Justice ( erReportCd W of/ the Supreme Jud DrDr Hon Criminal ShAt * CbleAs14Pre10APThakMHeadBeforeHighBeforeCaseBefore BenchBeforeJustice Mad Court TheIn Court The High Mad1 Mad High The Before Mad Before TheP The Court CourtIn TheBefore Mad MadBefore Mad TheBeforeInBeforeBefore BeforeBeforeBefore CourtBeforeBefore1 MadThe Mad Of MadJusticeBeforerasBeforeuraiBefore InBefore.Before DBefore : BeforeBleBefore . : Before NoBefore AtBeforeCrBeforeNoBeforeetition The Before The1 TheP The High CourtPIn1In Court Of Mad Of The1 The OfInThe MadThe TheTheIn BeforeInPInIn High ✗ ✗ ✗ |
| **DALE** | Before The Madurai Bench Of Madras High Court Dated : 02/08/2011 Coram The Honourable Mr.Justice K.N.Keshavanarayana M.A.C.M.A ( MD ) No.1 of 2011 and M.P.No.2 of 2011 M.V.Sasidharan ... Appellant vs . 1.The State of Tamil Nadu , Represented by its Secretary to Government , Home Department , Fort St . George , Chennai-600 009 . 2.The Superintendent of Police , Madurai . 3.The District Magistrate , Thiruvananthapuram District . 4.The Special Tahsildar , District Munger . ... Respondents Prayer : Appeal filed under Section 100 of the Code of Civil Procedure , against the judgment and decree dated 18.06.2011 passed in O.S.Nos.2 and 3 of 2001 on the file of the Learned Additional District and Sessions Judge , Tiruvallur . ! For Appellants ... Mr.T.R.Rajagopalan Fˆor Respondents ... Mr.K.Ravindran , Additional Public Prosecutor for R1 to R3 : Judgment In The Court Of Shri Puneet Ahuja Addl . District Judge-01 ( Central ) : Tis Hazari Courts : Delhi Suit No . 14/14 Unique Case Id No . 02401C0178532011 1 . M/s . Bajajakshmi Textilesh S/o Shri Kishore Shrivastava , R/o . Kirti Nagar , Near Kailash Bazar , New Delhi . .......... Plaintiff Versus 1 . Shri Suresh Chand Mishra 2 . Smt . Nirmala Devi W/o Late Shri Chandrashekar Mishra , Proprietor of Shop No.14/14 , Rz2/F 2nd Floor , Jafrara Apartment , Dda Flats , Laxmi Ganj , Kolkata , Distt . Bulandshwar , Udupi , Bihar . ............. Defendants Date of institution of the suit : 29.03.2011 Date of reserving the judgment : 31.05.2014 Date of pronouncement of judgment : 17.07.2014 Suit For Possession , Declaration , Cancellation And Recovery Of Damages And Permanent Injunction Judgment In The High Court. ✓ ✓ ✓ |

| | |
|---|---|
| Original | In The High Court Of Judicature At Madras Dated: 09/08/2006 Coram The Hon'Ble Mr. Justice A. Kulasekaran W.P.No.18050 of 2005 and W.P.No.18051 of 2005 R. Kumar .. Petitioner in W.P.No. 18050 1. Ramdass Bharadwaj 2. Meerabhat 3. Sukanya Rao 4. Shantharam Bharadwaj .. Petitioners in 5. Achyut Bharadwaj Wp No. 18051 -Vs- 1.State of Tamil Nadu rep. By its Secretary to Government Highways Department Fort St. George Chennai 600 009 2. The Member Secretary Chennai Metropolitan Development Authority Chennai 600 008 3. The District Collector Kancheepuram District Kancheepuram 4. The Special Tahsildar (L.A.) I.T. Expressway Scheme .. Respondents in Tambaram both the Writ Chennai 600 047 Petitions Wp No. 18050 and 18051 of 2005: Petitions filed under Article 226 of The Constitution of India praying for a Writ of Declaration declaring that the notification issued by the first defendant in G.O. Ms. No. 92, Highways (Hw1) 25.04.2005, published in Gazzette No.II (2)/Hw/(3 40-e-2)/ 2005 under Section 15 (1) of the Tamil Nadu Highways Act, 20 01 in so far as it relates to acquisition of the property of the petitioner situated at Government Manavari Survey No.277-5 (part) now sub-divided as 277-5B, No.44, Kottivakkam Village, Tambaram Taluk, Kancheepuram District, beyond 23 feet from the existing Western boundary of the petitioner's land as per the sanctioned plan and master plan of the second respondent, is illegal, arbitrary, discriminatory and colourable exercise of power, and inconsistent with the petitioner's lawful right acquired and become final under the provisions of the Tamilnadu Town and Country Planning Act, 1971 as per the permission issued by the 2nd respondent. !For Petitioner : Mr. K.M. Vijayan, Senior Counsel for M/s. La and Law in both the Writ Petitions For Respondents : Mr. P.S. Raman Additional Advocate General assisted by Mr. M. Dhandapani Additional Government Pleader :Common Order |
| MR | In The High Court Of Judicature At Madras Dated: 09/08/2006 Coram The Hon'Ble Mr. Justice A.Ramalingeswara Rao W.P.No.18050 of 2005 and W.P.No.18051 of 2005 R. Kumar .. Petitioner in W.P.No. 18050 1. Ramdass Bharadwaj 2. Meerabhat 3. Sukanya Rao 4. Bharat Shashikant Patel .. Petitioners in 5. Board Of High School & Intermediate Education, U. P., Wp No. 18051 -Vs- 1.State of Tamil Nadu rep. By its Secretary to Government Highways Department Fort St. George Chennai 600 009 2. Opera Clothing, 3. Anuj Arya, 4. The Special Tahsildar (L.A.) I.T. Expressway Scheme .. Respondents in Tambaram both the Writ Chennai 600 047 Petitions Wp No. 18050 and 18051 of 2005: Petitions filed under Article 226 of The Constitution of India praying for a Writ of Declaration declaring that the notification issued by the first defendant in G.O. Ms. No. 92, Highways (Hw1) 25.04.2005, published in Gazzette No.II (2)/Hw/(3 40-e-2)/ 2005 under Section 15 (1) of the Tamil Nadu Highways Act, 20 01 in so far as it relates to acquisition of the property of the petitioner situated at Government Manavari Survey No.277-5 (part) now sub-divided as 277-5B, No.44, Kottivakkam Village, Tambaram Taluk, Kancheepuram District, beyond 23 feet from the existing Western boundary of the petitioner's land as per the sanctioned plan and master plan of the second respondent, is illegal, arbitrary, discriminatory and colourable exercise of power, and inconsistent with the petitioner's lawful right acquired and become final under the provisions of the Tamilnadu Town and Country Planning Act, 1971 as per the permission issued by the 2nd respondent. !For Petitioner : Mr. K.M. Vijayan, Senior Counsel for M/s. La and Law in both the Writ Petitions For Respondents : Mr. P.S. Raman Additional Advocate General assisted by Mr. Shreya Parikh Additional Government Pleader :Common Order ✓ ✗ ✓ |

**LwTR**  In The High Court Of Judicature At Madras No.90 09/08/2006 Coram The Hon'Ble Mr. Justice A. Kulasekaran W.P.No.18050 of 2005 and W.P.No.18051 of 07.02.2014 R. Kumar .. Petitioner in W.P.No. 18050 1. Ramdass Bharadwaj 2. Meerabhat 3. Sukanya Rao 4. Shantharam Binjraj 91/1, Petitioners in 5. Achyut Bharadwaj Versus No. Respondent -Vs- 1.State of Tamil Nadu Appeal By its Secretary to 1979 E.41A, C.A.No.352/2007, Fort St. George Chennai 600 Special 2. The Member Secretary Chennai Devi Development Pvt.ltd. Chennai Kanyakumari 008 3. The District Collector Kancheepuram Highways Kancheepuram Sh.Mohd. The Special Tahsildar (L.A.) I.T. Expressway Scheme .. 2003, in Tambaram both the Writ Sc 600 047 Petitions Wp No. 18050 and 18051 of 2005: Petitions filed under 4. 11, of The W.A.No.997 of Bevur praying for a Writ of Advocate declaring that the Fir issued by the first defendant in G.O. Ms. No. Prosecutor Highways assisted Fir No.2472/2015 in No. No.II (2)/Hw/(3 40-e-2)/ 2005 under Section 15 Coram of the Tamil ——————————— ———————————- Highways Act, 20 01 in so far as it relates to acquisition of the property of the petitioner situated at Government Sh. Orders No.277-5 (part) now sub-divided as 277-5B, No.44, Kottivakkam Village, Tambaram Taluk, Kancheepuram Shri others. 23 feet from the T.V. Western boundary of the petitioner's land as per the sanctioned Appeals and master plan of the second respondent, is Of. Shri. discriminatory and Justice exercise of power, and inconsistent with the petitioner's Sh. right acquired and become final under the provisions of the Dr.D.Y.Chandrachud, Hon'ble and Country Planning versus 1971 as per the permission issued by the Sh. 15667 !For Deputy : Mr. K.M. Vijayan, Senior Counsel for Superintendent Ground and Law in both the Writ Petitions For Respondents : Mr. P.S. Raman Additional made Criminal assisted by Mr. M. Mardikar, No. Government Pleader :Common Order ✗  ✗  ✓

**GENIUS**  In The High Court Of Judicature At Madras Dated on May 15 , 2014 , Respondents in Tambaram both the Writ Chennai 600 047 Petitions Wp No . 1 and the Petition No . 2 filed a petition challenging the constitutionality of the Tamil Nadu Highways Act , 1961 . The petitioners argued that the Act was unconstitutional and that it was a violation of the fundamental rights of the people of Tamil Nadu . ✓  ✗  ✗

| | |
|---|---|
| **ChatGPT** | In The High Court Of Judicature At Madras Dated: 09/08/2006 Coram The Hon'ble Mr. Justice A. Kulasekaran W.P.No.18050 of 2005 and W.P.No.18051 of 2005 R. Kumar .. Petitioner in W.P.No. 18050 1. Ramdass Bharadwaj 2. Meerabhat 3. Sukanya Rao 4. Shantharam Bharadwaj .. Petitioners in 5. Achyut Bharadwaj Wp No. 18051 -Vs- 1. State of Tamil Nadu rep. By its Secretary to Government Highways Department Fort St. George Chennai 600 009 2. The Member Secretary Chennai Metropolitan Development Authority Chennai 600 008 3. The District Collector Kancheepuram District Kancheepuram 4. The Special Tahsildar (L.A.) I.T. Expressway Scheme .. Respondents in Tambaram. Both the Writ Petitions Wp No. 18050 and 18051 of 2005: Petitions filed under Article 226 of The Constitution of India, praying for a Writ of Declaration, declaring that the notification issued by the first defendant in G.O. Ms. No. 92, Highways (Hw1) 25.04.2005, published in Gazzette No.II (2)/Hw/(3 40-e-2)/ 2005, under Section 15 (1) of the Tamil Nadu Highways Act, 2001, insofar as it relates to the acquisition of the property of the petitioner situated at Government Manavari Survey No.277-5 (part) now sub-divided as 277-5B, No.44, Kottivakkam Village, Tambaram Taluk, Kancheepuram District, beyond 23 feet from the existing Western boundary of the petitioner's land as per the sanctioned plan and master plan of the second respondent, is illegal, arbitrary, discriminatory, and a colorable exercise of power, and inconsistent with the petitioner's lawful right acquired and become final under the provisions of the Tamilnadu Town and Country Planning Act, 1971, as per the permission issued by the 2nd respondent. For Petitioner: Mr. K.M. Vijayan, Senior Counsel for M/s. La and Law in both the Writ Petitions For Respondents: Mr. P.S. Raman, Additional Advocate General, assisted by Mr. M. Dhandapani, Additional Government Pleader Common Order ✓ ✗ ✓ |
| **Falcon** | The High Court Of Judicature At Madras Dated: 09/08/2006 Coram The Hon'Ble Mr. Justice A. Kulasekaran W.P.No.18050 of 2005 and W.P.No.18051 of 2005 R. Kumar . Petitioner in W.P.No. 18050 1. Ramdass Bharadwaj 2. Meerabhat 3. Sukanya Rao 4. Shantharam Bharadwaj . Petitioners in 5. Achyut Bharadwaj Wp No. 18051 -Vs- 1.State of Tamil Nadu rep. By its Secretary to Government Highways Department Fort St. George Chennai 600 009 2. The Member Secretary Chennai Metropolitan Development Authority Chennai 600 008 3. The District Collector Kancheepuram District Kancheepuram 4. The Special Tahsildar (L.A.) I.T. Expressway Scheme . Respondents in Tambaram both the Writ Chennai 600 047 Petitions Wp No. 18050 and 18051 of 2005: Petitions filed under Article 226 of The Constitution of India praying for a Writ of Declaration declaring that the notification issued by the first defendant in G.O. Ms. No. 92, Highways (Hw1) 25.04.2005, published in Gazzette No.II (2)/Hw/(3 40-e-2)/ 2005 under Section 15 (1) of the Tamil Nadu Highways Act, 20 01 in so far as it relates to acquisition of the property of the petitioner situated at Government Manavari Survey No.277-5 (part) now sub-divided as 277-5B, No.44, Kottivakkam Village, Tambaram Taluk, Kancheepuram District, beyond 23 feet from the existing Western boundary of the petitioner's land as per the sanctioned plan and master plan of the second respondent, is illegal, arbitrary, discriminatory and colourable exercise of power, and inconsistent with the petitioner's lawful right acquired and become final under the provisions of the Tamilnadu Town and Country Planning Act, 1971 as per the permission issued by the 2nd respondent. For Petitioner : Mr. K.M. Vijayan, Senior Counsel for M/s. La and Law in both the Writ Petitions For Respondents : Mr. P.S. Raman Additional Advocate General assisted by Mr. M. Dhandapani Additional Government Pleader :Common Order ✓ ✗ ✓ |
| **DALE-pt** | In The High Court Of Judicature At Madras Dated the Respondents in Tambaram both the Writ Chennai 600 047 Petitions Wp No . of the Chennai Chennai Highways Act ✓ ✓ ✓ |

**DALE-ft**  In High The Jud Court Madature Of . HonourP : icNo AtablerasInInIn HighIn CourtIn JudIn TheIn MadIn TheInatureIn CourtIn HighIn JudIn OfInPIn.In Mad High The Jud TheatureInature The High High Jud High The Court High Jud Judature The Jud The High High Mad The The Court High Courtatureature Jud Highature Jud Court Theature High Court Jud Jud Court Court . The MadIn OfInPIn.Iner1etitionam In No Pradesh of in D Nadu 1A-ated , Justiceer Highatureature Court The The Of HighicInic The Madature Courtature MadInIn JudInIn Mad Judature High . High Mad The AtInicIn At High At The . Mad Court Jud Mad.ature . High Of Jud . Jud Of The . The HonourIn AtInerInrasIn : InamIn Honour Jud Mad . Jud . Court Mad High Of HighPature Mad Court Court Of Mad Mad Of Jud Of The Ofature HonourInras Hon/Justice Writ theC Bench State & Jud Petition Respond PHighThe Mr ' Districtpurans Judge ( Civil2 India Cor App06S Appeal AndD 2006as Vs Chennai CourtsM W Criminalate Supremeinalach ✓ ✗ ✓

**DALE**  In The High Court Of Judicature At Madras Dated : 07/04/2006 Coram The Honourable Mr.Justice K.N.Keshavanarayana W.P.Nos.1881 and 1882 of 2005 and M.P . ( MD ) No.1 of 2005 K.Lakshmi ... Petitioner Vs . 1.The Union of India , Rep . by its Secretary to Government , Finance Department , Fort St.George , Chennai-600 009 . 2.The Special Tahsildar ( Road Transport Corporation of India ) ... Respondents Writ Petition filed under Article 226 of the Constitution of India praying for issuance of a Writ of Mandamus , directing the respondents 1 to 3 to pay a sum of Rs.2,00,000/- to the petitioner as compensation for the death of petitioner caused by electrocution . ! For Petitioner : Mr.S.Srinivasan F̂or Respondents : Mrs.M.Vijayakumar , Govt.Advocate General for R1 to R3 : Judgment In The Court Of Shri Puneet Pahwa Metropolitan Magistrate : Special Court - 05 ( Ni Act ) : Dwarka : Delhi Cc No . 05/13 Case Id No . 02402C0139632014 In the matter of : State Vs . Rajender Kumar & Ors . Fir No . : 279/13 U/s . : 138 N.I.Act JUDGMENT Petitioner : Commissioner Of Income-Tax , West Bengal , Calcutta Vs . Respondent : Union Of India & Anr . Date Of Judgment In The Supreme Court Of India Civil Appellate Jurisdiction Civil Appeal No.8665 Of 2010 ( Arising out of Slp ( C ) No.2178 of 2009 ) Narinder Kumar ..... Appellant Vs . State of A.P . , Represented by Learned Public Prosecutor , High Court of Karnataka , Hyderabad and four others ..... Respondents Date of institution of appeal : 06.08.2010 Date of reserving judgment : 29.09.2014 Date of pronouncement of judgment : 30.09,2014 :COMMON ORDER ✓ ✓ ✓

## LEDGAR

Original    The Borrower acknowledges, represents, warrants and agrees as to itself and all other Loan Parties, and each other Loan Party acknowledges, represents, warrants and agrees as to itself, that: (a) after giving effect to this Agreement, the representations and warranties contained in the Credit Agreement, as amended hereby, and the representations and warranties contained in the other Loan Documents are true and correct in all material respects on and as of the Effective Date and on the date hereof as if made on as and as of such date except to the extent that any such representation or warranty expressly relates solely to an earlier date, in which case such representation or warranty is true and correct in all material respects as of such earlier date; (b) the execution, delivery and performance of this Agreement are within the limited liability company or corporate power and authority of such Loan Party and have been duly authorized by appropriate limited liability company and corporate action and proceedings; (c) this Agreement constitutes the legal, valid, and binding obligation of such Loan Party enforceable in accordance with its terms, except as limited by applicable bankruptcy, insolvency, reorganization, moratorium, or similar laws affecting the rights of creditors generally and general principles of equity, and no portion of the Obligations are subject to avoidance, subordination, recharacterization, recovery, attack, offset, counterclaim, or defense of any kind; (d) there are no governmental or other third party consents, licenses and approvals required to be made or obtained by it in connection with its execution, delivery, performance, validity and enforceability of this Agreement; (e) no Defaults or Events of Default shall have occurred and be continuing; and (f) since the date of the financial statements most recently delivered pursuant to Section 6.01(a) of the Credit Agreement, there has been no event or circumstance, either individually or in the aggregate, that has had or could reasonably be expected to have a Material Adverse Effect.

EDA    the borrower acknowledges represents atomic number warrants and agrees as to itself and all other loan parties and each other loan party concord make acknowledges represents warrants and agrees particular date as to itself that a after giving effect to this agreement the representations and warranties receipt contained in the credit agreement as amended hereby and the representations and warranties contained in the other loan documents are true and aggregative correct in all material respects on and as of no more the effective date and on the date hereof as if edge made on as and as of such date except to the extent that any such lustiness representation default option or warranty expressly relates solely to an earlier date in atomic number which case such representation or warranty is true and inch correct in all material respects as of such lend earlier date b the execution operating room delivery and want performance abide by severally of this agreement are within the lend limited liability company or corporate natural law power and afterwards authority of such loan party and have been duly authorized by appropriate limited liability company and theatrical performance corporate action and proceedings c effectual this agreement theatrical performance constitutes the legal valid and binding obligation of such loan party enforceable in accordance with its terms except as limited by applicable bankruptcy insolvency reorganization moratorium or similar laws affecting the rights of creditors generally and general principles of equity and no portion of the obligations are subject to avoidance subordination recharacterization recovery attack offset reserve counterclaim or defense of set any inch kind d there are no governmental or other third party consents licenses and approvals society inch required to be made or obtained no more by it in connection with its execution delivery performance validity and enforceability of this agreement e no defaults or events of default shall have occurred and be continuing and f gist since the date of the financial statements most recently delivered pursuant gist to section equal a of the credit agreement there has been no event or circumstance either individually or in the aggregate that has had or could reasonably be expected to have a material adverse effect ✗ ✗ ✓

| | |
|---|---|
| **Legal EDA** | The borrower acknowledges, represents, warrant and agrees as to itself and all other lend political party, and each other lend political party acknowledges, represents, warrant and agrees as to itself, that: (a) after giving result to this accord, the representation and warranty contained in the credit entry accord, as amended herewith, and the representation and warranty contained in the other lend document are genuine and right in all stuff respect on and as of the efficient particular date and on the particular date hereof as if made on as and as of such particular date except to the extent that any such internal representation or guarantee expressly relates exclusively to an early particular date, in which lawsuit such internal representation or guarantee is genuine and correct in all stuff respect as of such early particular date; (b) the instruction execution, obstetrical delivery and public presentation of this accord are within the special financial obligation ship's company or collective major power and self-assurance of such lend political party and have been punctually authorised by allow limited financial obligation ship's company and collective legal action and legal proceeding; (c) this accord constitute the sound, valid, and binding responsibility of such lend political party enforceable in accordance of rights with its full term, except as limited by applicable failure, insolvency, reorganisation, moratorium, or like law affecting the right of creditor more often than not and full general principle of fairness, and no part of the obligations are subject to avoidance, subordination, recharacterization, recovery, attack, offset, counterclaim, or defense of any kind; (d) there are no governmental or other third party consents, licenses and approvals required to be made or obtained by it in connection with its execution, delivery, performance, validity and enforceability of this agreement; (e) no defaults or events of default shall have occurred and be continuing; and (f) since the date of the financial statements most recently delivered pursuant to section 6.01(a) of the credit agreement, there has been no event or circumstance, either individually or in the aggregate, that has had or could reasonably be expected to have a material adverse effect. ✗ ✗ ✓ |
| **AEDA** | The Borrower acknowledges, ? represents, warrants : and agrees as to itself and all . other ; Loan Parties, and each other Loan Party , acknowledges, represents, ; warrants and agrees as to itself, that: (a) after giving effect ? to this . Agreement, the representations and : warranties . contained ! in the Credit , Agreement, , as ! amended hereby, and the representations ! and warranties contained ? in the other Loan Documents are true . and correct in all : material ; respects on ! and as ! of the . Effective Date and ; on ! the ; date hereof . as if made on as and as of such date : except ? to , the extent that any such ; representation or warranty expressly relates solely to ? an earlier date, in ? which case such representation or , warranty . is : true and , correct in , all material ! respects as ? of such : earlier date; (b) the execution, . delivery ! and performance of this Agreement ? are , within the limited liability . company or corporate power ! and authority of ? such Loan Party and have been ! duly authorized , by appropriate limited liability company and ; corporate action : and proceedings; (c) this Agreement constitutes . the legal, valid, and ; binding obligation : of such Loan Party ! enforceable : in accordance with ; its , terms, except as limited by . applicable bankruptcy, insolvency, reorganization, moratorium, or similar laws affecting the rights of creditors ; generally and ! general principles of . equity, and no portion of the ? Obligations ; are subject . to avoidance, subordination, recharacterization, ? recovery, attack, : offset, counterclaim, or ! defense of any : kind; (d) there : are ? no governmental . or other third party . consents, licenses and approvals : required to be made or obtained ; by it : in connection with its , execution, ; delivery, : performance, validity and enforceability of this Agreement; (e) no Defaults or Events of Default shall have . occurred and be continuing; : and (f) since ; the . date of the , financial statements most , recently delivered pursuant . to Section 6.01(a) of the ! Credit Agreement, there has been no event or circumstance, either individually or in the ? aggregate, that has had ? or could : reasonably be expected to ? have a Material Adverse Effect. ✗ ✗ ✗ |

**SSMBA** the borrower acknowledges, represents, renders and agrees as to itself and all other loan parties, and each other loan party acknowledges, and all warrants and agrees as to itself in : or ( a ). giving effect of to agreement, the representations and warranties contained in the credit agreement, so amended hereby, and the representations and warranties contained into the new loan parties both true and correct in all other respects on occasion as for the effective term and on the date here made as if made on as and not of such date except at the extent that any such representation or warranty grossly been, at an earlier date, from which case such representation of warranty is true and valid in all material respects as of such earlier date... in bmse execution, delivery and performance of this agreement are by the actual liability, or legal actions and liability of such loan party and have been duly authorized by appropriate public liability company and corporate action or proceedings ; ( c ) this agreement constitutes an legitimate, valid, and effective performance of such loan party actionable into accordance with these terms, unless as limited by applicable bankruptcy, insrivency, reorganization and moratorium, or similar laws affecting individual rights of creditors generally and any freedom of property, and no portion of the obligations are subject to avoidance, subordinatory, recaracterization, recovery, attack, offset, counter balanceweight or defense e - kind ; ( f ) there are no governmental or other third party consent agreements, licenses and catastrophes shall to be made as prescribed by it in except to its execution, delivery of performance. validity in enforceability of this agreement ; ( e ) no defaults or events or default shall have occurred and be continuing ; and or f ) since the date of that financial year most are delivered pursuant to section 8. 01 ( a ) of the following article if there has been no event or circumstance, either individually / and nor individually, that has had or could reasonably be expected to have a material adverse effect ✓ ✗ ✓

**GENIUS** The Borrower acknowledges, warrants and agrees to the Loan. The Loan also warrants and acknowledges that this Agreement contains the representations and warranties contained in the Credit Agreement. None of the representations or warranties expressed in the Loan, nor any representation or warranty expressly relates to whether such representation or warranties is true or not. The terms of this Agreement are not limited liability and the Loan shall not be limited liability. This Agreement constitutes an obligation of the Loan Party enforceable by applicable bankruptcy laws and the rights of creditors. All of the Obligations are subject to the terms and conditions set forth in this Agreement. The Credit Agreement does not include any warranties or warranties as to the accuracy or completeness of the information contained herein. ✓ ✗ ✗

**ChatGPT** The Borrower acknowledges, represents, warrants, and agrees, on behalf of itself and all other Loan Parties, and each Loan Party acknowledges, represents, warrants, and agrees, on behalf of itself, that: (a) Following the implementation of this Agreement, the representations and warranties outlined in the Credit Agreement, as amended by this Agreement, as well as the representations and warranties contained in the other Loan Documents, are true and accurate in all material respects as of the Effective Date and the present date, as if made on and as of such date, unless explicitly stated otherwise for representations and warranties pertaining solely to an earlier date. In such cases, those representations and warranties are true and correct in all material respects as of the earlier date. (b) The execution, delivery, and performance of this Agreement fall within the limited liability company or corporate power and authority of the respective Loan Party, and have been duly authorized by appropriate limited liability company and corporate actions and proceedings.(c) This Agreement constitutes a legally binding obligation of the respective Loan Party, enforceable in accordance with its terms, except as limited by applicable bankruptcy, insolvency, reorganization, moratorium, or similar laws that affect the rights of creditors in general, as well as general principles of equity. Furthermore, no portion of the Obligations is subject to avoidance, subordination, recharacterization, recovery, attack, offset, counterclaim, or defense of any kind. (d) No governmental or third-party consents, licenses, or approvals are required to be obtained or made by the respective Loan Party in connection with the execution, delivery, performance, validity, and enforceability of this Agreement. (e) No Defaults or Events of Default have occurred and are currently ongoing. (f) Since the date of the most recently provided financial statements in accordance with Section 6.01(a) of the Credit Agreement, no event or circumstance, whether individually or in aggregate, has taken place that has had or could reasonably be expected to have a Material Adverse Effect.✓  ✗  ✓

**Falcon** The Borrower acknowledges, represents, warrants and agrees as to itself and all other Loan Parties, and each other Loan Party acknowledges, represents, warrants and agrees as to itself, that: (a) after giving effect to this Agreement, the representations and warranties contained in the Credit Agreement, as amended hereby, and the representations and warranties contained in the other Loan Documents are true and correct in all material respects on and as of the Effective Date and on the date hereof as if made on as and as of such date except to the extent that any such representation or warranty expressly relates solely to an earlier date, in which case such representation or warranty is true and correct in all material respects as of such earlier date; (b) the execution, delivery and performance of this Agreement are within the limited liability company or corporate power and authority of such Loan Party and have been duly authorized by appropriate limited liability company and corporate action and proceedings; (c) this Agreement constitutes the legal, valid, and binding obligation of such Loan Party enforceable in accordance with its terms, except as limited by applicable bankruptcy, insolvency, reorganization, moratorium, or similar laws affecting the rights of creditors generally and general principles of equity, and no portion of the Obligations are subject to avoidance, subordination, recharacterization, recovery, attack, offset, counterclaim, or defense of any kind; (d) there are no governmental or other third party consents, licenses and approvals required to be made or obtained by it in connection with its execution, delivery, performance, validity and enforceability of this Agreement; (e) no Defaults or Events of Default shall have occurred and be continuing; and (f) since the date of the financial statements most recently delivered pursuant to Section 6.01(a) of the Credit Agreement, there has been no event or circumstance, either individually or in the aggregate, that has had or could reasonably be expected to have a Material Adverse Effect.✓  ✗  ✓

| | |
|---|---|
| **DALE-pt** | Each Loan Party hereby represents and warrants and agrees to the other Loan Parties that such Loan Party acknowledges and agrees as an inducement to this Agreement : i The representations and warranties contained in the Credit Agreement, as amended and supplemented by this Agreement , and ii All other representations , warranties and covenants contained in this Agreement or any other Loan Documents are true and correct except to the extent such representation or warranty expressly relates to a specified date or period of time when the failure to be so true or correct without regard to any qualification as to materiality or Material Adverse Effect set forth therein or words of similar import contained in such representations or warranties is true ; ii The obligations of each Loan Party of this Agreement are within the limited liability company limits of such Loan Partys limited liability companies and this Agreement constitutes the valid and legally binding obligation of Such Loan Party enforceable in accordance with its terms except as limited by applicable bankruptcy , insolvency or similar laws affecting the rights of creditors generally and subject to general principles of equity regardless of whether enforcement of the Obligations is considered in a proceeding in equity or at law and iii All third party consents , authorizations and approvals necessary to authorize the execution and delivery and enforceability of this AGREEMENT have been obtained and are in full force and effect pursuant to the terms and conditions of Section 5.1b of the Revolving Credit Agreement. ✓ ✓ ✓ |
| **DALE-ft** | The Borrower acknowledges, represents, warrants and agrees as to itself and all other Loan Parties, that: (a) the Loan Party acknowledges, acknowledges, warrants, agrees and agrees that (i) after giving effect to this Agreement, the representations and warranties contained in the Credit Agreement are true and correct on all material respects as of the date hereof and as of such date in each other Loan Document executed and delivered by such Loan Party; (ii) no representation or warranty expressly relates to an Event of Default or a liquidation of the Assets; provided , however , that in the event of a default by the Administrative Agent or any other designated holder of the Debt (other than a holding company, investment vehicle or trust for the primary benefit of the primary beneficiaries of the indebtedness incurred by the Loan Party) or (iii) any other Loan Party liquidating its assets prior to the making of any payment required by law, this Agreement constitutes the legal, valid and binding obligation of each Loan Party enforceable in accordance with its terms, except as limited by applicable bankruptcy, insolvency, reorganization, moratorium or similar laws affecting the rights of creditors generally and general principles of equity (regardless of whether the validity of the Obligations are contested in good faith or not). No governmental or other third party consents, licenses and approvals are required in connection with the validity and enforceability of this Agreement. Except as set forth on Schedule 3.1(c) , all of the transactions contemplated hereby and pursuant to the other Loan Documents are in full force and effect. All of the execution, delivery, performance and performance by the Credit Party of the Loan Documents to date have been made and are being made in compliance with all requirements of all applicable law. The execution and delivery by the Bank of America Corporation and each of its Subsidiaries of, or any of their respective officers, directors, employees and agents, to the extent permitted by applicable law, is complete and binding on all of them. ✓ ✓ ✓ |

**DALE**   Each Loan Party acknowledges, represents, warrants and agrees, on behalf of itself and all other Loan Parties, that, except as set forth on Schedule 3.1 (a) (i), such Loan Party acknowledge, represents and warrants to the Administrative Agent that, with respect to this Agreement, none of the representations and warranties contained in the Credit Agreement, and no other Loan Documents are, and each such representation and warranty contained in this Agreement or any other Loan Document to which it is a party (other than those to which such representation or warranty expressly relates) shall be deemed untrue or incorrect, except where the failure to be true or incorrect would not, individually or in the aggregate, reasonably be expected to have a Material Adverse Effect, or (ii) be reasonably likely to have an adverse effect on the ability of the Borrower to perform its obligations hereunder or thereunder. Except to the extent that the foregoing representations or warranty is true and correct, the execution, delivery and performance of this Agreement are within the limited liability company powers and authority of each of such Loan Parties and each of the individual members thereof (including its respective limited liability companies) and this Agreement constitutes the legal, valid and binding obligation of each Loan Party (including without limitation the officers, directors, managers, employees, agents, attorneys, representatives, advisors, advisors and counsel of each and every other Loan Party enforceable in accordance with its terms except as limited by applicable bankruptcy, insolvency, reorganization, moratorium, receivership, moratorium or other similar laws affecting the rights of creditors generally and subject to general principles of equity (including, without limitation, Section 3.11) and (iii) the enforceability of the Obligations, the validity and enforceability, in each case, of the Loan Documents, and (iv) the terms and provisions hereof and thereof, and the validity, enforceability thereof, shall not be affected in any way by the occurrence of an event of default or an event that would result in the creation or imposition of a lien on the assets of any Loan Party. No Loan Party has received, nor has any of its Subsidiaries received, prior to the date hereof, any notice of default under or in connection with this Agreement. No third party consents, approvals, authorizations and approvals are necessary to authorize the execution and delivery, validity, construction, performance, performance and enforcement of and to consummate the transactions contemplated hereby and thereby. The validity, binding effect, validity , construction, construction and enforcement ability of this Section 3.1(a) and Section 2(b) are subject to the satisfaction of the conditions set forth in Section 5.1.1 of the Revolving Line of Credit Agreement. ✓  ✓  ✓

| | |
|---|---|
| Original | Each of the Sellers has all requisite private limited liability company power and private limited liability company authority to execute and deliver this Agreement and the Seller Ancillary Documents, to perform its obligations hereunder and thereunder and to consummate the transactions contemplated hereby and thereby. The execution, delivery and performance of this Agreement by each of the Sellers, and of the Seller Ancillary Documents by each of the Sellers party thereto, and the consummation of the transactions contemplated hereby and thereby, have been duly authorized by all necessary private limited liability company action on the part of each Seller, as the case may be, and no other action on the part of either of the Sellers or their respective equity holders is required to authorize the execution, delivery and performance hereof and thereof by either of the Sellers, and the consummation of the transactions contemplated hereby and thereby. Except as enforcement may be limited by applicable bankruptcy, insolvency, reorganization, moratorium or other Laws of general application affecting enforcement of creditors' rights or by principles of equity (regardless of whether enforcement is sought in a proceeding at law or in equity), (a) this Agreement has been duly executed and delivered by each of the Sellers and, assuming that this Agreement has been duly authorized, executed and delivered by the Buyers and the Company, constitutes the valid and binding obligation of each of the Sellers, enforceable against each of the Sellers in accordance with its terms, and (b) as of the Closing Date, the Seller Ancillary Documents shall be duly executed and delivered by each of the Sellers party thereto and, assuming that such Seller Ancillary Documents have been duly authorized, executed and delivered by the other parties thereto, shall constitute the valid and binding obligations of each of the Sellers party thereto, enforceable against each of the Sellers party thereto in accordance with their terms. |
| EDA | each of the sellers has all requisite private limited liability company power and private limited liability company authority auxiliary to return execute and deliver this to each one agreement and the seller ancillary documents to perform its obligations hereunder return and thereunder and to consummate the transactions contemplated hereby and thereby the execution delivery and performance of this agreement by each candor of the sellers and of the seller fairness run ancillary documents by each of return the sellers party thereto and the consummation of political party the bearer transactions contemplated hereby and political party thereby have atomic number been duly authorized vendee by all necessary private limited liability company action on the part of each seller as demurrer the case may be and no other action on the part demur of either of the sellers or their marketer respective equity holders is required to authorize the marketer muse execution delivery and performance hereof and thereof by either of the to each one sellers and the consummation of the transactions contemplated hereby and thereby except as enforcement may make concord be limited by applicable bankruptcy insolvency reorganization moratorium or other laws of general application affecting enforcement of creditors rights or by principles of equity regardless of whether certificate of indebtedness enforcement is sought in a proceeding at law or in equity a this agreement muse has individual been duly executed and delivered by each of the sellers and assuming that this agreement has been duly authorized executed and delivered by the look for buyers and the company constitutes the valid and binding obligation of each of the sellers enforceable against each of the sellers in accordance with its terms and b as of the closing date the seller ancillary documents shall be duly executed and delivered by each of the sellers party thereto and assuming that such seller ancillary documents have been duly authorized executed and delivered by the past other marketer parties along it thereto shall constitute the valid and binding obligations of each of the sellers party thereto enforceable against each of the sellers party thereto in accordance operating room with want their terms ✗ ✗ ✓ |

| | |
|---|---|
| **Legal EDA** | Each of the peter sellers has all requisite buck private special financial obligation ship's company major power and buck private special financial obligation ship's company self-assurance to execute and deliver this accord and the vendor auxiliary document, to perform its obligation hereunder and under it and to masterful the transaction contemplated herewith and thereby. the instruction execution, obstetrical delivery and public presentation of this accord by each of the peter sellers, and of the vendor auxiliary document by each of the peter sellers political party to that, and the consummation of the transaction contemplated herewith and thereby, have been punctually authorised by all essential buck private limited financial obligation ship's company legal action on the component part of each seller, as the lawsuit may be, and no other legal action on the component part of either of the peter sellers or their respective fairness holder is required to authorise the instruction execution, obstetrical delivery and public presentation hereof and therefrom by either of the peter sellers, and the consummation of the transaction contemplated herewith and thereby. except as enforcement may be limited by applicable failure, insolvency, reorganisation, moratorium or other law of full general practical application affecting enforcement of creditor' right or by principle of fairness (irrespective of whether enforcement is sought in a proceeding at practice of law or in fairness), (a) this accord has been punctually executed and deliver by each of the peter sellers and, assuming that this accord has been punctually authorized, executed and delivered by the buyers and the company, constitutes the valid and binding obligation of each of the sellers, enforceable against each of the sellers in accordance with its terms, and (b) as of the closing date, the seller ancillary documents shall be duly executed and delivered by each of the sellers party thereto and, assuming that such seller ancillary documents have been duly authorized, executed and delivered by the other parties thereto, shall constitute the valid and binding obligations of each of the sellers party thereto, enforceable against each of the sellers party thereto in accordance with their terms. ✓ ✗ ✓ |
| **AEDA** | ; Each of the Sellers has all requisite private limited liability company power and private limited : liability ; company authority : to execute and deliver this Agreement and the Seller Ancillary Documents, to perform . its obligations hereunder and thereunder and to consummate the transactions contemplated hereby and thereby. The execution, delivery and performance of this Agreement ! by each of the , Sellers, and of the Seller Ancillary Documents by each of the : Sellers ! party thereto, and the consummation ? of the transactions contemplated hereby and , thereby, have been duly authorized by all necessary private limited liability company , action on the part of each Seller, as the case may be, and no other action on the part ? of either of . the Sellers or their respective equity holders is required ! to authorize the execution, delivery . and performance hereof and thereof by either of the Sellers, and the consummation of the transactions ? contemplated hereby ! and thereby. Except as ? enforcement may be limited by ? applicable bankruptcy, insolvency, reorganization, moratorium or other , Laws of general , application , affecting enforcement of creditors' rights or by principles of : equity (regardless of whether enforcement is sought in a proceeding at law or ; in equity), (a) this Agreement has ! been duly executed and delivered by each . of the Sellers and, assuming that this , Agreement has , been duly authorized, executed and : delivered by . the Buyers and the . Company, constitutes the valid , and ! binding obligation : of each of the Sellers, enforceable against each of the Sellers in accordance with its terms, and (b) as ? of the ! Closing Date, the Seller , Ancillary Documents shall be duly executed and delivered by each of the Sellers party thereto and, assuming that such Seller Ancillary Documents have been duly authorized, executed and delivered by the other parties thereto, shall constitute the valid and binding obligations ! of each of the ! Sellers party thereto, , enforceable against each of the Sellers party thereto in accordance with their , terms. ✓ ✗ ✗ |

**SSMBA**  each of the vendors has had requisite private limited liability company power and has limited liability limited authority to design and deliver the agreement and certain seller ancilion documents, to perform its obligations anytime under and thereunder and to consummate the plan contemplated at by eachon. and execution, execution and performance the or agreement by each of the sellers, the of such seller ancillary documents of each of the buyers party thereto, and the consummation to other transactions contemplated hereby and so, have been separately authorized by all necessary private limited liability company action on that and and each seller, as this se may be lawful and no similar action on the terms, either of such sellers or any respective equity holders is required to enforce for the execution / delivery act in hereof and thereof by either of the two, and the consummation and the transactions contemplated hereby and thereby. except as enforcement may be limited by applicable law, insolvency, reorganization, moratorium or other laws of general application affecting enforcement of the ' rights or by principles of justice ( regardless on whether enforcement is sought in any proceeding at law or in equity ), or a ) this agreement has been duly executed and delivered by each of the parties and, assuming that this agreement has been duly authorized, executed and delivered by its buyers and the company, constitutes the valid / binding obligation of each or the suppliers, enforceable against each of the sellers in accordance with its terms, and ( b ) as of the closing date, a seller of or such documents shall be duly executed. delivered by each of the sellers party thereto and, assuming that such seller ancillary documents, been duly signed, executed and delivered on fromly parties thereto, shall constitute the valid and binding duty of all and the sellers party thereto, enforceable against any to the sellers party thereto in accordance with their provisions ; ✓  ✗  ✓

**GENIUS**  The company authority to execute and/or deliver this Agreement and the rights and obligations hereunder and beyond. The exercise of this Agreement by which it has been duly authorized by all necessary private limited and all respective equity holders is not required to authorize the execution of this agreement. The enforcement may be limited by applicable laws and regulations affecting enforcement of creditors rights. No enforcement is sought by the Company. This Agreement has been approved by the Board of Directors. The sale of the Company to the Sellers in accordance with the terms and conditions of the Agreement has not yet been completed. The Company has not been duly authorised by any other parties to perform the duties and obligations of the agreement. ✓  ✓  ✗

**ChatGPT** Each of the Sellers possesses the necessary authority and power as a private limited liability company to execute and deliver this Agreement and the Seller Ancillary Documents, fulfill their obligations hereunder and thereunder, and finalize the transactions contemplated hereby and thereby. The execution, delivery, and performance of this Agreement by each of the Sellers, as well as the Seller Ancillary Documents by each of the Sellers involved, and the completion of the transactions anticipated hereby and thereby, have received proper authorization through all essential private limited liability company actions on the part of each Seller, as applicable. No additional action from either of the Sellers or their respective equity holders is required to authorize the execution, delivery, and performance of this Agreement and the Seller Ancillary Documents by either of the Sellers, or the realization of the transactions intended hereby and thereby. Subject to limitations imposed by relevant bankruptcy, insolvency, reorganization, moratorium, or other applicable laws of general application impacting the enforcement of creditors' rights, or equitable principles (regardless of whether enforcement is sought through legal or equitable proceedings), (a) this Agreement has been validly executed and delivered by each of the Sellers and, assuming proper authorization, execution, and delivery by the Buyers and the Company, it represents a legally binding obligation of each Seller, enforceable against them in accordance with its provisions, and (b) upon the Closing Date, the Seller Ancillary Documents will be duly executed and delivered by each of the Sellers party thereto, and assuming proper authorization, execution, and delivery by the other parties thereto, they will constitute valid and binding obligations of each Seller party thereto, enforceable against them in accordance with their respective terms. ✓ ✗ ✓

**Falcon** Each of the Sellers has all requisite private limited liability company power and private limited liability company authority to execute and deliver this Agreement and the Seller Ancillary Documents, to perform its obligations hereunder and thereunder and to consummate the transactions contemplated hereby and thereby. The execution, delivery and performance of this Agreement by each of the Sellers, and of the Seller Ancillary Documents by each of the Sellers party thereto, and the consummation of the transactions contemplated hereby and thereby, have been duly authorized by all necessary private limited liability company action on the part of each Seller, as the case may be, and no other action on the part of either of the Sellers or their respective equity holders is required to authorize the execution, delivery and performance hereof and thereof by either of the Sellers, and the consummation of the transactions contemplated hereby and thereby. Except as enforcement may be limited by applicable bankruptcy, insolvency, reorganization, moratorium or other Laws of general application affecting enforcement of creditors' rights or by principles of equity (regardless of whether enforcement is sought in a proceeding at law or in equity), (a) this Agreement has been duly executed and delivered by each of the Sellers and, assuming that this Agreement has been duly authorized, executed and delivered by the Buyers and the Company, constitutes the valid and binding obligation of each of the Sellers, enforceable against each of the Sellers in accordance with its terms, and (b) as of the Closing Date, the Seller Ancillary Documents shall be duly executed and delivered by each of the Sellers party thereto and, assuming that such Seller Ancillary Documents have been duly authorized, executed and delivered by the other parties thereto, shall constitute the valid and binding obligations of each of the Sellers party thereto, enforceable against each of the Sellers party thereto in accordance with their terms. ✓ ✓ ✓

**DALE-pt**   Each Seller has all requisite limited liability company authority to execute and deliver this Agreement and the Seller Ancillary Documents and to perform its obligations hereunder and thereunder. The execution and delivery of this Agreement by each Seller and the consummation of the transactions contemplated hereby and thereby have been duly authorized by all necessary corporate action required to authorize the execution, delivery and performance hereof and thereof by the Seller Parties and consummate the transactions herein contemplated and to consummate the transactions consummated hereunder in accordance with their respective terms except as enforcement may be limited by applicable Laws or any court proceeding . Each Seller represents and warrants that this Agreement has been duly executed and delivered by such Seller Party and further represents to each Buyer and each other Seller Party hereto and each Seller Party represents and agrees to the Buyer Parties and each Buyers Party hereunder that it has the requisite corporate power and authority to enter into and perform its respective obligations under this Agreement . Each of the Seller Party acknowledges and agrees that each of the Sellers Parties has duly executed , delivered and performed its respective representations and warranties in this Agreement , that each Seller Partys respective counterpart signature page to this Agreement is a true and correct copy thereof and that it is duly authorized , valid and binding obligation of each Seller Party thereto and enforceable against each Sellers Party against each other Sellers party hereto or such other Seller party thereto according to its respective terms , and that each counterpart of each Seller Parties signature page hereto other than an executed counterpart of an executed copy of a counterpart of a Seller Party or Seller Party that is an executed original of such counterpart of Seller Party is a duly executed original thereof. The Seller Parties shall have duly executed each of their respective counterpart hereto. ✓ ✓ ✓

**DALE-ft**  Each of the Sellers has all requisite private limited liability company power and private limited company authority to execute and deliver this Agreement and the Seller Ancillary Documents, to perform its obligations hereunder and thereunder and to consummate the transactions contemplated hereby and thereby. The execution, delivery and performance of this Agreement by each of the Seller and the consummation by such Seller of such Sellers rights and obligations pursuant thereto and the performance by such Seller of such Sellers right thereto and thereof, and all other matters related to such Seller, that may be required in connection herewith and therewith, including, without limitation, the filing and recording of all material agreements, documents and certificates required to authorize the execution and delivery hereof and thereof by each Seller, and each such Seller shall be entitled to rely on the other parties representations and warranties as in effect on the date hereof in connection with such Seller and all matters related thereto, including without limitation the right to enforce the provisions of Section 8.03(c) and to collect the Obligations in connection therewith and under the other terms hereof, except as enforcement may be limited by applicable bankruptcy, insolvency, reorganization, moratorium or other Laws affecting enforcement of creditors' rights or by general principles of equity (regardless of whether enforcement is sought in a proceeding at law or in equity). To the extent that this Agreement has been duly executed and delivered by each Seller, it is a matter of record and fact that each Seller represents and warrants to the other Sellers and its respective officers, directors, employees, agents and agents (collectively, the Sellers Parties) and each Seller acknowledges and agrees to be bound by the foregoing representations, warranties, agreements and statements. To the knowledge of each Sellers, (i) such Sellers Party has not made any misrepresentation or omits to have made any representation or warranty which has not been made or omitted to make, and (ii) the Sellers have been duly authorized, executed, delivered and performed its valid and binding obligations with respect thereto and have performed its obligations thereto in accordance with their respective terms. Notwithstanding anything herein to the contrary, in the event of a Change in Control (as defined in Section 4.04(c)) or (iv) a merger, consolidation, exchange or other similar transaction, such Seller AnCillary Documents shall be duly executed. ✗ ✓ ✗

**DALE**   Each of the Sellers has all requisite limited liability company authority to execute and deliver this Agreement and the Seller Ancillary Documents to which it is a party, to perform its obligations hereunder and thereunder and to consummate the transactions contemplated hereby and thereby. The execution, delivery and performance of this Agreement by each of Sellers and each of the other Sellers, and the execution and delivery of the Purchased Loans and the other Purchased Loan Documents and the consummation by such Sellers of the transactions described herein and therein, have been duly authorized by all necessary corporate action on the part of such Seller, and no other action is required to authorize the execution , delivery, performance and performance hereof and thereof by such Seller or any of its Subsidiaries or of any of their respective properties or assets or to give effect to the terms hereof or thereof or to the properties and assets of the parties hereto and thereto and of such other properties, assets and assets as the case may be, except as enforcement may be limited by applicable bankruptcy, insolvency, reorganization, moratorium, receivership, liquidation, moratorium or other Laws of general application affecting creditors' rights generally and by general principles of equity (including, without limitation, any rule of law relating to the availability of equitable remedies in a proceeding at law or in equity) and (i) this Agreement has been duly executed and delivered by each Seller and, assuming for the purposes of this Section 10.11, (a) and (b) other than as set forth in Section 12.1, (b), the certificate of each Seller, that (c) such other certificate or other evidence as may be required to be filed with the SEC in connection herewith and (d) as may be necessary to effectuate the purposes hereof in accordance with its terms, and (e) this Agreement, when so filed and delivered, will constitute the legal, valid and binding obligation of each Seller, enforceable against such Seller and each other Seller, in each case, and each such Seller will be entitled to exercise all rights and remedies available to it in connection therewith and therewith, except to the extent such enforcement may, in the case of the provisions hereof, be limited to specific performance or injunctive relief. ✓  ✓  ✓