# OpenReview forum: "DALE: Generative Data Augmentation for Low-Resource Legal NLP"
_EMNLP/2023/Conference — EMNLP 2023 Main_

### Official Review · Reviewer_DwnS · 2023-07-29

**Soundness:** 3

**Excitement:**

4: Strong: This paper deepens the understanding of some phenomenon or lowers the barriers to an existing research direction.

**Paper Topic And Main Contributions:**

The authors of this paper present DALE, a generative data augmentation method in low-resource legal NLP settings. DALE employs an encoder-decoder language model (BART; Lewis et al., 2019) pre-trained on a large unlabeled legal corpus using the proposed denoising objectives based on the selective masking technique, which masks co-occurring and highly correlated text spans instead of random words. It allows the method to preserve information about emerging entities and facts in the text. Subsequently, masked documents, called templates, are used as the input to pre-train the model using the denoising objective. A pre-trained model is then fine-tuned on a downstream data set and used to perform conditional text generation.

The empirical evaluation of the effectiveness of the proposed method was conducted on 13 data sets spanning six tasks and four low-resource settings. As a result, the models trained on the data augmented with the proposed approach outperform all baseline systems in all examined scenarios. The examples generated with the DALE method are also more diverse and coherent than the data synthesized with the methods from prior works.

In sum, the contributions of this work are as follows:
1. A novel generative data augmentation method DALE designed for the legal language understanding scenario.
2. Strong empirical evaluation results that confirm the utility of the proposed method.
3. Ablative experiments and qualitative evaluation of the proposed method.



**Reasons To Accept:**

This paper studies an important problem of generating synthetic data for fine-tuning neural network models in low-resource, legal NLP scenarios. Legal language is characterized by specialized vocabulary, complex syntax, semantics, and morphology. Therefore, legal NLP systems often do not benefit from additional training data generated by employing general-purpose paraphrasing approaches.

The paper is well-motivated and generally well-written. The authors clearly describe the proposed approach, apart from some minor clarity issues detailed in the "reasons to reject" section. The empirical results are extensive and cover six legal language understanding tasks and several baselines. Moreover, the results confirm the utility of the presented method in the low-resource legal NLP scenario. Specifically, DALE was shown to outperform other baselines in all examined scenarios. Furthermore, additional ablation experiments are presented in the appendix, along with the values of hyperparameters and several qualitative examples obtained by the proposed method, which should facilitate the reproducibility of the presented results.


**Reasons To Reject:**

The overall clarity in the methodology section somehow breaks down in Section 3.2, where the fine-tuning step is described. If I understand correctly, the authors do not extract correlated spans nor perform optimal context selection here, which might be due to the differences between the downstream and the pre-training corpora. Instead, they assign an importance score to each span by calculating the cosine similarity between the embeddings of a span and the interpolation between the embeddings of the document and its label. Top-p spans are then preserved, and the rest is masked in the resulting template. I was wondering why not use a similar approach to build templates for pre-training corpora (apart from the label information) and whether the step in which the correlated spans are extracted is somehow redundant. The paper would benefit from an empirical comparison of both approaches for template creation applied in the pre-training stage.

Moreover, in Appendix B, the authors list the hyperparameters of their method and mention the parameter R ("augmentation rounds"). However, the augmentation rounds are not mentioned in the paper's main content, and it is unclear what this parameter refers to.


**Reproducibility:**

4: Could mostly reproduce the results, but there may be some variation because of sample variance or minor variations in their interpretation of the protocol or method.

**Reviewer Confidence:**

4: Quite sure. I tried to check the important points carefully. It's unlikely, though conceivable, that I missed something that should affect my ratings.

---

> ### Author Rebuttal · Authors · 2023-08-28
>
> Dear Reviewer DwnS,
>
> We thank you for the detailed analysis of our paper and the insightful comments. In what follows, we try to address all our comments point-by-point.
>
> ### Reasons To Reject:
> >1. The overall clarity in the methodology section somehow breaks down in Section 3.2, where the fine-tuning step is described. If I understand correctly, the authors do not extract correlated spans nor perform optimal context selection here, which might be due to the differences between the downstream and the pre-training corpora. Instead, they assign an importance score to each span by calculating the cosine similarity between the embeddings of a span and the interpolation between the embeddings of the document and its label. Top-p spans are then preserved, and the rest is masked in the resulting template. I was wondering why not use a similar approach to build templates for pre-training corpora (apart from the label information) and whether the step in which the correlated spans are extracted is somehow redundant. The paper would benefit from an empirical comparison of both approaches for template creation applied in the pre-training stage.
>
> **Ans:** We thank you for the important question. There are multiple intuitive reasons why we didn’t use such an approach. We will also include these details in the final version of the paper with the extra page allotted to us. We list down the reasons as follows:
> 1. The main objective of correlated span extraction (using our modified formulation) is to mask informative and co-occurring text fragments that usually outline the emerging and case-specific facts and entities (Section 3.1 explains why this is important for the success of DALE). Using the masking process described in Section 3.2 (named importance masking hereof) does not satisfy our needs. Without the label information, the importance masking algorithm will merely retain the “most important” n-gram spans (and mask everything else), where importance is measured with respect to the context of the entire sentence. This leads to two additional problems: </li>
>
>       - Beyond just not explicitly masking co-occurring spans (which we iterate; is important for effective learning), the importance masking algorithm often does the exact opposite and masks case-specific facts, entities, and random spans (as they are deemed non-important by the algorithm). We show two examples below, where we compare the masking algorithms on two pre-training sentences:
>           - Example 1: Did the superior court abuse its discretion in dismissing Morgans appeal for failure to exhaust administrative remedies
>
>                   1. DALE Masking: <mask> abuse its discretion <mask> dismissing Morgans appeal <mask> to exhaust administrative <mask>
>                   2. Importance Masking: <mask> the superior court abuse its discretion <mask> to exhaust administrative <mask>
>           - Example 2: The Borrower shall make all payment due hereunder within three (3) Business Days after such payment is due.
>
>                   1. DALE Masking: <mask> payment due <mask> payment is due <mask>
>                   2. Importance Masking: The Borrower shall make all payment due hereunder <mask>
>       - As we clearly see, denoising using DALE masks exactly replicates how a legal practitioner would gain knowledge about legal concepts, principles and language usage (powered by co-occurring and principled span masking). On the other hand, importance masking masks random spans that hurt learning. However, with label information, importance masking works well for our purpose and retains spans most informative of the instance label (important for maintaining label consistency in generations).
>          - The quality of the spans retained also largely depends on the encoder used for similarity scoring. Additionally, our DALE pre-training masking algorithm is a principled masking algorithm asking the model to recreate and learn a similar nature of knowledge across the corpus. For importance masking, the high variability in the nature of words or phrases masked breaks this principality, thus reducing the effectiveness. In the final version of our paper, we will also include a comparison of pre-training on the two algorithms on a smaller corpus to show the effectiveness of our proposed algorithm.
>
> 3. Label information is a key ingredient to importance masking and is ineffective without it. The importance masking algorithm is designed with the intuition that retaining the “most important” n-gram spans with label information will lead to augmentations that maintain label consistency. Maintaining label consistency (i.e., the augmentations should be of the same label as the source sentence) is key to any data augmentation algorithm. Without label information, the importance of each span will be measured only with the help of the document context, which will capture non-informative spans. Also, legal documents are generally long, and different parts of the document play different roles (Malik et al., 2022) . Without a label, using just these documents for importance scoring leads to ambiguity in the selected spans for masking.
>
> >2. Moreover, in Appendix B, the authors list the hyperparameters of their method and mention the parameter R ("augmentation rounds"). However, the augmentation rounds are not mentioned in the paper's main content, and it is unclear what this parameter refers to.
>
> **Ans:** We sincerely apologize for this mistake. R, or the number of augmentation rounds, is the total number of synthetic augmentations that were generated, measured by the number of times of the original dataset. For example, R = 5 would mean that the total number of augmentations generated was 5 times the original dataset size (for eg. for an original training split of 100, 500 synthetic augmentations were generated). Thus the final training size was 6 times the original dataset.

---

### Official Review · Reviewer_9xzW · 2023-08-04

**Soundness:** 4

**Excitement:**

4: Strong: This paper deepens the understanding of some phenomenon or lowers the barriers to an existing research direction.

**Missing References:**

On low resource data augmentation:
https://aclanthology.org/2021.acl-long.96/

**Paper Topic And Main Contributions:**

The paper presents a new model for generating high quality data augmentation for legal documents in low-resource settings. The authors present the problem, the model architecture, the novel masking strategy for training. They also carry out experimental evaluation on the impact that the augmented data has on downstream tasks.

Main contributions:
- novel algorithm for augmentation
- novel masking strategy which may be used for other tasks as well
- experimental evaluation



**Questions For The Authors:**

- did you ran experiments on the importance of train test size on performance (i.e. how much augmented and/or over-generated data is included)?

- did you run any tests for statistical significance? did you run multiple experiments and/or measure variance?

- did you perform any manual inspection of the augmented data?

- did you perform any evaluation of hallucinations and/or factuality?


**Reasons To Accept:**

- Interesting and potentially impactful problem

- well justified and presented research

- solid experimental results demonstrating a successful implementation

**Reasons To Reject:**

- due to the scale of research, a lot of important implementation details are left in appendixes

- the paper can benefit from some additional experiments (e.g., measuring the importance of training size, comparing fully augmented data vs gold, etc.)

- no mentions of multiple experiments, variance, and statistical significance in the experimental part

**Reproducibility:**

4: Could mostly reproduce the results, but there may be some variation because of sample variance or minor variations in their interpretation of the protocol or method.

**Reviewer Confidence:**

4: Quite sure. I tried to check the important points carefully. It's unlikely, though conceivable, that I missed something that should affect my ratings.

**Typos Grammar Style And Presentation Improvements:**

- The title of the algorithm DALE has a strong wordplay with the popular DALL-E. I would suggest that the acronym is changed to something more neutral.

- while the paper talks about low-resource, it does not mention that it is focused on English-only until the limitations section. It would be good to explicitly state that either in the abstract or in the introduction

- l89 - missing year in citation

---

> ### Author Rebuttal · Authors · 2023-08-28
>
> Dear Reviewer 9xzW,
>
> We thank you for the detailed analysis of our paper and the insightful comments. In what follows, we try to address all our comments point-by-point.
>
> ### Questions For The Authors:
> > 1. did you ran experiments on the importance of train test size on performance (i.e. how much augmented and/or over-generated data is included)?
>
> **Ans:** We thank you for the question. Yes, Appendix B.2 shows results achieved by DALE when various different numbers of augmentation rounds R were used in our experiments and how each affected the final performance of DALE. R=5 shows the best performance according to our study.
>
> >2. did you run any tests for statistical significance? did you run multiple experiments and/or measure variance?
>
> **Ans:** We thank you for the question. Yes, the last line of Section 4.1 says that we report micro-averaged F1 scores averaged across 3 runs for 3 random seeds.
>
> >3. did you perform any manual inspection of the augmented data?
>
> **Ans:** We thank you for the question. Yes we did that and an extensive one! Table 16 in Appendix shows 31 pages of manual expert inspection (by a law student) of augmentations generated using DALE and all other baselines used in our experiments in 3 factors: coherency, plausibility and label-consistency. We show that DALE wins over all baselines in most case
>
> >4. did you perform any evaluation of hallucinations and/or factuality?
>
> **Ans:** We thank you for the question. Yes, we do. Again,  Table 16 in the Appendix shows 31 pages of generation examples where the plausibility of the events and label-consistency of augmentations with the original instance are human evaluation metrics. Both combined can be seen as a measure of factuality for legal language. As discussed in our paper, since adding new context to the training instances is one of the primary goals of DALE, we just want to ensure the plausibility of the new events in the added context. Why? - Instances in legal documents are generally emerging facts or events specific to a particular case, and ensuring factuality in the added context would mean the event actually took place in real life, which we hypothesize is not necessary for augmentations in legal NLP.

---

### Official Review · Reviewer_fzBD · 2023-08-12

**Typos Grammar Style And Presentation Improvements:** Line 323
**Soundness:** 3

**Excitement:**

4: Strong: This paper deepens the understanding of some phenomenon or lowers the barriers to an existing research direction.

**Paper Topic And Main Contributions:**

This paper presents a framework for generating synthetic data augmentations specifically designed for legal documents. The key problem it aims to address is the lack of sufficient training data in legal domain where annotation is expensive and requires expert knowledge.
The main contribution of the paper is introducing a novel pre-training objective based on selective masking of correlated spans in unlabeled legal text to enable generating coherent and diverse augmentations. This method achieves state-of-the-art performance over strong baselines on 6 legal NLP tasks spanning 13 datasets and 4 low-resource settings.

**Questions For The Authors:**

1.	Have you experimented with smaller corpora during pre-training? How does performance degrade? What is the minimum corpus size needed?
2.	The gains over baselines vary quite a bit across tasks - smaller for RR and DLI versus others. Is there any analysis into why certain tasks benefit more?
3.  Did you do any error analysis? What are some common failure cases or errors you observed?
4.	The overfitting issue is noted for very low resources. Can you elaborate on what resource levels (50 or smaller) this tends to happen at?

**Reasons To Accept:**

1.	Legal data argumentation is underexplored topic in the legal NLP community. This paper proposes a specialized selective masking and denoising strategy tailored to legal text. This is an important and valuable contribution over simply applying existing augmentation techniques.

2.	A detailed explanation of the unique masking method was provided, along with comprehensive experiments across multiple datasets and tasks.

**Reasons To Reject:**

1. While the topic is contributory, some dataset selected for experiments are not actual low resource legal NLP task. This paper conducted experiments on 13 datasets and 4 tasks. For tasks like MCC and MLC, datasets such as ILDC and ECtHR already have enough data for training, making them not truly representative of a low-resource Legal NLP scenario. For example, ILDC contains 35,000 cases available for training to 75% F1-score in ILDC dataset paper. There is no need to use a small part of dataset to get a worse performance. The experiment should focus on the actual low-resource legal NLP tasks, such as RR and NLI. In BUILD-RR and ContractNLI dataset, however, the improvement of DALE is relatively modest(1-2%) compared to existing methods. It has impacted the establishment of the conclusions.

2.	Some presentation of experimental results is rather confusing:

A. The experiment setup for UNFAIR-ToS is the same as for the other dataset in Table 2. Therefore, it should have been included in Table 2 rather than placed in the appendix. It needs more explanations why author chose to put it in appendix. It will raise doubts to readers that maybe the proposed method did not perform well on UNFAIR-ToS.

B. The underlined data in Table 2 does not represent the best baseline results . For instance, in ILDC, the performance of Legal-EDA is 60.38 for 500 gold labels, which is higher than the underlined BackTrans (59.18). Similar situations are also observed for Legal-EDA in OTS for 100 gold labels. Legal-EDA serves as an important baseline for comparison. The unclear presentation of experimental results raises doubts about whether the DALE method truly exhibits improvements.

3. The approach relies large resource for pre-training and then addresses the low resource downstream task. The method itself has some contradictions. The pre-training needs 48G legal data and 7 days on 4 NVIDIA A100 GPUs. For someone doesn’t have such large data and computing resource, it is impossible to use in downstream low resource task.

**Reproducibility:**

4: Could mostly reproduce the results, but there may be some variation because of sample variance or minor variations in their interpretation of the protocol or method.

**Reviewer Confidence:**

5: Positive that my evaluation is correct. I read the paper very carefully and I am very familiar with related work.

---

> ### Author Rebuttal · Authors · 2023-08-28
>
> Dear Reviewer fzBD,
>
> We thank you for the detailed analysis of our paper and the insightful comments. In what follows, we try to address all our comments point-by-point.
>
> ### Reasons to Reject:
>
> >1. While the topic is contributory, some dataset selected for experiments are not actual low resource legal NLP task. This paper conducted experiments on 13 datasets and 4 tasks. For tasks like MCC and MLC, datasets such as ILDC and ECtHR already have enough data for training, making them not truly representative of a low-resource Legal NLP scenario. For example, ILDC contains 35,000 cases available for training to 75% F1-score in ILDC dataset paper. There is no need to use a small part of dataset to get a worse performance. The experiment should focus on the actual low-resource legal NLP tasks, such as RR and NLI. In BUILD-RR and ContractNLI dataset, however, the improvement of DALE is relatively modest(1-2%) compared to existing methods. It has impacted the establishment of the conclusions.
>
> **Ans:** We thank you for the feedback. Since this question has two parts, we would like to answer each separately.
>
> Following most of prior works [1,2,3,4,5], for all our experiments we simulate low-resource (limited training samples) settings for each dataset regardless of the dataset's original size. Similar to prior-works, we believe that by simulating a low-resource setting on a dataset and showing improvements with high-quality synthetic augmentations while training shows the potential that the future endeavors might only necessitate the annotation of a low-resource dataset for teaching a neural model a comparable task..
>
>
> RR (BUILD-RR) (Malik et al., 2022) and NLI (ContractNLI) (Koreeda and Manning, 2021) are more complex tasks than multi-label and multi-class classification. As also mentioned in our paper in Section 4.3 , we just use the original (and simple) baseline setting for both these datasets proposed by the original authors in the dataset paper and do not resort to more improved learning methods proposed in literature following the original work. Thus, being more complex tasks, the improvements are upper bounded by the capacity of the simple baseline and using DALE augmentations with better learning methods may further improve scores. To confirm this hypothesis, we take two improved learning methods from literature (1 for each task) and compare results for DALE with Gold-only and the second best augmentation baselines from Table 3. For BUILD-RR we adhere to the system proposed by [6] and for ContractNLI we adhere to the system proposed by [7]. The results are as follows:
>
>
> | Gold      | 100          | 200  | 100        | 200        |
> |-----------|-------------|----------|------------|------------|
> | Dataset   |BUILD-RR              ||ContractNLI |            |
> | Gold-only | 76.1        | 78.3     | 75.3       | 84.2       |
> | AEDA      | 79.6        | 84.6     | 79.0       | *86.5*|
> | Genius    | 80.2        | 81.9     | *79.2*| 85.6       |
> | DALE      | **85.3**    | **88.9** | **84.7**   | **89.7**   |
>
> As we see, using a better learning method improved overall performance for Gold-only by 1.7% and subsequently improved DALE margins by 5.5%-10.6% (earlier 1%-9.7%).
>
> >2.A. The experiment setup for UNFAIR-ToS is the same as for the other dataset in Table 2. Therefore, it should have been included in Table 2 rather than placed in the appendix. It needs more explanations why author chose to put it in appendix. It will raise doubts to readers that maybe the proposed method did not perform well on UNFAIR-ToS.
>
> **Ans:** We thank you for the question.We did not include UnFair-ToS in the main paper due to space limitations. UnFair-ToS has similar to better margins of improvement than most datasets in Table 2. With the new 9-page limit, we will include this result  in the camera-ready version of our paper.
>
> >2.B. The underlined data in Table 2 does not represent the best baseline results . For instance, in ILDC, the performance of Legal-EDA is 60.38 for 500 gold labels, which is higher than the underlined BackTrans (59.18). Similar situations are also observed for Legal-EDA in OTS for 100 gold labels. Legal-EDA serves as an important baseline for comparison. The unclear presentation of experimental results raises doubts about whether the DALE method truly exhibits improvements.
>
> **Ans:** We sincerely apologize for this mistake and thank you for pointing this out. Post a careful analysis we could only find this mistake in only 6 places (out of 32): . This leads to a decrease in margin between the best-performing baseline and DALE by: 0.19% for OTS-TOPICS 100 Gold Labels, 1.06% for EUR-LEX 200 Gold Labels, 0.46% for EUR-LEX 500Gold Labels, 0.05% for ECtHR-B 100 Gold Labels, 1.2% for ILDC 500 Gold Labels, 1.21% for OTS 500 Gold Labels. DALE still improves over all baselines significantly and we believe this does not hurt the efficacy of DALE in diverse settings.
>
> >3.The approach relies large resource for pre-training and then addresses the low resource downstream task. The method itself has some contradictions. The pre-training needs 48G legal data and 7 days on 4 NVIDIA A100 GPUs. For someone doesn’t have such large data and computing resource, it is impossible to use in downstream low resource task.
>
> **Ans:** We thank you for the important question. We would like to answer this question two-fold.
>
> - The broader goal of DALE is to solve the low-resource labeled data problem for specialized domains in NLP with complex and formal language, like the legal domain, and we do not focus on our pre-training corpus size. As  mentioned in our paper in Section 1, in a real-world setting, labeling data using expert annotators is an expensive affair, whereas un-labeled data is freely and openly available (even for the legal domain). This is also in-lines with prior-art on generative data augmentation for NLP [1,2,3,4,5] that often focuses on improving performance in low-resource labeled data scenarios and is generally based on a pre-trained language model, pre-trained on a huge unlabeled corpus. Since everyday natural language is quite different from  language used in legal documents, DALE pre-training can also be seen as just an extra step for effective adaptation of Pre-trained Language Models to the legal domain.
>
> - Pre-training large foundation models often consumes much more resources than used for DALE. DALE pre-training can be seen as a step for effectively adapting a PLM, originally trained on internet-scale everyday natural language data, to the legal domain. We in fact emphasize that the adaptation to the legal domain proved to be effective with only a fraction of the total unlabeled data due to the efficacy of our masking algorithm.
>
> Additionally, pre-trained DALE can also act as a foundation model for legal NLP and be used for learning other tasks beyond just data augmentation. We leave this exploration for future research.
>
> ### Questions For The Authors:
> >1.Have you experimented with smaller corpora during pre-training? How does performance degrade? What is the minimum corpus size needed?
>
> **Ans:** We thank you for the important question. Before large-scale pre-training, we had experimented with a relatively smaller dataset for pre-training. This pre-training corpus includes only ~50% of CaseLaw (cas, 2018) and ~10% of the total Pile of Law (Henderson et al., 2022) documents used in the original paper, amounting to ~1.5M total documents. Upon request, we are providing the results for the same, averaged across all datasets. We will also include these results in the original paper with the extra page allotted to us.
>
> || w/ low resource | w/ original setting |
> |-----------|-------------|----------  |
> | Gold-only  |    54.8     | 54.8     |
> | DALE       |    **55.2**    | **58.9**     |
>
>
> >2. The gains over baselines vary quite a bit across tasks - smaller for RR and DLI versus others. Is there any analysis into why certain tasks benefit more?
>
> **Ans:** We attribute this to the complexity of the RR (Malik et al., 2022) and DLI (Koreeda496 and Manning, 2021) tasks and the simplicity of the downstream learning algorithm we use from the original papers. A more detailed explanation can be found in the answer to Question which we also accompany with results using stronger downstream learning algorithms. The gains for RR and DLI get on-par with other tasks when a stronger downstream learning algorithm is used for training.
>
> >3. Did you do any error analysis? What are some common failure cases or errors you observed?
>
> **Ans:** We thank you for the questions. Most of the wrong predictions happen due to the limitation of BERT's 1024 tokens. For datasets like ILDC, SCOTUS, EULREX, and ECTHR-A/B, the model tends to predict wrong when the input gets truncated to 1024, and the majority of the information for label prediction lies in the truncated part.
>
> >4. The overfitting issue is noted for very low resources. Can you elaborate on what resource levels (50 or smaller) this tends to happen at?
>
> **Ans:** Thank You for the question. We notice some amount of overfitting (with only fine-tuned DALE and not pre-trained) with anything of 100 gold training samples and below. However, this phenomenon was only seen for datasets like UNFAIR-TOS, OTS, OTS-TOPICS, and LEDGAR, with small single sentences per training instance, and not the datasets like SCOTUS, ILDC, and EURLEX, with multi-sentence documents per training instance. As also mentioned in our paper in Section 4.4, this can be alleviated using pre-trained DALE.
>
> ### References
>
> [1] Feng, Steven Y., Aaron W. Li, and Jesse Hoey. "Keep calm and switch on! Preserving sentiment and fluency in semantic text exchange." arXiv preprint arXiv:1909.00088 (2019).
>
> [2] Kumar, Varun, Ashutosh Choudhary, and Eunah Cho. "Data augmentation using pre-trained transformer models." arXiv preprint arXiv:2003.02245 (2020).
>
> [3] Yu, Adams Wei, et al. "Qanet: Combining local convolution with global self-attention for reading comprehension." arXiv preprint arXiv:1804.09541 (2018).
>
> [4] Karimi, Akbar, Leonardo Rossi, and Andrea Prati. "AEDA: an easier data augmentation technique for text classification." arXiv preprint arXiv:2108.13230 (2021).
>
> [5] Perçin, Sezen, et al. "Combining WordNet and word embeddings in data augmentation for legal texts." Proceedings of the Natural Legal Language Processing Workshop 2022. 2022.
>
> [6] Marino, Gabriele, et al. "Automatic Rhetorical Roles Classification for Legal Documents using LEGAL-TransformerOverBERT." (2023).
>
> [7] Ivgi, Maor, Uri Shaham, and Jonathan Berant. "Efficient long-text understanding with short-text models." Transactions of the Association for Computational Linguistics 11 (2023): 284-299.

---

### Meta-Review · Area_Chair_xk5g · 2023-09-16

**Recommendation:** 3

**Metareview:**

In this paper,  the authors present a framework named DALE for generating synthetic data augmentations specifically designed for low-resource legal NLP text tasks settings. The model has designed an encoder-decoder language model pre-trained on the large-scale unlabeled legal corpus. The main contributions of this paper are proposing denoising objectives based on the selective masking technique, which masks co-occurring and highly correlated text spans instead of random words. The experiments show the state-of-the-art performance over strong baselines on 6 legal NLP tasks and 4 low-resource settings. However, the modeling process, experiments' settings and results are confusing the reviewers such as the fine-tuning step and multiple experiments, variance, and statistical significance.

---

### Decision · Program_Chairs · 2023-10-07

**Decision:**

Accept-Main

**Comment:**

In this paper,  the authors present a framework named DALE for generating synthetic data augmentations specifically designed for low-resource legal NLP text tasks settings. The model has designed an encoder-decoder language model pre-trained on the large-scale unlabeled legal corpus. The main contributions of this paper are proposing denoising objectives based on the selective masking technique, which masks co-occurring and highly correlated text spans instead of random words. The experiments show the state-of-the-art performance over strong baselines on 6 legal NLP tasks and 4 low-resource settings. However, the modeling process, experiments' settings and results are confusing the reviewers such as the fine-tuning step and multiple experiments, variance, and statistical significance.